Resource

# Carbohydrate-active enzymes from *Akkermansia muciniphila* break down mucin *O*-glycans to completion

Cassie R. Bakshani[1], Taiwo O. Ojuri[1], Bo Pilgaard[2], Jesper Holck [2],
Ross McInnes [1], Radoslaw P. Kozak[3], Maria Zakhour[4], Sara Çakaj[1],
Manon Kerouedan[1], Emily Newton[1], David N. Bolam[4] & Lucy I. Crouch [1]✉

*Akkermansia muciniphila* is a human microbial symbiont residing in the mucosal layer of the large intestine. Its main carbon source is the highly heterogeneous mucin glycoprotein, and it uses an array of carbohydrate-active enzymes and sulfatases to access this complex energy source. Here we describe the biochemical characterization of 54 glycoside hydrolases, 11 sulfatases and 1 polysaccharide lyase from *A. muciniphila* to provide a holistic understanding of their carbohydrate-degrading activities. This was achieved using a variety of liquid chromatography techniques, mass spectrometry, enzyme kinetics and thin-layer chromatography. These results are supported with *A. muciniphila* growth and whole-cell assays. We find that these enzymes can act synergistically to degrade the *O*-glycans on the mucin polypeptide to completion, down to the core *N*-acetylgalactosaime. In addition, these enzymes can break down human breast milk oligosaccharide, ganglioside and globoside glycan structures, showing their capacity to target a variety of host glycans. These data provide a resource to understand the full degradative capability of the gut microbiome member *A. muciniphila*.

The mucosal surface of the human large intestine is predominantly composed of gel-forming secreted mucins, and approximately 80% by dry weight of this glycoprotein is *O*-glycan[1,2]. Mucin *O*-glycans include only five different monosaccharides; thus, their considerable structural heterogeneity is attributed to linkage diversity between these monosaccharides and different sulfation patterns (Fig. 1). The mucosal layer is the barrier between the host epithelial cells and the dense community of microorganisms residing in the colon[3]. Mucin-degrading activities are a key aspect of the complex interactions between host and microorganism, and under healthy conditions, the production and breakdown of mucin is balanced[4]. However, disease states of the colon,

which are on the rise, are typified by the mucosal layer being disrupted and changes in the proportion of mucinolytic bacteria[4,5]. For *Akkermansia muciniphila*, multiple studies have shown that its prevalence is inversely proportional to disease and inflammation markers[6–9], but for the mucinophile *Ruminococcus gnavus*, the opposite is observed and the population increases in disease[5]. It is estimated that 60% of the human gut microbiota can access mucin as a nutrient source[10,11], yet this process and its links to disease has yet to be comprehensively understood.

*A. muciniphila* is from the relatively understudied Planctomycetota–Verrucomicrobiota–Chlamydiota superphylum[12], is detectable in

[1]Department of Microbes, Infection and Microbiomes, School of Infection, Inflammation and Immunology, College of Medicine and Health, University of Birmingham, Birmingham, UK. [2]Protein Chemistry and Enzyme Technology Section, DTU Bioengineering, Department of Biotechnology and Biomedicine, Technical University of Denmark, Lyngby, Denmark. [3]Ludger Ltd, Abingdon, UK. [4]Biosciences Institute, Medical School, Newcastle University, Newcastle upon Tyne, UK. ✉e-mail: l.i.crouch@bham.ac.uk

most people, colonizes early on in life and typically constitutes 1–3% of the total microbiota[13–15]. *Akkermansia*-like sequences have also been detected in a wide variety of vertebrates, which suggests a long evolutionary history between the mucosal surface of the gastrointestinal tract in vertebrates and *Akkermansia* species[13]. A biochemical characterization of some of the carbohydrate-active enzymes (CAZymes) from *A. muciniphila* ATCC BAA-835 (AM) has been carried out in detail to provide insights into how AM tackles this complicated structure[16–20]. Furthermore, characterization of glycopeptidases from AM has also illuminated how mucins are broken down[21–23]. However, a systematic approach to understanding mucin degradation has not previously been undertaken for any bacterial species and enzymes from AM remain uncharacterized. Here we provide a comprehensive picture elucidating how AM sequentially degrades mucin *O*-glycans, related structures and other host carbohydrates.

## Results

### In vivo studies of AM with mucin

AM is highly restricted in terms of the substrates it can access, with its main carbon source being mucin. These observations were reproduced here (Extended Data Fig. 1). RNA-sequencing was performed on porcine gastric mucin III (PGMIII) to investigate the enzymes that AM uses to break down this nutrient source (Fig. 1). We found that 20 CAZymes (from glycoside hydrolase (GH) families 2, 16, 20, 27, 29, 36, 89, 95, 97, 105, 109 and 123) and 4 sulfatases encoded in the AM genome were upregulated. These results were complementary to similar studies published previously (Supplementary Fig. 1). A pangenome analysis was also undertaken to examine the prevalence of different CAZymes throughout strains and species (Supplementary Fig. 2 and Table 1). Strikingly, most enzymes were highly conserved between strains and close homologues were identified in other species.

Whole-cell assays were used to assess enzyme activities on the surface of AM (Fig. 1 and Supplementary Figs. 3–5). Results were initially assessed using thin-layer chromatography (TLC), and a smear, increasing in concentration over time, could be identified for PGMIII. This corresponded to the typical migration pattern for glycan fragments, rather than monosaccharides (Fig. 1, Extended Data Figs. 2–4 and Supplementary Figs. 3 and 4). Released glycan fragments were labelled with the fluorophore procainamide, and detailed characterization was performed using liquid chromatography–fluorescence detection–electrospray–mass spectrometry (LC–FLD–ESI–MS). The data reveal that a range of mucin *O*-glycan fragments are produced by the enzymes on the surface of AM (Fig. 1 and Extended Data Figs. 2–4). Galactose is typically present at the reducing end, indicative of GH16 endo-*O*-glycanase activity, and the degree of sulfation and fucosylation

is variable[18]. We also used PGMIII-grown cells to carry out whole-cell assays against several defined oligosaccharides (Supplementary Figs. 3 and 4). Activities were observed against TriLacNAc, human milk oligosaccharides (HMOs), Forssman antigen (FORS1) and lacto-*N*-biose. Whole-cell assays of PGMIII-grown cells against bovine submaxillary mucin (BSM), which has core 1 decorations[24], showed only the release of sialic acids, and this was confirmed by high-performance anion exchange chromatography with pulsed amperometric detection (HPAEC-PAD; Extended Data Fig. 5).

### AM CAZymes can get down to the core GalNAc in PGMIII

Two putative GH31 enzymes are encoded in the AM genome. Amuc_1008[GH31] is from subfamily 18, and other members of this subfamily, from *Bacteroides caccae*, *Phocaeicola plebius*, *Enterococcus faecalis*, *Clostridium perfringens* and *Bombyx mori* (domestic silk moth), have specificity for removing α-linked *N*-acetylgalactosamine (GalNAc) from peptide[25–27]. A phylogenetic tree of characterized bacterial GH31 enzymes shows clustering of activities, and Amuc_1008[GH31] clusters with the other characterized enzymes from GH31_18 (Supplementary Fig. 6). We also found that Amuc_1008[GH31] had specificity for the core α-GalNAc linked to peptide (BSM) with no activity against any other substrates (Supplementary Figs. 7–10). Quantification of GalNAc release using HPAEC-PAD showed that Amuc_1008[GH31] removed the highest concentration of all the enzymes tested here (Extended Data Fig. 5). Notably, Amuc_1008[GH31] could not hydrolyse the Tn antigen and therefore requires more than one amino acid for its activity. For the more complex substrate PGMIII, Amuc_1008[GH31] could not liberate GalNAc when tested in isolation; however, when a cocktail of AM enzymes was used, GalNAc could then be released (Fig. 1). This observation shows that enzymes characterized from AM are capable of complete *O*-glycan degradation down to the polypeptide. This enzyme is predicted to be periplasmic, and there was no obvious removal of GalNAc from BSM during whole-cell assays, supporting this prediction (Supplementary Fig. 5). A comparison of the Amuc_1008[GH31] model and the *E. faecalis* GH31_18 is provided in Supplementary Discussion and Supplementary Fig. 11.

### Hydrolysis of α-linked galactose from PGMIII *O*-glycans

Galactose, GalNAc, *N*-acetylglucosamine (GlcNAc), fucose and sialic acid all cap mucin *O*-glycans via α-linkages at the non-reducing end, and the prevalence of these monosaccharides will vary according to mucin type and genetics, for instance. Here we determined which enzymes AM uses to tackle these different capping monosaccharides by using a panel of different substrates of varying complexity in overnight end-point assays.

---

**Fig. 1 | Activity of the recombinantly expressed glycoside hydrolases from AM against α-linked monosaccharides. a**, Structural features that are expected in natural secreted mucin glycoproteins with epitopes highlighted. Only α-linkages are labelled apart from the core GalNAc monosaccharides that are also α-linkages. The structure of an *O*-glycan chain is generally accepted to be categorized into three sections: (1) the core, consisting of an α-linked GalNAc attached to a serine or a threonine, (2) the polyLacNAc extensions linked to the core GalNAc and (3) the terminal 'capping' epitopes. The polyLacNAc extensions are generally LacNAc disaccharides linked through β1,3-bonds, but variable sulfation, fucosylation and branching add to the complexity and heterogeneity along these chains. Key ganglioside and globoside structures are also included. **b**, Volcano plot highlighting the differential gene expression of AM when grown with PGMIII compared with glucose. Genes that did not pass the threshold of significance (FDR (false discovery rate; Benjamini–Hochberg) < 0.05) are coloured grey. Genes that passed the threshold of significance (FDR < 0.05) but had a $\log_2$(fold change (FC)) between −1.5 and 1.5 are coloured blue. Genes that were significantly differentially expressed (FDR < 0.05) and had a $\log_2$(FC) <−1.5 or >1.5 are coloured red. Enzymes from this study that were significantly differentially expressed are labelled. The total number of genes in the analysis was 2,171. **c**, TLC of the whole-cell assay of PGMIII-grown AM against fresh PGMIII.

A smear can be seen increasing in concentration over time. The glycans in these samples were then labelled with procainamide and analysed by LC–FLD–ESI–MS. The chromatogram for the 9 h sample is shown, and the different glycan peaks are labelled. The data for all the time points are in Supplementary Fig. 6. The data allow the reconstruction of the order of monosaccharides in an oligosaccharide but do not provide information about linkages. It is also not always possible to tell where a fucose or sulfate group is along the chain. Example mass spectra and MS–MS fragment data are presented in Extended Data Figs. 2 and 3. **d**, A cocktail of enzymes from *A. muciniphila* BAA-835 can completely degrade the *O*-glycans from PGMIII down to the core GalNAc. The top and bottom panels are a TLC and HPAEC-PAD of the results, respectively. PGMIII was incubated with a cocktail of enzymes (in the pale-blue box), and the reaction terminated by boiling (1). The result of this reaction was a large amount of monosaccharides that can be seen on the TLC and by HPAEC-PAD. This reaction was then dialysed to remove all free monosaccharides and glycans (2) and concentrated (3). Untreated PGMIII was also included as a control (4). The application of Amuc_1008[GH31] is indicated by plus signs, and in samples 3 and 4, GalNAc can be seen to be released. Standards have also been included. Enzyme assays were carried out at pH 7 and 37 °C, overnight, and with 1 μM enzymes. Neu5Ac, *N*-acetylneuraminic acid; Neu5Gc, *N*-glycolylneuraminic acid; o/n, overnight.

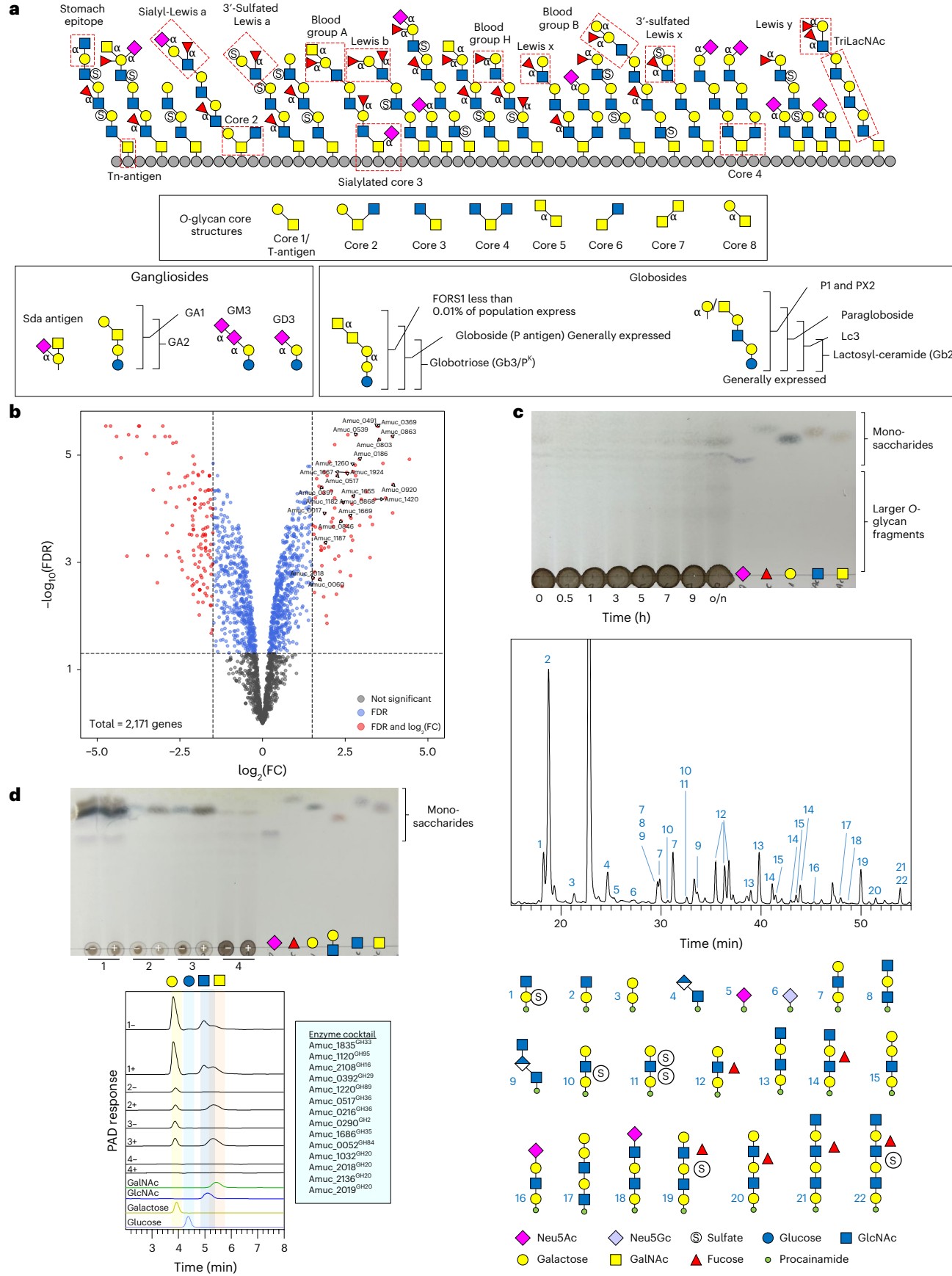

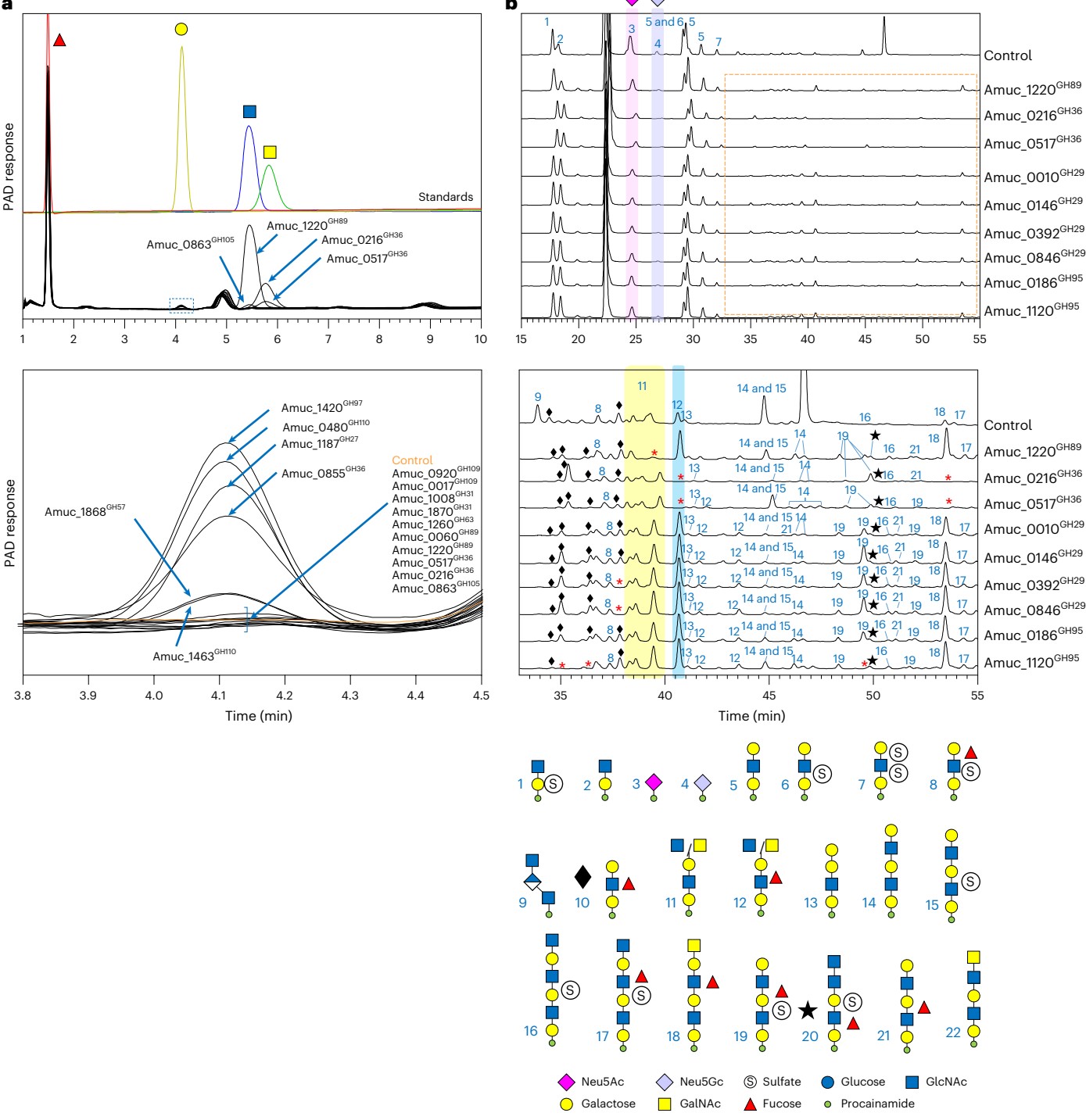

**Fig. 2 | Activity of the recombinantly expressed glycoside hydrolases from AM against α-linked monosaccharides capping mucin. a**, Activity of enzymes against PGMIII (pre-treated with Amuc_1835^GH33 and Amuc_1120^GH95) and analysed using HPAEC-PAD. Standards were also run to identify monosaccharide products. Top panel: relevant area of chromatograms. Bottom panel: enlarged image of the area where galactose elutes, showing details (dashed blue box in the top panel). **b**, Activity of enzymes releasing α-GlcNAc, α-GalNAc and α-fucose from GH16-released *O*-glycans from PGMIII. Top panel: full chromatograms with the two types of sialic acid highlighted in pink and purple. Bottom panel: enlarged image of the smaller peaks (highlighted by the dashed orange box in the top panel). Black diamonds and black stars indicate glycan number 10 and number 20 peaks, respectively. Highlighted in yellow and blue are the number 11 and number 12 peaks. The red asterisks indicate where a glycan is absent.

There are two putative GH110 enzymes (Amuc_0480^GH110 and Amuc_1463^GH110) encoded by the AM genome, which share 28% identity. These enzymes showed activity towards blood group B (BGB) types I and II and α1,3-linked galactose (Gal) to either Gal or GalNAc (Extended Data Fig. 6 and Supplementary Figs. 7–9). BGB is slightly less prevalent than blood group A (BGA; 10–40%) in the human population, ranging between 0 and 30%, depending on geographical location[28]. These are the only enzymes that are active against BGB from AM, which predicate the removal of fucose by the fucosidases. The structure of Amuc_1463^GH110 has recently been solved[20].

One putative GH27 (Amuc_1187[GH27]) is encoded in the AM genome and was active against α-linked galactose from the non-reducing end of most of the defined substrates tested here, except BG structures. Amuc_1187[GH27] can hydrolyse Galα1,3-Gal/GalNAc and globotriose, but the P1 antigen could not be completely broken down in an end-point assay. This substrate specificity is discussed in the context of a model of Amuc_1187[GH27] compared with solved structures of enzymes from *Homo sapiens* (Supplementary Discussion and Supplementary Fig. 11).

AM also has three putative GH36 enzymes encoded in the AM genome, which have low sequence homology (Supplementary Table 2), and all three cluster in different locations on a phylogenetic tree when compared with characterized GH36 enzymes (Supplementary Fig. 12). Amuc_0855[GH36] has specificity for the Galili antigen and globotriose in the defined substrate screen. Finally, AM also has one putative GH97 enzyme (Amuc_1420[GH97]), which showed a preference for α-linked galactose in the defined substrate screen, except in the context of BGB. A model of Amuc_1420[GH97] is discussed in the context of its activity (Supplementary Discussion and Supplementary Fig. 13). All these enzymes were able to remove galactose from PGMIII, albeit in relatively small amounts, but this is likely because of the type of mucin used. The identity of the galactose was confirmed by HPEAC (Fig. 2 and Supplementary Fig. 14).

### Hydrolysis of α-linked GalNAc from PGMIII *O*-glycans

The AM enzymes with activity against α-GalNAc capping structures are from families GH36 and GH109. All four enzymes were active against BGA structures, and they must remove this sugar before the fucosidases can act. Amuc_0216[GH36] was able to act on all the substrates with α-GalNAc decorations we could test (Extended Data Fig. 6 and Supplementary Figs. 7–9). Notably, this was the only enzyme we found in AM to degrade the Tn antigen. A phylogenetic tree of currently characterized bacterial GH36 enzymes, including those from the AM genome described here, showed that Amuc_0216[GH36] clustered with the other two examples of α-GalNAcases in the family (Supplementary Fig. 12). One of these previously characterized enzymes also has activity on BGA and has been shown to convert BGA whole blood to universal-type blood for transfusion applications[29]. Building on this application, recent work has also shown that some of the AM CAZymes can be used to generate universal blood[20]. Amuc_0517[GH36] was much narrower in its strict specificity for BGA only. There are two putative GH109 enzymes (Amuc_0017[GH109] and Amuc_0920[GH109]) encoded by the AM genome, which have 66% identity between them and show specificity towards α-GalNAc substrates apart from the Tn antigen. When tested against PGMIII, both GH36 enzymes could release relatively large amounts of GalNAc, with or without sialidase and fucosidase pre-treatment, and this was confirmed using HPAEC-PAD (Fig. 2 and Supplementary Fig. 14).

### Hydrolysis of α-linked GlcNAc from PGMIII *O*-glycans

There are two putative GH89 enzymes (Amuc_0060[GH89] and Amuc_1220[GH89]) encoded by the AM genome, and both showed activity only against GlcNAcα1,4-Gal disaccharide (stomach epitope); however, Amuc_0060[GH89] could not hydrolyse all the GlcNAcα1,4-Gal overnight, which suggests that this is not the preferred substrate (Extended Data Fig. 6 and Supplementary Figs. 7–9). Amuc_1220[GH89] was able to remove relatively large amounts of GlcNAc from PGMIII, and this was confirmed using HPAEC-PAD (Fig. 2 and Supplementary Fig. 14). Further discussion on models of these enzymes compared with solved structures is presented in Supplementary Discussion and Supplementary Fig. 15.

### Hydrolysis of sialic acid and fucose PGMIII *O*-glycans

AM has three sialidases and six fucosidases that together can tackle a wide range of substrates, and their specificities have been characterized in depth previously[16,17]. The specificities observed for these enzymes presented in this report are comparable to previous characterizations, but we were able to identify notable further observations

(Extended Data Figs. 7 and 8 and Supplementary Fig. 16). In terms of the sialidases, we found that the two GH33 enzymes from AM can act on the relatively complex GD1a and GT1b ganglioside structures and Sda antigen, which are common features of the human glycome[30,31]. In terms of the fucosidases, Amuc_0010[GH29] shows specificity towards type 2 BG structures (not type 1) and the lacto-*N*-neotetraose HMO series (not the lacto-*N*-tetraose), whereas Amuc_1120[GH95] can hydrolyse fucose from both type 1 and 2 structures. This specificity facilitates the probing of mucin glycan structures in more detail. For instance, only Amuc_1120[GH95] can hydrolyse fucose from untreated PGMIII, indicating that this substrate has only type 1 structures immediately available to the fucosidases at the non-reducing ends of the *O*-glycans (Extended Data Fig. 8).

### Combining endo-acting and exo-acting CAZymes

To further explore the specificities of the CAZymes removing α-linked monosaccharides from PGMIII, we used a series of sequential reactions, which were then labelled with procainamide and analysed by LC−FLD−ESI−MS (Fig. 2 and Supplementary Fig. 17). Complementary to the relatively high monosaccharide release seen by HPAEC-PAD, the LC−FLD−ESI−MS data show that Amuc_1220[GH89], Amuc_0216[GH36] and Amuc_0517[GH36] could act on GH16-derived *O*-glycan fragments (Fig. 2). These assays therefore provide the ability to characterize the fragments produced by GH16. For example, glycan number 11, which is absent in the Amuc_1220[GH89] assay, has a GlcNAcα1,4-Gal capping structure. This also confirms that GH16 can accommodate this epitope in its active site. Amuc_0216[GH36] and Amuc_0517[GH36] both acted on glycans number 12 and 18, confirming that they have α-GalNAc caps. Furthermore, we could also observe four of the fucosidases acting on GH16-derived number 10 *O*-glycan fragments. The different peaks corresponding number 10 glycan compositions will be different combinations of linkages and positioning of the fucose. For instance, the structures hydrolysed by Amuc_1120[GH95] are not hydrolysed by Amuc_0010[GH29], thus confirming them as type 1 α1,2-fucose structures. In addition, Amuc_0392[GH29] and Amuc_0846[GH29] both hydrolyse a different glycan number 10 (eluting at a different time to the glycan number 10 acted on by Amuc_1120[GH95]) complementary to their comparable activities seen in the defined substrate screen.

### Investigating β-galactosidase activity

The combination of GH16 endo-*O*-glycanase activity and removal of α-capping monosaccharides from the non-reducing ends lead us to how the remaining *O*-glycan fragments will be broken down. This will require contribution from β-galactosidases, β-HexNAcases, fucosidases and sulfatases. There are nine putative CAZymes encoded in the AM genome that were highlighted as possible β-galactosidases from families GH2, GH35 and GH43 subfamily 24 (Supplementary Table 3). Recombinant enzymes were screened against a variety of substrates to determine their specificity (Fig. 3 and Supplementary Figs. 18–24). The results revealed a range of specificities, but between them, the β-galactosidases could break down all the defined oligosaccharides tested. There are examples of very broad-acting (Amuc_0290[GH2]) and highly specific β-galactosidases (Amuc_0539[GH2] was found to be specific to β1,4-linked Gal with either a Gal or GalNAc in the +1 position). None of these enzymes showed activity towards 3-fucosyllactose (3-FL), Lewis A- or Lewis X-based structures, which confirms that there is a strict order of degradation, with fucosidases acting on these substrates first, followed by galactosidases.

We tested this panel of β-galactosidases against PGMIII that had been sequentially treated with different AM CAZymes. It should be noted that, although with the addition of only fucosidase, sialidase and a GH16 a range of sizes and compositions of *O*-glycan fragments were produced, the majority of these were disaccharides and trisaccharides of alternating galactose and GlcNAc, ascertained by the intensity of the fluorescence signals being relatively high for these structures

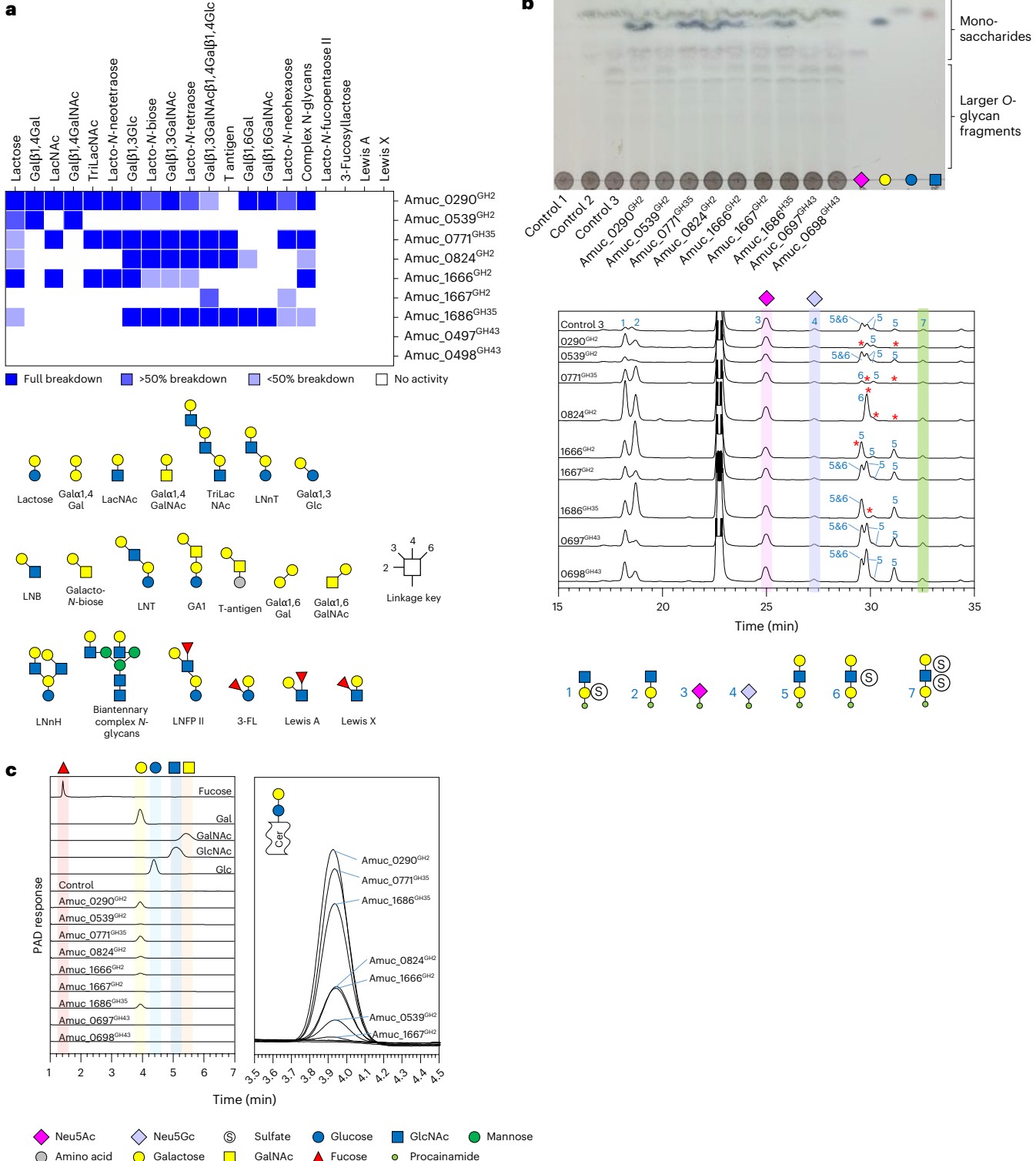

**Fig. 3 | Activity of the glycoside hydrolases from families 2, 35 and 43 from AM against β-linked galactose substrates. a**, A heat map of recombinant enzyme activities against defined oligosaccharides. The dark blue and white indicate full and no activity, respectively, and partial activities are represented by the lighter blues. Partial activity is when all the substrate has not been broken down in an end-point assay. **b**, The activity of the panel of β-galactosidases against PGMIII that had been sequentially degraded. Control 1, no enzymes added; control 2, Amuc_1835GH33 and Amuc_1120GH95; control 3, control 2 plus Amuc_2108GH16. The top and bottom panels are the TLC and the LC–FLD–ESI–MS results, respectively. These assays were performed in stages, so the reactions were boiled in between steps. Enzyme assays were carried out at pH 7 and 37 °C, overnight and with 1 µM enzymes. Monosaccharide standards are shown in the right lanes. **c**, Activity of the GH2, GH35 and GH43 enzymes against lactosylceramide. The assays were analysed using HPAEC-PAD, and the different controls confirm the release of galactose. Left panel: the different controls and samples are stacked. Right panel: the assay chromatograms are overlaid for comparison so the galactose peaks can be observed in more detail. The substrate could not be resolved using this method, so whether the reaction had gone to completion could not be determined and TLC of the samples was also inconclusive. Enzyme assays were carried out at pH 7 and 37 °C, overnight and with 1 µM enzymes. LNT, lacto-*N*-tetraose; LNnH, lacto-*N-neo*hexaose; LNFP II, lacto-*N*-fucopentaose II; Cer, ceramide.

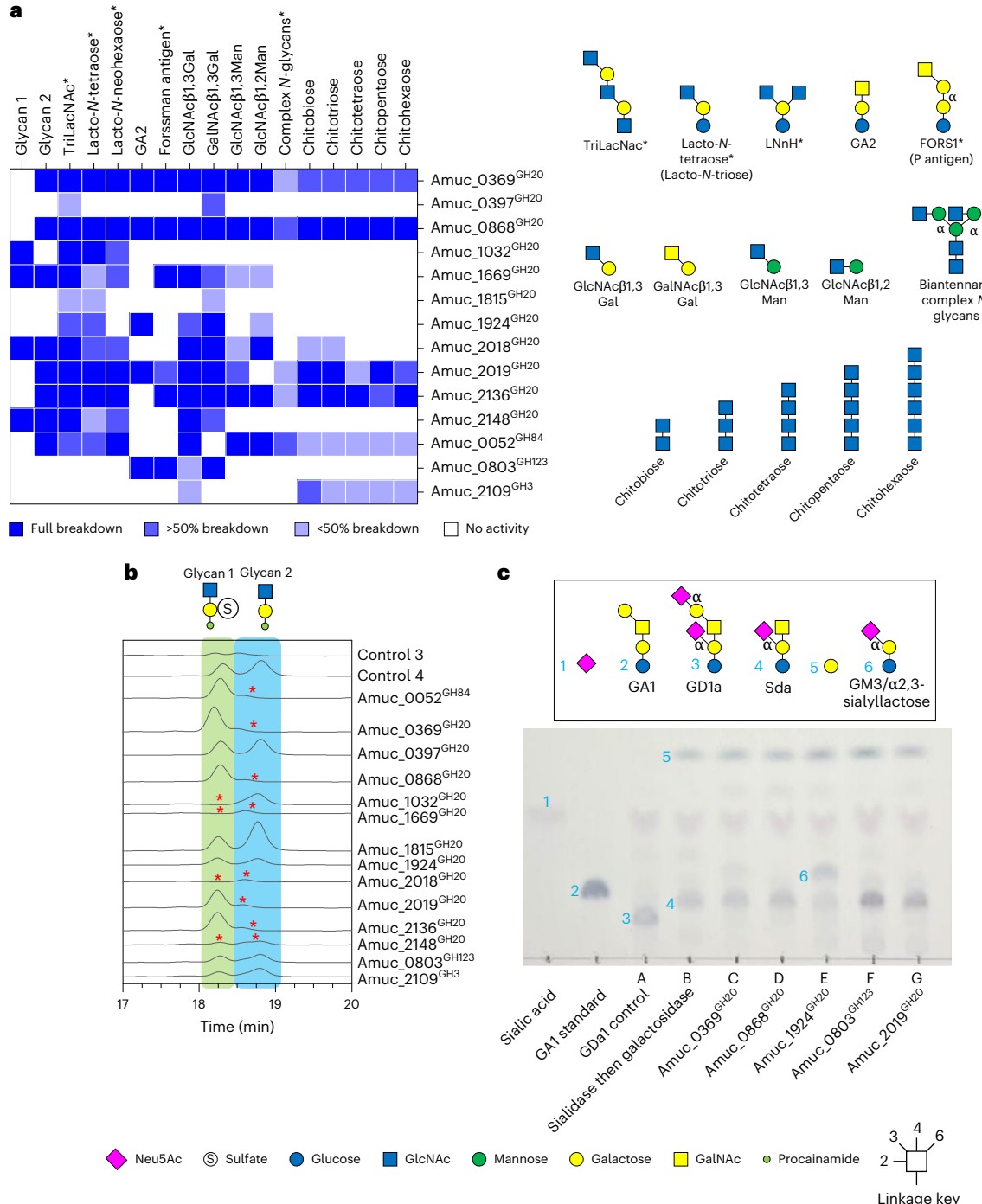

**Fig. 4 | Activity of β-HexNAcases from glycoside hydrolase families 20, 84, 123 and 4 from AM. a**, Heat map of recombinant enzyme activities against defined oligosaccharides. The dark blue and white indicate full and no activity, respectively, and partial activities are represented by the lighter blues. Asterisks represent those substrates that were generated by pre-treating with other CAZymes. **b**, The activity of the panel of β-HexNAcases against PGMIII that had been sequentially degraded. Control 1, no enzymes added; control 2, Amuc_1835[GH33] and Amuc_1120[GH95]; control 3, control 2 plus Amuc_2108[GH16]; and control 4, control 3 plus Amuc_0771[GH35]. The relevant area of the chromatograms of the LC–FLD–ESI–MS data are shown, and the two glycans are highlighted. Red asterisks indicate where a glycan is not present. These assays were performed in stages, so the reactions were boiled in between steps. Enzyme assays were carried out at pH 7 and 37 °C, overnight and with 1 μM enzymes. **c**, TLC results of sequential assays against ganglioside GD1a. GDa1 was sequentially treated with Amuc_1835[GH33] and then Amuc_0771[GH35] (lane B). The sample was then boiled and a panel of β-HexNAcases added.

(Fig. 3 and Supplementary Fig. 22). When the panel of β-galactosidases was tested, the sample was pre-treated with the three enzymes that remove α-linked GlcNAc and GalNAc (Amuc_1220[GH89], Amuc_0216[GH36] and Amuc_0517[GH36]) and a fucosidase (Amuc_0392[GH29]) to maximize the O-glycans with β-galactose at the non-reducing ends. A TLC of

these assays clearly shows galactose being released for five of the enzymes (Fig. 3, blue-stained bands). These samples were labelled with procainamide and analysed by LC–FLD–ESI–MS (Fig. 3 and Supplementary Fig. 22). Activities for four of the β-galactosidases were observed against the trisaccharides. Amuc_0539[GH2] showed no activity, and this

supports the observation of only being active when a Gal or GalNAc is in the +1 subsite. Notably, doubly sulfated trisaccharide glycan number 7 was not acted upon by any β-galactosidase. The GH2 and GH35 enzymes also showed activity against lactosylceramide (ganglioside and globoside core structure), with Amuc_0290[GH2], Amuc_0771[GH35] and Amuc_1686[GH35] releasing the most galactose (Fig. 3).

### Investigating β-HexNAc activities

The GH20 family is the largest represented family in AM, with 11 putative enzymes encoded with generally low identity between sequences (Supplementary Fig. 25 and Table 4). The activity predominantly observed in this family is exo-acting β-GlcNAcases, but the activity on GalNAc has also been observed. We also included Amuc_0052[GH84], Amuc_0803[GH123] and Amuc_2109[GH3] in this screen, with the latter having been shown to have relevant activities (pNPGlcNAc>GalNAc)[32] and clusters with other β-HexNAcases on a phylogenetic tree when characterized enzymes were compared (Supplementary Fig. 26). In summary, we found a range of different β-HexNAcase specificities encoded by the AM genome and the specificities of these enzymes are generally towards the glycan structures found in mucins rather than complex N-glycans or chitin. Between them, these β-HexNAcases could access all the defined substrates provided (Fig. 4 and Supplementary Figs. 27–29). The panel of β-HexNAcases were then tested against PGMIII that had been sequentially treated (controls 1–4; Supplementary Fig. 17). The samples were then labelled with procainamide and analysed by LC–FLD–ESI–MS. The predominant glycans in 'control 4' were two disaccharides, one sulfated on the galactose and one not (Fig. 4). Four of the β-HexNAcases were able to break down the sulfated disaccharide, providing further insight into the different specificities of these enzymes. To further explore the capacity of β-HexNAcases against host glycan structures, we used the AM CAZymes to prepare Sda antigen from the ganglioside GD1a. Intriguingly, out of the five β-HexNAcases that could act on GA2, only Amuc_1924[GH20] was able to accommodate the branching sialic acid to access the GalNAc present in the Sda antigen in an end-point reaction (Fig. 4). In addition to the sequential degradation of mucin O-glycans, the activity of the AM CAZymes against HMOs was also explored and is detailed in Supplementary Discussion and Supplementary Figs. 30 and 31.

### Tackling sulfated substrates

We highlighted 12 potential sulfatases from the AM genome using upregulation data[4,33–35] and the SulfAtlas database[36]. These enzymes could be assigned to seven different families and one unknown, which ultimately turned out not to be a sulfatase (Supplementary Table 1). The different subfamilies cluster on a phylogenetic tree, and the activities (mainly explored in *Bacteroides thetaiotaomicron*) within a subfamily correspond to those already described (Supplementary Fig. 32)[37,38]. The sequence identity between the sulfatases is low, with the highest being ~60% for pairs in the same subfamily (Supplementary Table 5). The defined sulfated substrates tested were sulfated monosaccharides and Lewis structures (Extended Data Fig. 9 and Supplementary Figs. 33 and 34). We first screened the sulfatases against the sulfated monosaccharides and found that Amuc_1655[Sulf16] and Amuc_1755[Sulf16] showed activity against 4S-Gal; Amuc_1755[Sulf16] also showed activity against 4S-GalNAc. The subfamily 11 enzymes Amuc_1033[Sulf11] and Amuc_1074[Sulf11] showed activity against 6S-GlcNAc, and Amuc_0121[Sulf15] showed partial activities against 6S-Gal and 6S-GalNAc. These specificities align with those previously observed for these subfamilies[37].

We then used combinations of the sulfatases, α-fucosidases and β-galactosidase encoded in the AM genome to determine how the different sulfated Lewis structures are broken down. For the 3′S and 6S-Lewis A and X glycans, Amuc_0392[GH29] removes the fucose to produce 3′S and 6S-N-acetyllactosamine (LacNAc) and lacto-N-biose (LNB). From here, the 3′S disaccharides are then acted on by sulfatases from subfamily 20. By contrast, the 6S disaccharides are acted on by the

β-galactosidases first. In this study, we could not find a way to break down the 6′S-Lewis structures. Previous activities against these substrates have been observed for BT1624 from subfamily 15 (ref. 37), so we had predicted that Amuc_0121[Sulf15] may be able to do this, but only partial activity against sulfated galactose could be detected.

### AM interactions with GAGs, N-glycoproteins and starch

There are two observations from this study worthy of note, concerning the ability of AM to colonize the mucosal surface and interact with other members of the microbiota. Firstly, AM did not grow directly on any of the glycosaminoglycans (GAGs) tested (Extended Data Fig. 1). However, its genome encodes Amuc_0778[PL38] and Amuc_0863[GH105] and activity against all GAGs except heparin sulfate was observed for these enzymes (Extended Data Fig. 10, Supplementary Figs. 35–37 and Supplementary Tables 6 and 7). Secondly, AM grew on a high-mannose N-glycoprotein, and by analysing substrates remaining in the broth, we deciphered that AM was using the protein component and leaving the glycan (Supplementary Fig. 38).

In total, we were unable to find activities for 11 of the putative CAZymes and 3 of the sulfatases during this study (Supplementary Table 1). Some of these CAZymes are from the GH13 and 77 families, which are currently solely associated with breaking down α-glucose polymers (Supplementary Fig. 39).

## Discussion

The human gut microbiota plays an intrinsic role in health and disease, with impacts now recognized to extend to many other parts of the body. One of the key processes occurring at the interface between microorganisms and host is the breakdown of mucins, and this holistic investigation of AM CAZymes described here is an important step forwards in understanding this complex relationship. Previously, we characterized the GH16 endo-O-glycanases from AM (Amuc_0724[GH16], Amuc_0875[GH16], Amuc_2136[GH20]) that hydrolyse within the polyLacNAc chains of mucin to produce fragments of O-glycans[18]. Here we expand on this work by taking a systematic approach to assessing the full enzymology of this organism. The capacity of the AM CAZymes to break down most of the substrates tested reveals how AM has adapted to accessing mammalian-derived glycans. The results have allowed an order of enzyme activities on different substrates to be assembled (Fig. 5), a sequential breakdown of complex substrates to be demonstrated and the determination that a cocktail of AM CAZymes could be applied to reach the core GalNAcs of a complex mucin substrate. A model of how AM degrades mucin can now be proposed (Fig. 6).

A limitation of this work was the use of porcine gastric mucin rather than human mucin native to AM owing to lack of availability. Future work will aim to explore human-derived mucins with CAZymes from gut microorganisms as these substrates become available. Greater availability of mucin-type substrates will also allow a more thorough exploration of mucin-targeting enzymes and organisms, such as those described here. Another limitation of this work was the approach we took to analysing AM in isolation and away from a community of microorganisms and the host environment. Our approach is useful in providing clear enzymology, but neglects to understand the role and importance of these enzymes in the context of their native environment. However, due to mucin being the sole carbon source for AM, understanding the enzymology will be critical to understanding the complicated impact of AMs on health and disease[39,40].

The glycomes of different humans vary considerably, and characterization of enzymes with specificity towards different structures will now enable research into these epitopes much more broadly. For instance, blood groups are a recognizable example, which varies globally[41], but Galα1,4Gal epitopes of P1 and P[k] antigens are also generally present in the mucosal surface of the human large intestine. These glycans, and others like them, are receptors for a variety of pathogens

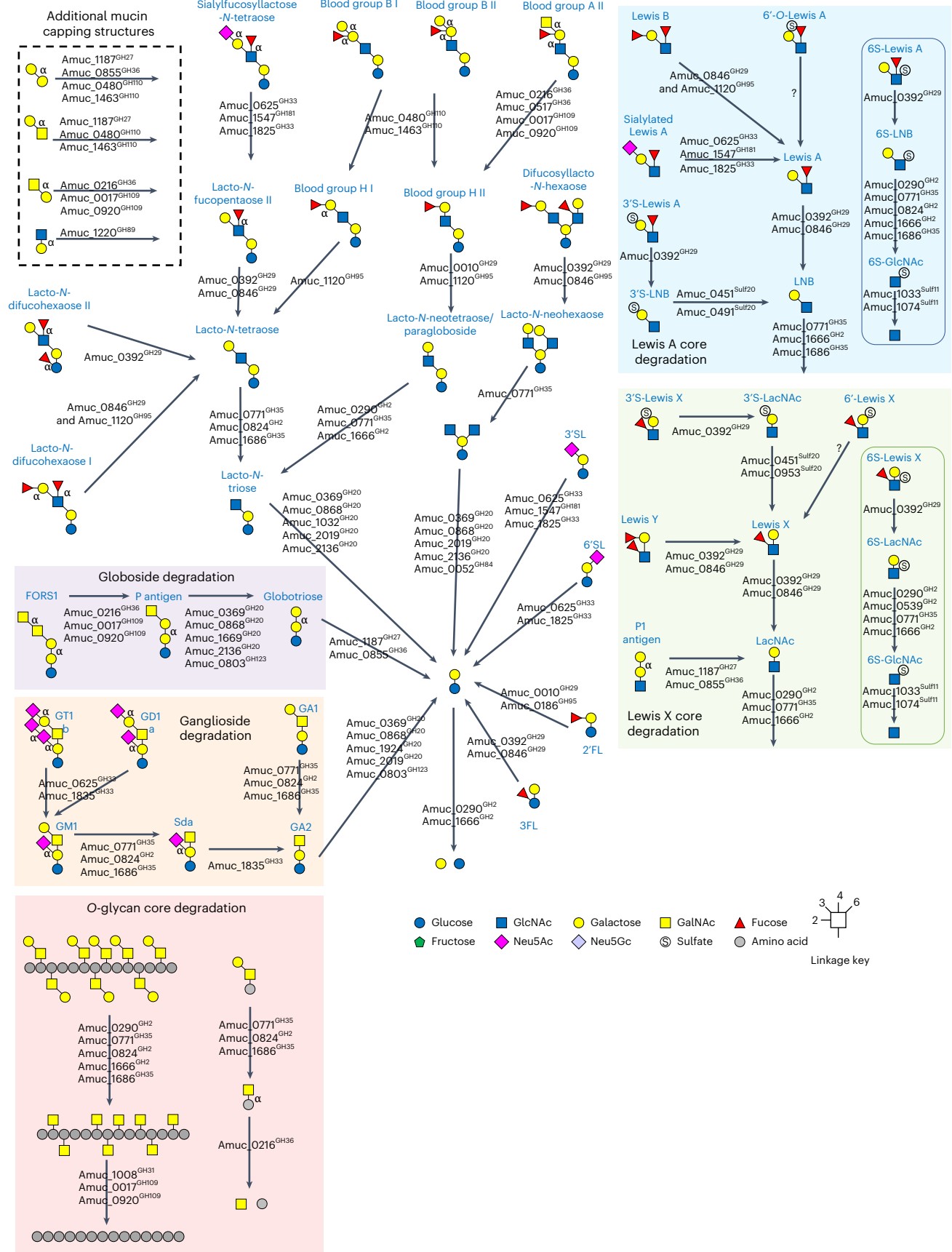

**Fig. 5 | Illustration of the CAZymes characterized in this report against host glycans by AM.** The different types of substrate are grouped where possible. The enzymes listed for each reaction are the ones that will work alone, but for the Lewis B structures, 'and' signifies that both enzymes are required to remove the fucose. For the sialylfucosyllacto-*N*-tetraose (SFLNT), the possible fucosidase activities were not included, but Amuc_0392^GH29 can act on this substrate also. Only α-linkages are labelled apart from the core GalNAc monosaccharides that are also α-linkages.

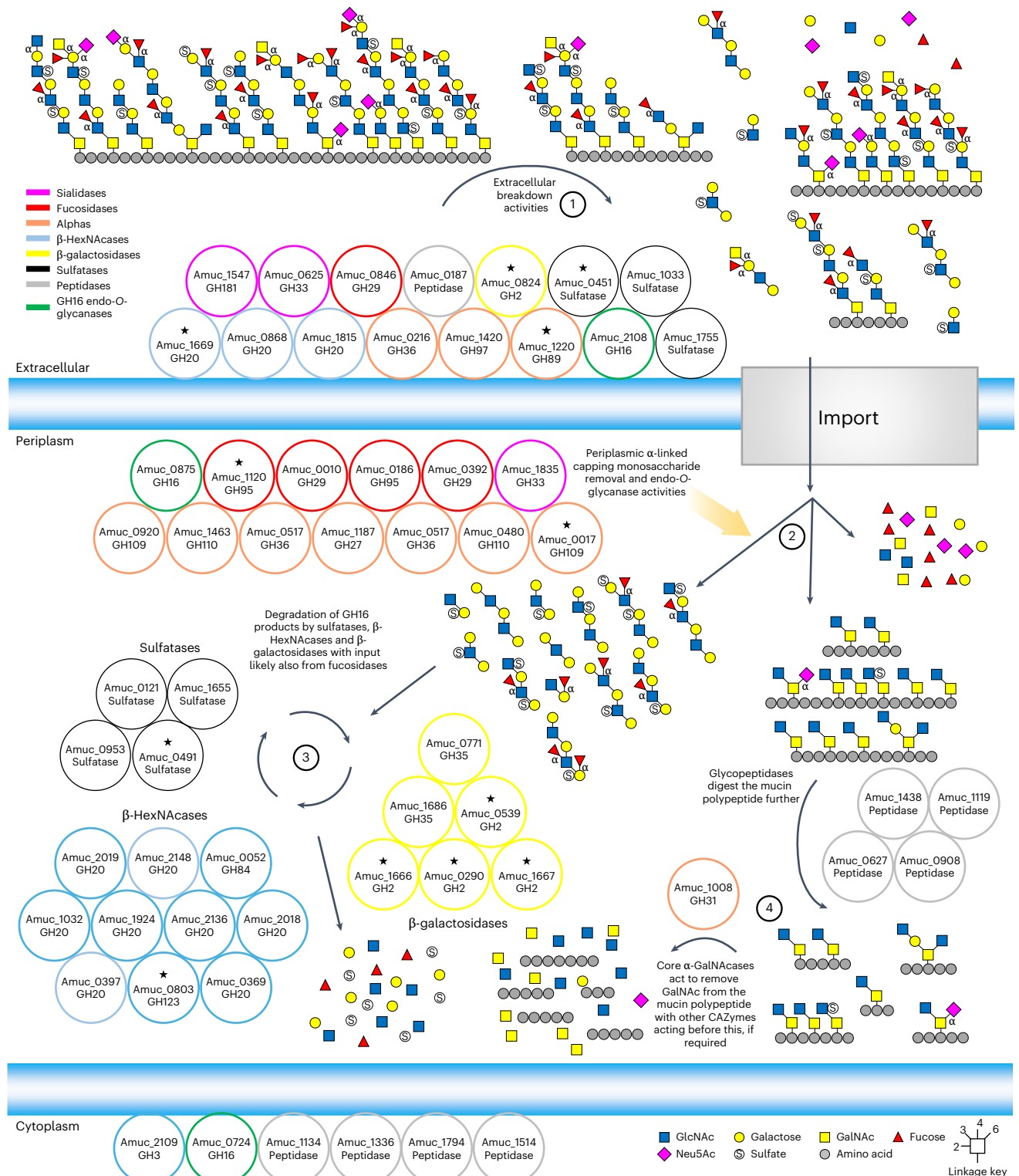

**Fig. 6 | A model for the degradation of mucins by *A. muciniphila* based on current understanding.** The GH enzymes included are only where activity has been observed, and the colour indicates the type of activity (see key). The localization of all the enzymes is based on the SignalP 6.0 prediction, apart from Amuc_2108[GH16] owing to it being observed in outer membrane analysis when AM was grown on mucin[65], and our whole-cell assays support a GH16 being localized to the outside of the cell. The peptidases included are both those that have been characterized and those that have been highlighted as upregulated on mucin. The signal sequences of a peptidase and sulfatases potentially localize them to the outside of the cell also. The numbers indicate the order that mucin is degraded. 1: There is some processing of mucin on the surface, with both exo- and endo-GH activities. Large sections of mucin are then imported into the cell for further breakdown. 2: Initially, there will be further exo- and endo-GH activities to produce fragments of *O*-glycans and mucin polypeptide with only the core glycan decoration remaining. 3: The *O*-glycan fragments will then be broken down to monosaccharides through the alternating action of sulfatases, fucosidases, β-galactosidases and β-HexNAcases. 4: The remaining glycopeptides are known targets for characterized *A. muciniphila* glycopeptidases, and Amuc_1008[GH31] will remove core GalNAc from the polypeptide. The stars indicate Mul1A association.

and their toxins, such as Shiga toxin[42]. FORS1 is normally present in only 0.01% of the human population, but in that relatively small population, this antigen would be present in the gastrointestinal tract[43]. It can change the susceptibility to some diseases and is also expressed in some cancers, but its true prevalence and impact on different diseases are understudied[44]. One of the major impacts of this work will be the ability to characterize different mucins in much more detail and in a higher-throughput manner. We hope that these enzymes provide new glycotools for the community and have far-reaching applications, such as monitoring the change in mucin *O*-glycans between healthy and diseased tissue or the detection of disease-associated epitopes.

## Methods

### Substrates

A list of substrates and their sources are listed in Supplementary Data 2. PGMIII (Sigma) was prepared by dissolving it overnight in sterile deionized water. This was too turbid for bacterial growth, so the precipitate was removed by centrifugation at $1,500 \times g$ for 5 min in sterile 15 ml Falcon tubes. Polyglucuronic acid was prepared as described[45].

### Recombinant protein expression and purification

Recombinant plasmids were transformed into Tuner cells (Novagen) in Luria–Bertani broth containing 50 µg ml$^{-1}$ or 100 µg ml$^{-1}$ kanamycin or ampicillin, respectively. The plates were used to inoculate 1 l flasks (also including relevant antibiotics), and the cells were grown at 37 °C with shaking at 180 rpm to mid-exponential growth phase; the flasks were then cooled to 16 °C, and isopropyl β-D-thiogalactopyranoside was added to a final concentration of 0.2 mM. Recombinant His-tagged protein was purified from cell-free extracts using immobilized metal affinity chromatography (using Talon resin; Takara Bio). The buffer used during the purification process was 20 mM Tris–HCl and 100 mM NaCl, pH 8. Cell-free extracts (~30 ml) were passed through a 5 ml bed volume of resin. This was then washed with 50 ml of buffer and the protein eluted with 10 mM and then 100 mM imidazole. The purification is assessed by sodium dodecyl sulfate–polyacrylamide gel electrophoresis (precast 4–20% gradient from Bio-Rad and stained with Coomassie Brilliant Blue), but proteins typically elute in the 100 mM fraction. The proteins were then concentrated using centrifugal concentrators (Vivaspin, 10 kDa molecular weight cut off), the absorbance was determined at 280 nm (NanoDrop One, Thermo Scientific) and concentrations were calculated using extinction coefficients taken from ProtParam.

### Recombinant enzyme assays

The concentrations of defined oligosaccharide substrates used in the assays are presented in Supplementary Table 2. The activities of the recombinant enzymes were typically assessed in 20 mM 4-morpholinepropanesulfonic acid (pH 7) at 37 °C and a final enzyme concentration of 1 µM. The volume varies according to how much was required for subsequent analysis. For the sulfatase assays, 5 mM of CaCl was included. PNGaseL (LZ-PNGaseL-50-KIT) was used where required.

### TLC

For defined oligosaccharides, 3 µl of an assay containing 1 mM substrate was spotted onto silica plates (Sigma Z740230). For assays against glycoproteins, GAGs and starch substrates, this was increased to 9 µl, 6 µl and 9 µl, respectively. The plates were resolved in running buffer containing butanol, acetic acid and water. Different ratios were used for different substrates—2:1:1 (typically resolved once) was used for most of the defined oligosaccharides, and 1:1:1 (typically resolved twice) was typically used for substrates in which large products were expected (for example, PGMIII). All TLCs were stained using a diphenylamine–aniline–phosphoric acid stain[46].

### Growth of AM on a variety of substrate

AM was grown on chopped meat broth (CMB) as described in ref. 18. Overnight starter cultures were inoculated from a glycerol stock into 5 ml of CMB and grown in an anaerobic cabinet (Whitley A35 Workstation, Don Whitley Scientific). Growth curves were collected in 96-well plates using Cerillo Stratus plate readers. Then, 300 µl of 2× CMB (minus the monosaccharides) was mixed with 300 µl of substrate, and 200 µl was pipetted into 3 wells to provide replicates. Growth curves were also repeated with different starter cultures and on different days.

### RNA-seq of AM

AM was grown overnight in CMB at 37 °C under anaerobic conditions. The culture was diluted 1:20 into CMB minus the monosaccharides with the addition of either glucose or mucin and grown to an optical density (OD) of 0.5 at 600 nm. Cells were pelleted by centrifugation at $16,000 \times g$ for 30 s and flash-frozen using a dry ice and ethanol bath. Sample processing, library construction and sequencing were performed by Azenta. Sequencing adapter removal and read quality trimming were performed with fastp v.0.23.2 using default parameters[47]. Reads were mapped to the AM (GCF_000020225.1) and enumerated using kallisto v.0.46.2 (ref. 48). Differential gene expression analyses were performed with Voom/limma v.3.40.6 (ref. 49) implemented in Degust v.4.2-dev (https://zenodo.org/records/3501067). The volcano plot was created using the package EnhancedVolcano v.1.22.0 (https://github.com/kevinblighe/EnhancedVolcano) in R v.4.4.0.

### Whole-cell assays

Overnight starter cultures were inoculated from a glycerol stock into 5 ml of CMB and grown under anaerobic conditions. This was then used to inoculate a 5 ml culture of AM in CMB (minus the monosaccharides) with substrate; the cells were pooled, collected at the mid-exponential growth phase and washed twice with PBS; and 250 µl of 2× PBS was used per 5 ml culture to resuspend the cells. The assays were then mixed at a 50:50 cells-to-substrate ratio, in which substrate concentrations were the same as used in the recombinant enzyme assays. PGMIII, BSM, hyaluronic acid, chondroitin sulfate, RNaseB and *Saccharomyces cerevisiae* (Sc) mannan assays were 500 µl in total, and 50 µl samples were taken for each time point. Assays with defined oligosaccharides and HMOs were 200 µl, and 20 µl samples were taken for each time point. Assays were carried out at 37 °C and reactions were terminated by boiling samples.

### HPAEC-PAD

To analyse monosaccharide release, sugars were separated using a CarboPac PA-1 anion exchange column with a PA-1 guard using a Dionex ICS-6000 (Thermo Fisher) and detected using PAD. Flow was 1 ml min$^{-1}$ and elution conditions were 0–25 min 5 mM NaOH and then 25–40 min 5–100 mM NaOH. The software was the Chromeleon Chromatography Data System. Monosaccharide standards were 0.1 mM; all data were obtained by diluting assays 1/10 before injection, apart from the ceramide samples, which were diluted 15 µl in 100 µl.

### Steady-state kinetics of AmPL38

Initial velocities of purified recombinant Amuc_0778[PL38] were quantified on hyaluronic acid (HA), chondroitin sulfate A (CSA), chondroitin sulfate C (CSC) and dermatan sulfate (DS) substrates, with concentrations ranging from 0.25 to 12 g l$^{-1}$ at 37 °C, 100 mM NaCl and UB4 buffer at pH 7 (ref. 50). The average initial velocities, quantified in milli-absorbance units at A235nm per second, were converted to µM of product generated by measuring the amount of Δ4,5 bonds formed per second, using the experimentally confirmed extinction coefficient for unsaturated glucuronic acid of 6,150 M$^{-1}$ cm$^{-1}$ (ref. 45). Kinetic parameters were determined by plotting initial velocities against substrate concentrations and fitting the Michaelis–Menten model using GraphPad Prism.

### Initial rates of AmGH105

The steady-state reactions of Amuc_0778[PL38] were sealed to prevent evaporation and incubated overnight at 37 °C. Subsequently, Amuc_0863[GH105] was added to a final concentration of 1 µM, and the decrease in A235nm was monitored for a minimum of 2 h at 37 °C. The substrate concentration was calculated by converting the initial absorbance of the Amuc_0778[PL38] steady-state reactions, minus absorbance backgrounds, to µM double bonds. The initial rates of Amuc_0863[GH105] were then calculated as the loss of double bonds in µM per minute in absolute values.

### LC−MS of GAG substrates

Duplicate time-course reactions for Amuc_0778[PL38] were prepared under the same conditions as for the kinetics at 2 g l$^{-1}$ substrate concentrations. Reactions were terminated by heating the samples at 95 °C for 5 min. Amuc_0863[GH105] was added to the 20 h reaction to a final concentration of 1 µM and left to run for 2 h before heat inactivation. The final sample preparation and liquid chromatography–mass spectrometry (LC−MS) analysis were carried out using an ion trap coupled with GlycanPac chromatography as previously described[51]. Compounds were observed as single- or double-charge m/z, primarily as deprotonated adducts. The compounds were identified by MS and MS$^2$ fragmentation, if possible. Extracted ion chromatograms of identified compounds were prepared, and the areas of the peaks were used to quantify the products (Supplementary Fig. 36). Fragments follow the nomenclature of ref. 52.

### Glycan labelling

Released O-glycans were fluorescently labelled by reductive amination with procainamide as described previously using the LudgerTagTM Procainamide Glycan Labelling Kit (LT-KPROC-96)[53]. Briefly, samples in 10 µl of pure water were incubated for 60 min at 65 °C with procainamide labelling solution. Residual chemicals were removed from the procainamide-labelled samples using LudgerClean S-cartridges (LC-S-A6). The purified procainamide-labelled O-glycans were eluted with pure water (1 ml). The samples were dried by vacuum centrifugation and resuspended in pure water (50 µl) for further analysis.

### LC−FLD−ESI−MS

Procainamide-labelled samples were analysed by LC−FLD−ESI−MS using an ACQUITY UPLC BEH-Glycan 1.7 µm 2.1 × 150 mm column at 40 °C on a Thermo Scientific UltiMate 3000 UPLC instrument with a fluorescence detector (λex = 310 nm, λem = 370). MS: Gradient conditions were 0–10 min, 15% A at a flow rate of 0.4 ml min$^{-1}$; 10–95 min, 15–43% A at a flow rate of 0.4 ml min$^{-1}$; 95–98 min, 43–90% A at 0.4–0.2 ml min$^{-1}$; 98–99 min, 90% A at 0.2 ml min$^{-1}$; 99–100 min, 90–15% A at 0.2 ml min$^{-1}$; 100–103 min, 15% A at 0.2 ml min$^{-1}$; and 103–115 min, 15% A at 0.2–0.4 ml min$^{-1}$. Solvent A was 50 mM ammonium formate, pH 4.4, made from Ludger Stock Buffer (LS-N-BUFFX40); solvent B was acetonitrile (Acetonitrile 190 far UV/gradient quality; Romil number H049). Samples were injected in 15% aqueous solution and 85% acetonitrile, with an injection volume of 20 µl. The UPLC system was coupled on-line to an AmaZon Speed ETD electrospray mass spectrometer (Bruker Daltonics) with the following settings: source temperature, 180 °C; gas flow, 10 l min$^{-1}$; capillary voltage, 4,500 V; ion-charge control (ICC) target, 200,000; maximum accumulation time, 50 ms; rolling average, 2; number of precursor ions selected, 3, release after 0.2 min; positive-ion mode; scan mode, enhanced resolution; and mass range scanned, 400–1,700. A glucose homopolymer ladder labelled with procainamide (Ludger; CPROC-GHP-30) was used as a system suitability standard as well as an external calibration standard for glucose units (GU) allocation. ESI−MS and MS−MS data were analysed using Bruker Compass DataAnalysis V4.1 software and GlycoWorkbench software[54].

### Bioinformatics

Putative signal sequences were identified using SignalP 6.0 (ref. 55). Identities between different sequences were determined using Clustal Omega using full sequences[56]. The CAZy database (http://www.cazy.org) was used as the main reference for CAZymes[57]. Phylogenetic trees were completed in SeaView[58] and final trees were produced in the Interactive Tree of Life[59].

**Pangenome analysis.** To analyse the CAZymes across different species and strains, CAZyme sequences were downloaded from UNIPROT using the get.seq function from the Bio3d package in RStudio[60]. Sequence selection was based on the accession numbers of the CAZymes from the most up-to-date version of the CAZy repository at the time. Sequence txt files were converted to a FASTA format using the TabulartoFasta function from the following GitHub repository: https://github.com/lrjoshi/FastaTabular. All multiple sequence alignment and phylogenetic analysis was performed using Clustal Omega[56]. Percentage identity matrices were downloaded directly from the programme and merged with the original file from CAZy according to shared accession numbers. The presence or absence of different enzymes and how well conserved they were across strains and species were curated by hand.

**Structural comparisons.** Where possible, protein models were retrieved from the AlphaFold Protein Structure Database[61,62]. Where the models had not been built, this was completed using AlphaFold2 Colab[63]. Previously published protein structures were retrieved from the Research Collaboratory for Structural Bioinformatics (RCSB) Protein Data Bank. Different structures or models were superimposed using COOT[64], and PyMOL was used to look at structures and models and generate figures. Glycan models were built using the Carbohydrate Builder on http://glycam.org. ChemDraw 18.1 was used to draw carbohydrate structures.

### Reporting summary

Further information on research design is available in the Nature Portfolio Reporting Summary linked to this article.

## Data availability

The full RNA-seq data are provided in Supplementary Data 1 and submitted to https://www.ebi.ac.uk/ena/browser/home with accession number PRJEB76658. Source data are provided with this paper.

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

## Acknowledgements

Many thanks to W. Willat's group (Newcastle University, UK) for providing some of the substrates for us to test. Thank you to C. Morland (Newcastle University, UK) for technical support. Many thanks to A. S. Meyer and W. van Schaik for their insightful comments about the paper. We would also like to thank S. Janecek from the Slovak Academy of Sciences for sharing his insights regarding GH57 enzymes. The work was funded by The Academy of Medical Sciences (SBF0061175), the Wellcome Trust and Royal Society (224240/Z/21/Z) awarded to L.I.C. and a BBSRC/Innovate UK IB catalyst award to D.N.B. 'Glycoenzymes for Bioindustries' (BB/M029018/1) and T.O.O. and S.Ç. are funded by the BBSRC Midlands Integrative Biosciences Training Partnership (MIBTP) with his studentship in collaboration with industrial partners Ludger (Oxford, UK) awarded to L.I.C. B.P and J.H. are supported by the Department of Biotechnology and Biomedicine, Section for Protein Chemistry and Technology, Technical University of Denmark.

## Author contributions

C.R.B. and L.I.C. designed the study. Enzymology was carried out by C.R.B., T.O.O., B.P., J.H., M.Z., S.Ç., M.K., E.N., D.N.B. and L.I.C. Growth assays were performed by L.I.C. RNA-seq was performed by C.R.B. and the data processed by R.M. Whole-cell assays and HPAEC were carried out by L.I.C. Enzyme kinetics and LC–MS of GAG-active enzymes were done by B.P. and J.H. Glycan labelling was performed by C.R.B. and LC–FLR–ESI–MS by R.P.K., and the data were processed by L.I.C. Bioinformatics was carried out by L.I.C. The draft of this paper was written by C.R.B. and L.I.C. All authors contributed in editing the paper.

## Competing interests

The authors declare no competing interests.

## Additional information

**Extended data** is available for this paper at https://doi.org/10.1038/s41564-024-01911-7.

**Correspondence and requests for materials** should be addressed to Lucy I. Crouch.

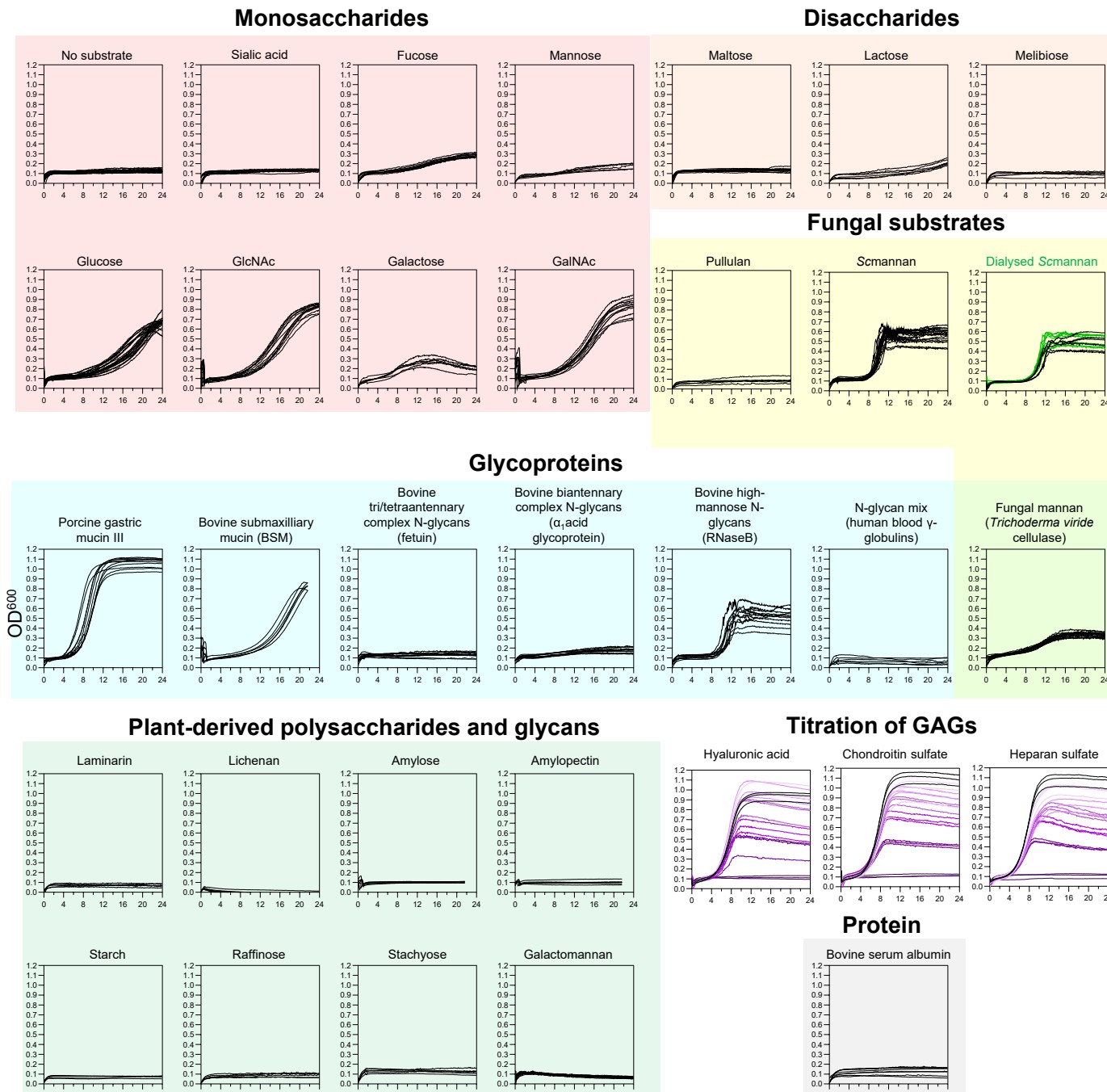

**Extended Data Fig. 1 | See next page for caption.**

**Extended Data Fig. 1 | Growth of *A. muciniphila* ATCC BAA-835 on a variety of glycans and polysaccharides.** *A. muciniphila* was grown anaerobically in minimal media containing a different potential nutrient sources. Growth was monitored continuously at OD600 using a 96-well plate in a plate reader. Concentrations of substrates are listed in Supplementary Table 2. For the dialysed Scmannan, the black curves are original sample and the green is dialysed. For the GAG substrates, this was completed with varying concentrations of mucin:GAG. The black lines are the mucin growth curves and the darkest purple are the GAGs alone. The lighter the purple gets the more mucin in those growths. Each line is one bacterial culture.

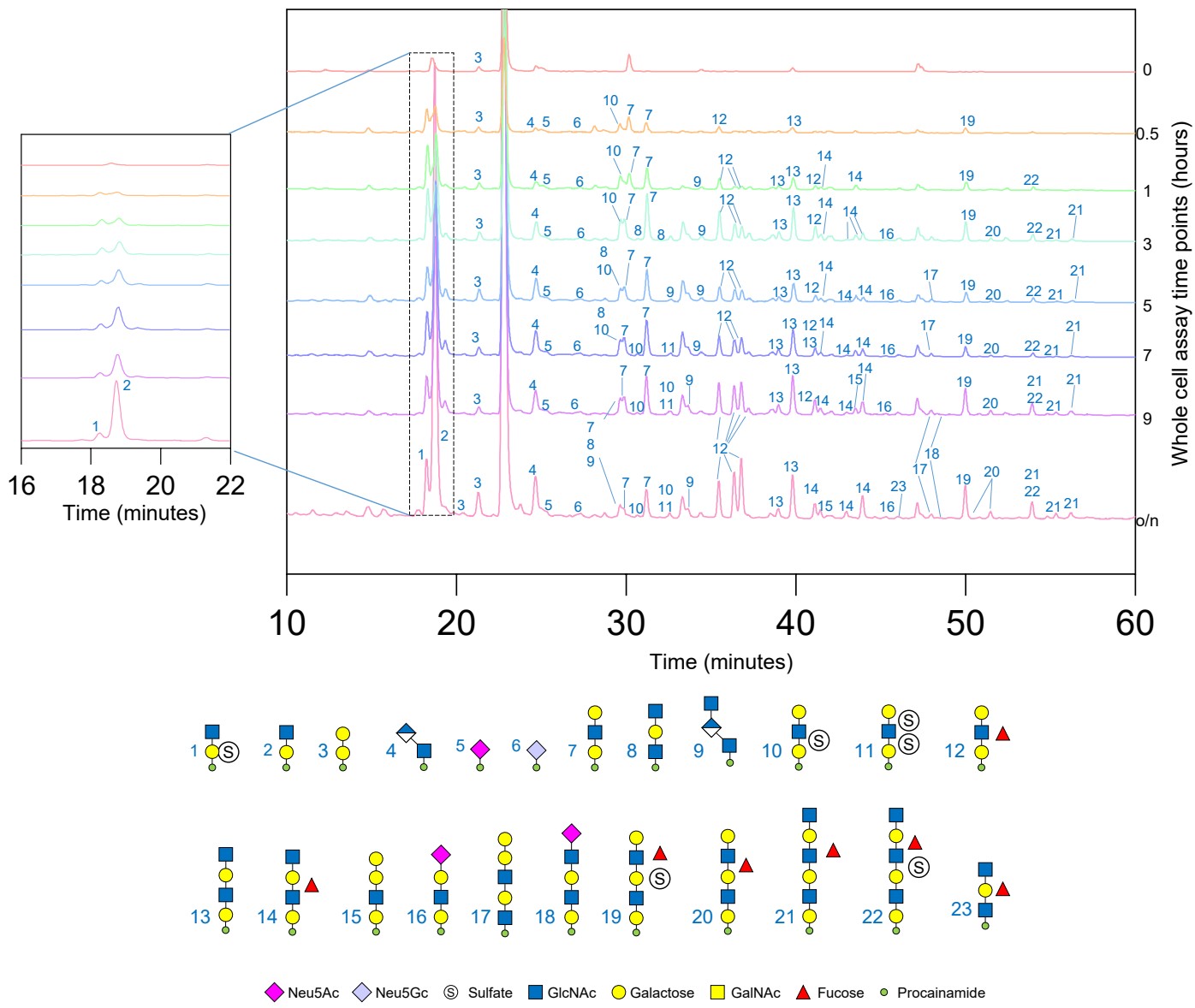

**Extended Data Fig. 2 | Characterisation of surface enzymes activity of AM using whole cell assays.** The glycan products from the different whole cell assay timepoints were labelled with procainamide at the reducing end and analysed by LC-FLD-ESI-MS. The results show a release of a variety of different O-glycan fragments, predominantly with galactose at the reducing end, which is indicative of GH16 endo-O-glycanase activity. The panel on the left emphasizes glycans 3 and 4 and the panel on the right emphasizes glycans eluting between 25-60 minutes.

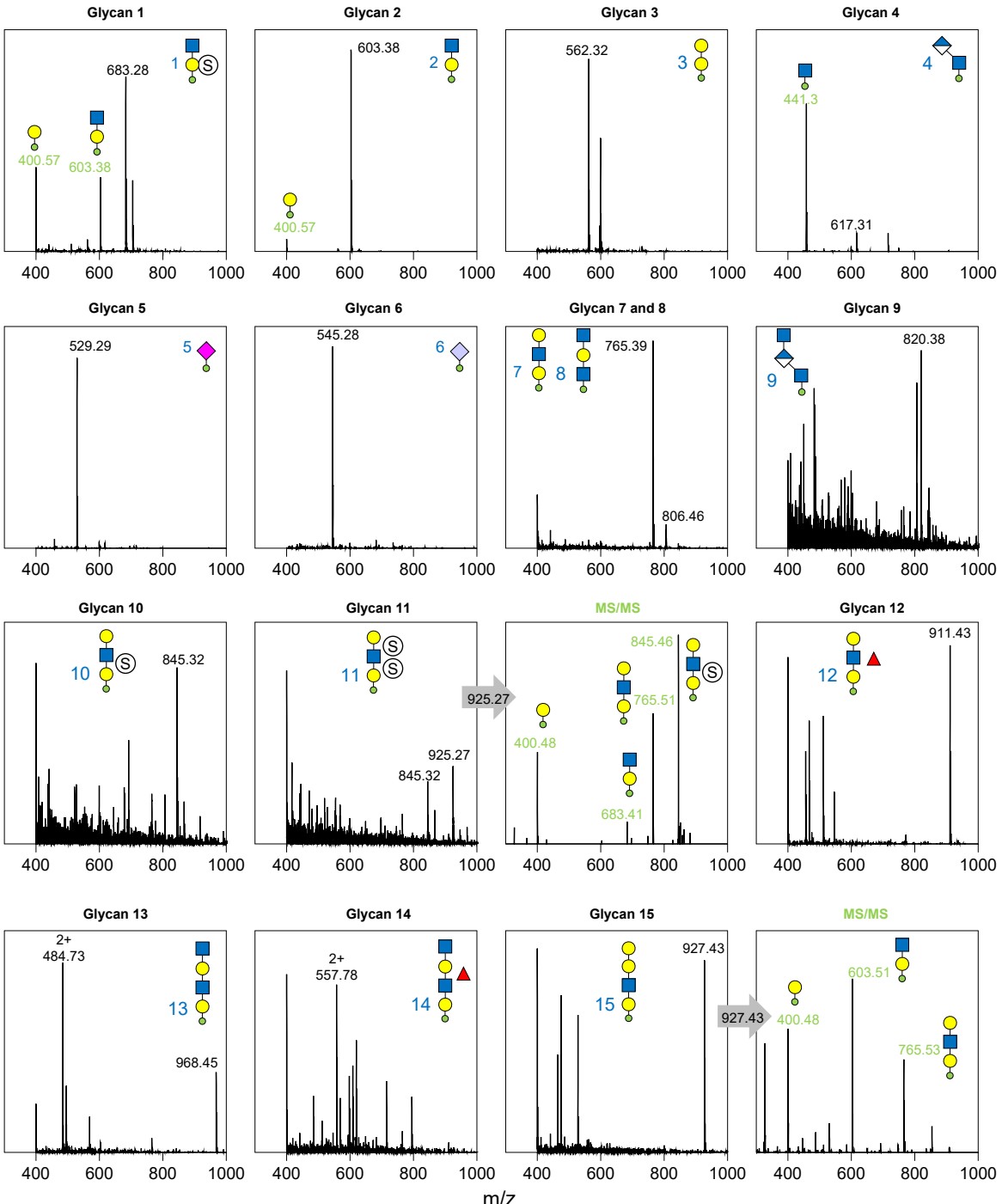

**Extended Data Fig. 3 | Characterisation of surface enzymes activity of AM using whole cell assays.** Examples of mass spectra for each glycan from the whole cell assay. The glycan products from the different whole cell assay timepoints were labelled with procainamide at the reducing end and analysed by LC-FLD-ESI-MS. The numbering corresponds to the numbering in the numbering in Fig. 1. Black and green are Y and B fragments, respectively. MS/MS data is shown for some glycans to provide examples of how structures were determined. The grey arrows clarify which peak the MS/MS data belongs to.

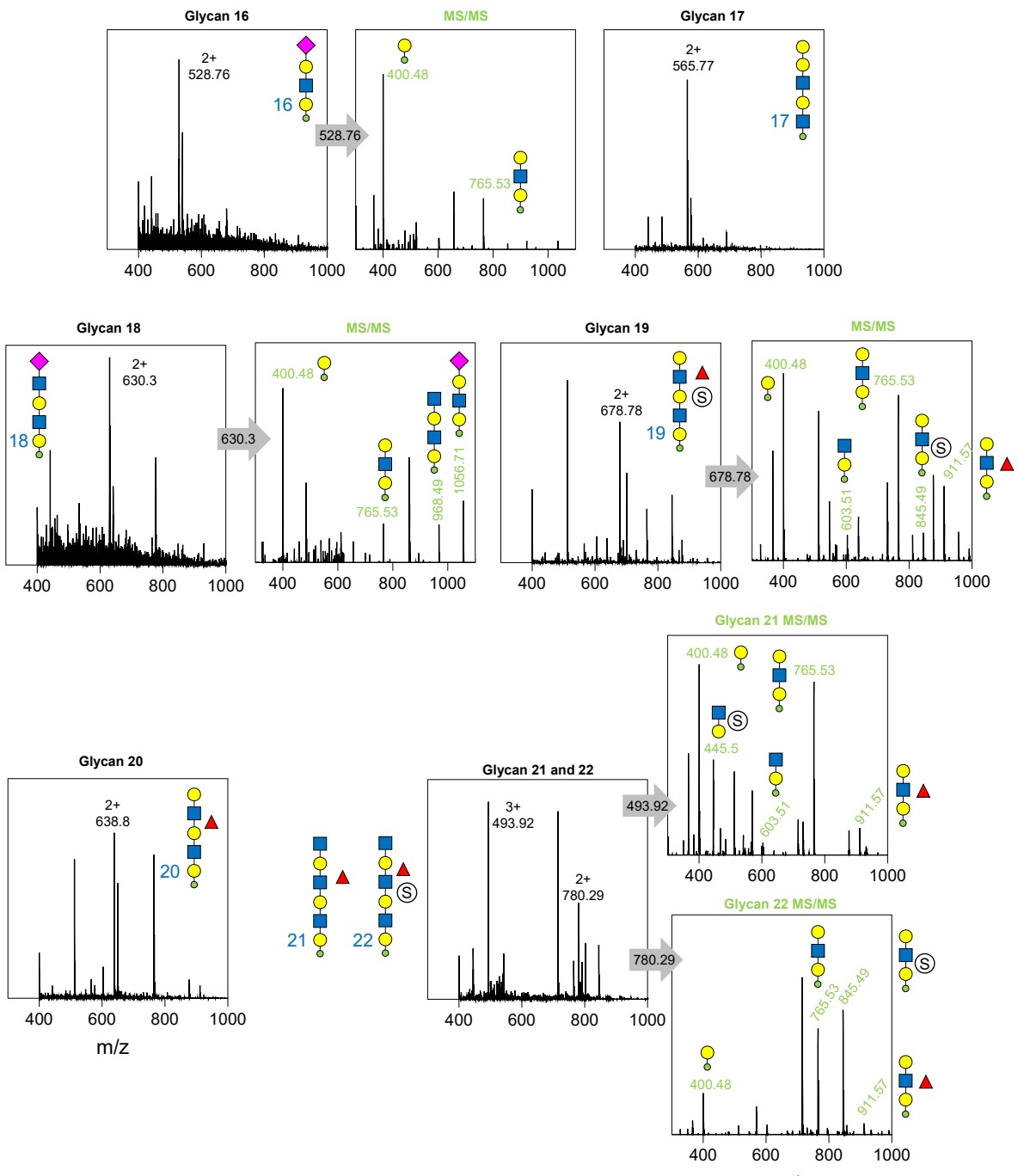

**Extended Data Fig. 4 | Characterisation of surface enzymes activity of AM using whole cell assays continued.** Examples of mass spectra for each glycan from the whole cell assay. The glycan products from the different whole cell assay timepoints were labelled with procainamide at the reducing end and analysed by LC-FLD-ESI-MS. The numbering corresponds to the numbering in the numbering in Fig. 1. Black and green are Y and B fragments, respectively. MS/MS data is shown for some glycans to provide examples of how structures were determined. The grey arrows clarify which peak the MS/MS data belongs to.

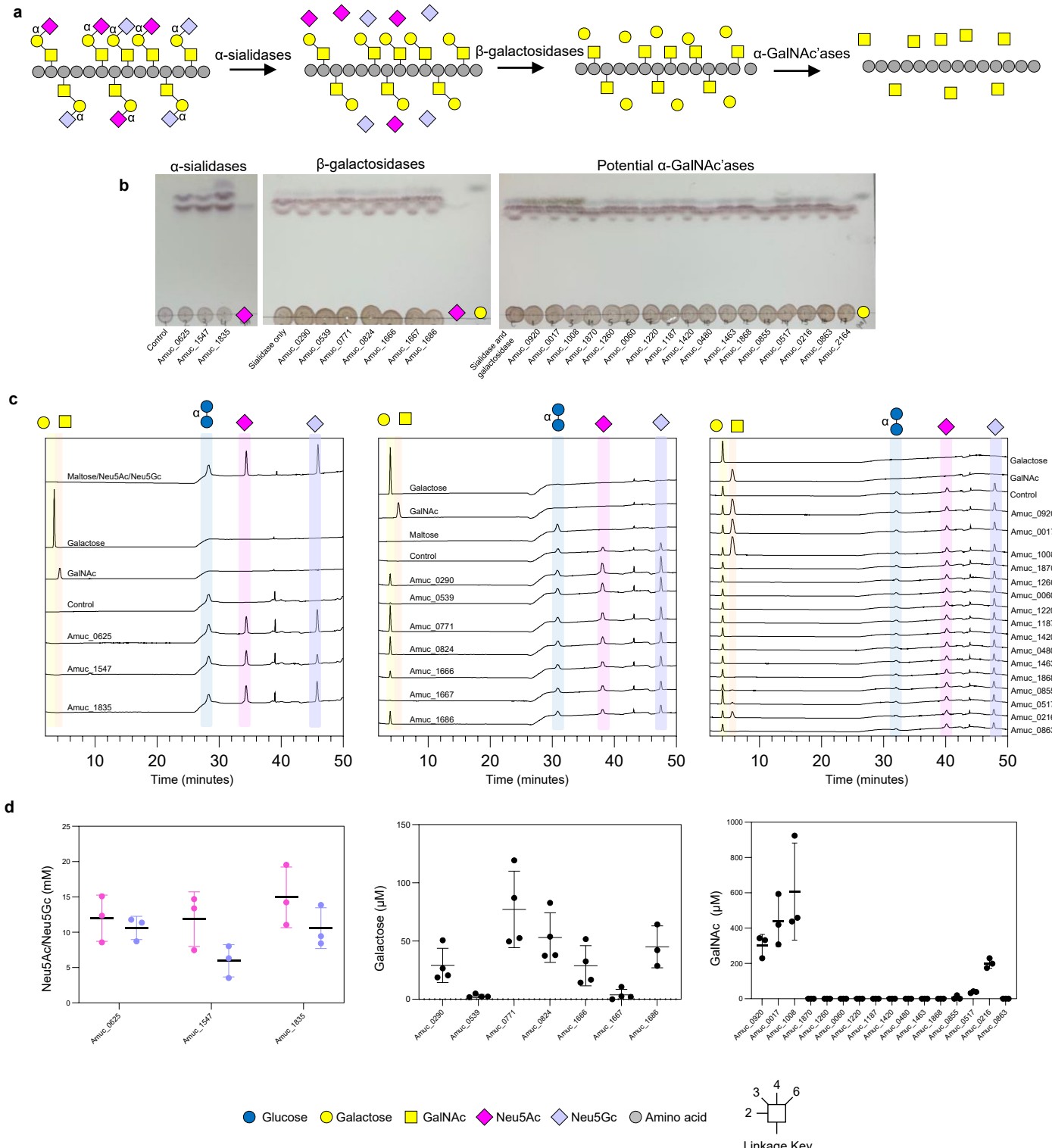

**Extended Data Fig. 5 | Sequential degradation of O-glycans on BSM with CAZymes from *A. muciniphila* BAA-835.** The sialidases, β-galactosidases, and CAZymes associated with α-linked monosaccharide hydrolysis were added sequentially to understand specificity of the CAZymes. **a**, Structural features that are expected in bovine submaxilliary mucin (left) and how it is broken down in this experiment. Only alpha linkages are labelled apart from the core GalNAc monosaccharides which are also alpha linkages. **b**, Thin layer chromatography results of the sequential degradation. Standards have also been included on the left of the TLCs. **c**, The chromatograms of the assays analysed by HPAEC-PAD. Maltose was used as the internal standard and other standards were run separately. **d**, The areas of the peaks were quantified and are presented on a scatter dot plot with the mean and SD. The data are from at least three distinct reactions.

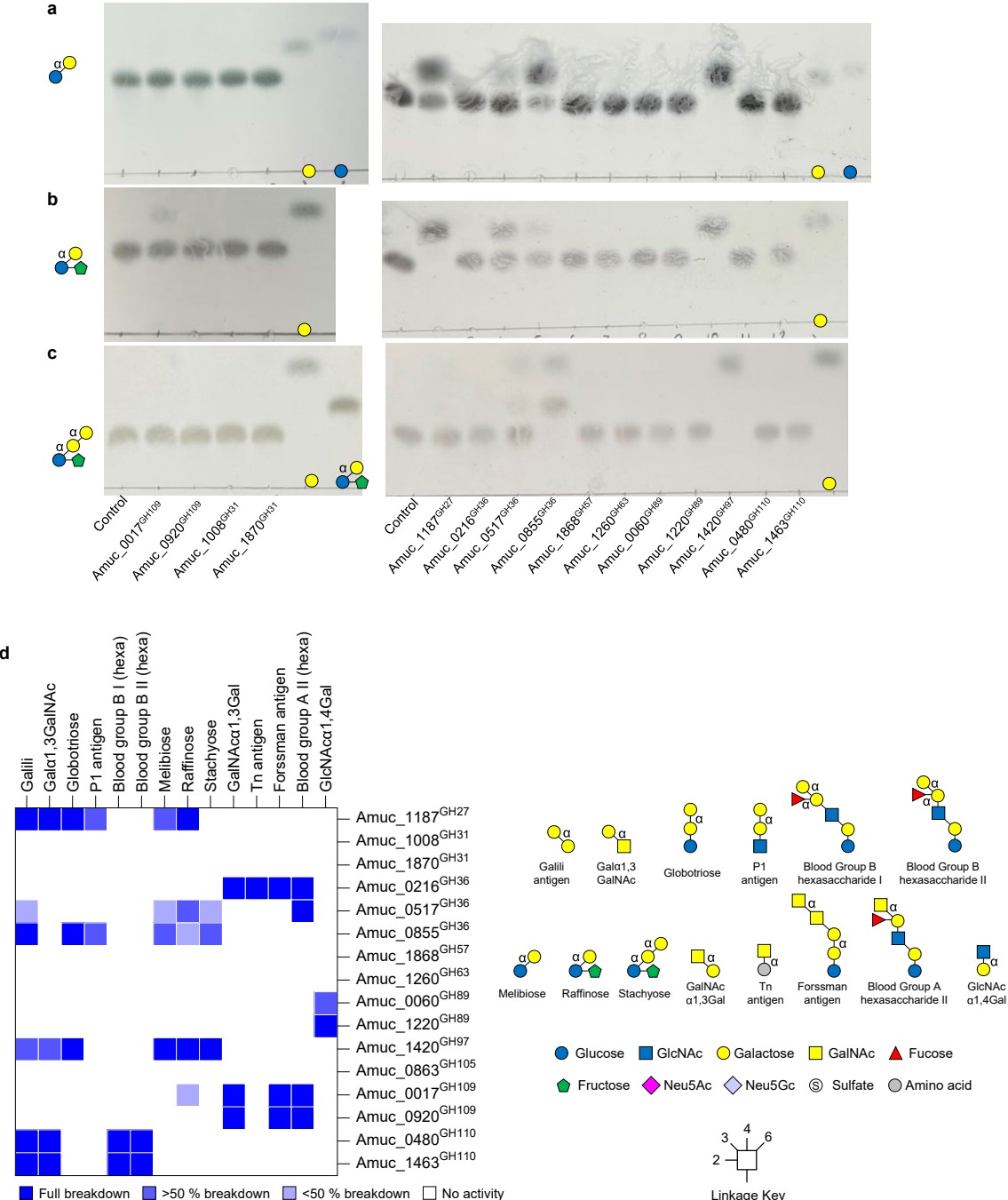

**Extended Data Fig. 6 | Activity of GH enzymes from *A. muciniphila* BAA-835 against raffinose-type defined oligosaccharides and a summary of activities against substrates with α-capping sugars. a**, melibiose. **b**, raffinose. **c**, stachyose. Standards have also been included on the TLCs on the right. Enzyme assays were carried out at a final substrate concentration of 1 mM, pH 7, 37 °C, overnight, and with 1 μM enzyme. **d**, Heat map of recombinant enzyme activities against defined oligosaccharides. The dark blue and white indicate full and no activity, respectively, and partial activities are represented by the lighter blues.

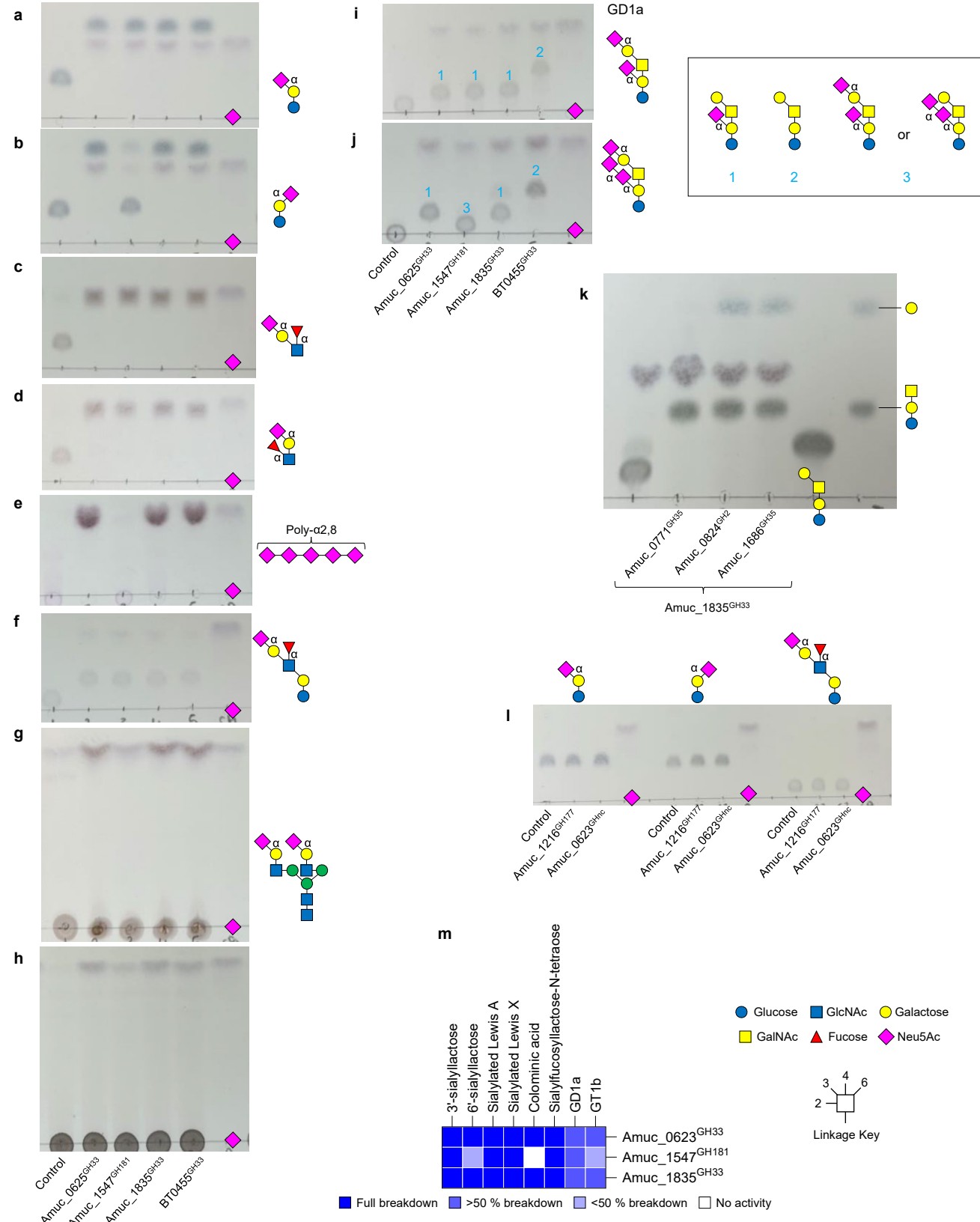

**Extended Data Fig. 7 | See next page for caption.**

**Extended Data Fig. 7 | Activity of the GH33 and GH181 family members from**
***A. muciniphila* BAA-835 against defined oligosaccharides, polysaccharides,**
**and mucin substrates. a**, 3'-sialyllactose. **b**, 6'-sialyllactose. **c**, Sialylated Lewis
A. **d**, Sialylated Lewis X. **e**, colominic acid. **f**, sialylfucosyllacto-N-tetraose. **g**,
biantennary complex N-glycans ($\alpha_1$ acid glycoprotein). **h**, PGM III. **i**, GD1a **j**, and
GT1b. **k**, Exploring if removing the galactose also allows the release of the final
sialic acid. **l**, Testing Amuc_1216$^{GH177}$ and Amuc_0623$^{GHnc}$ against three different
sialylated substrates. **m**, Heat map of recombinant enzyme activities against

defined oligosaccharides. The dark blue and white indicate full and no activity,
respectively, and partial activities are represented by the lighter blues. Standards
have also been included on the TLCs and substrates have been included on the
right where possible. For the gangliosides (I and j) the structures of the different
products are in the key to the right. Enzyme assays were carried out at a final
substrate concentration of 1 mM (except N-glycans at 10 mg/ml and colominic
acid at 25 mg/ml), pH 7, 37 °C, overnight, and with 1 µM enzymes. BT0455 from
*Bacteroides thetaiotaomicron* has been used as a positive control a-i.

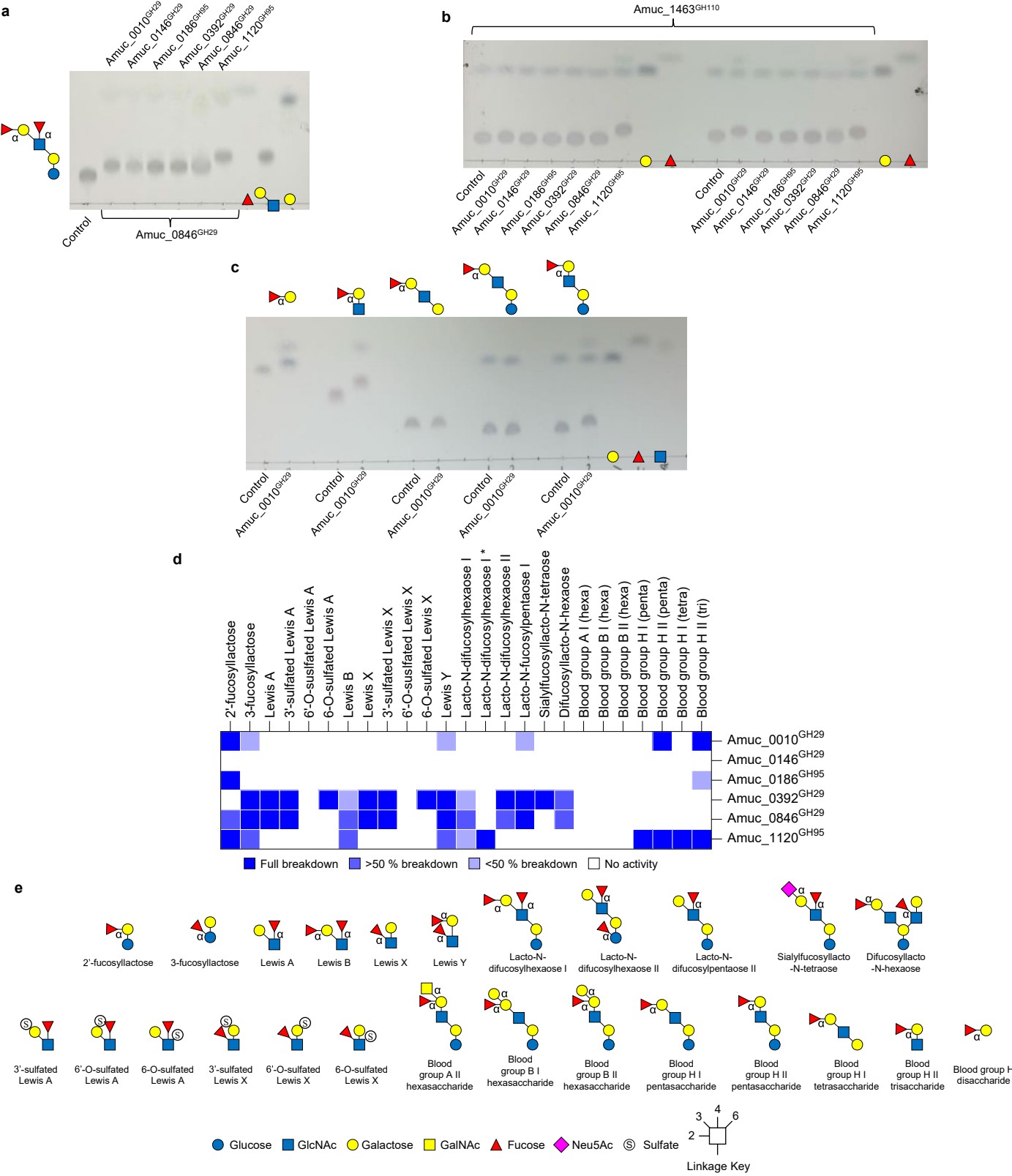

**Extended Data Fig. 8 | Activity of the GH29 and GH95 family members from *A. muciniphila* BAA-835 against defined oligosaccharides, Part 2. a**, Exploring the full degradation of LNDFH I by pre-treating with Amuc_0846$^{GH29}$ and then adding the other fucosidases. **b**, Fucosidase activity against different types of blood group B structures where the α-linked galactose has been removed. **c**, Amuc_0010$^{GH29}$ activity against a range of blood group H structures. For all assays, standards have been included on the TLCs. Enzyme assays were carried out at a final substrate concentration of 1 mM, pH 7, 37 °C, overnight, and with 1 μM enzyme. **d**, A. heat map of recombinant enzyme activities against defined oligosaccharides. The dark blue and white indicate full and no activity, respectively, and partial activities are represented by the lighter blues. The asterisk indicates that one fucose has been remove by Amuc_0846$^{GH29}$. **e**, Structures of the defined substrates used to characterize the fucosidases.

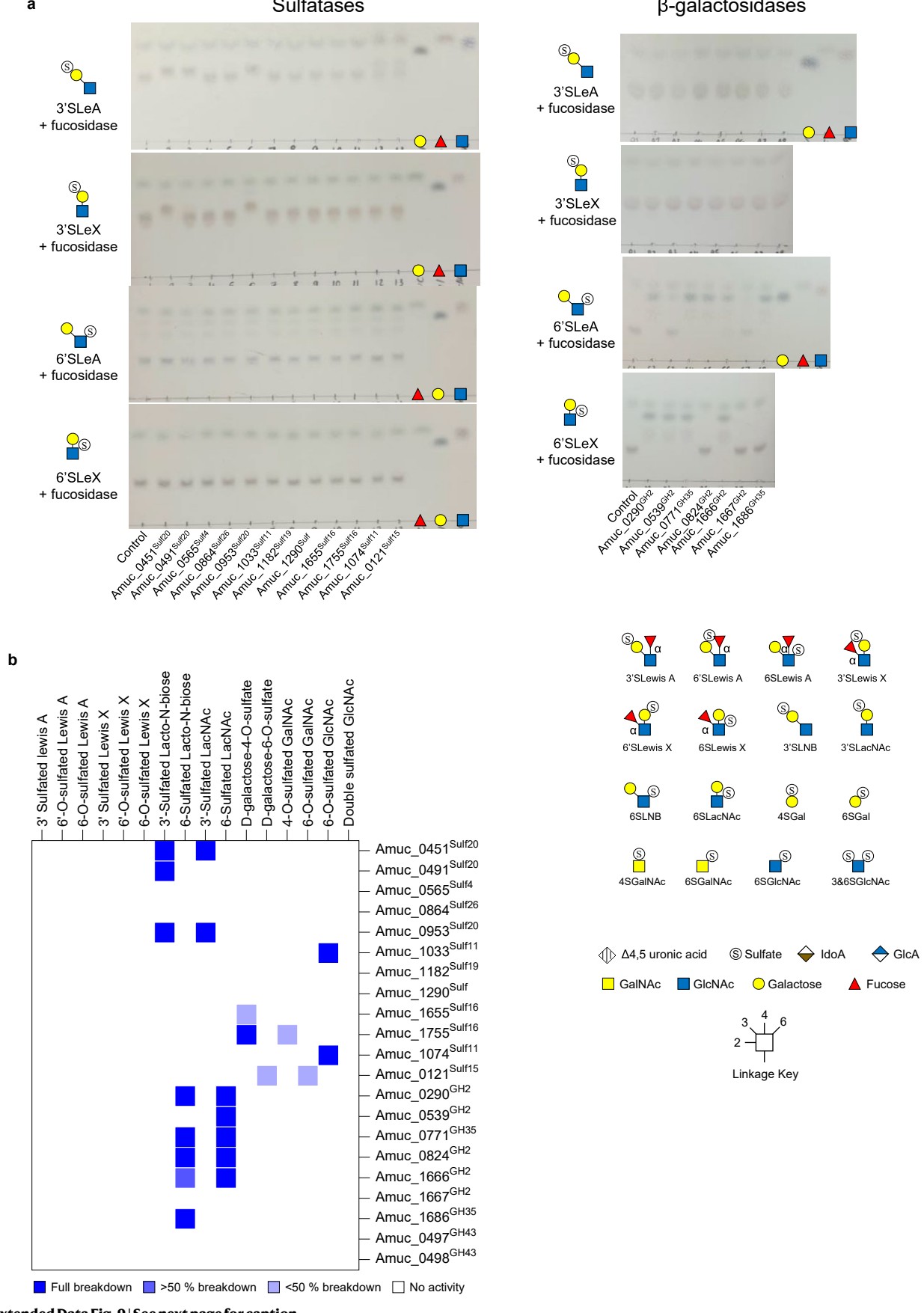

**Extended Data Fig. 9 | See next page for caption.**

**Extended Data Fig. 9 | Activity of fucosidases and β-galactosidases from *A. muciniphila* BAA-835 against sulfated Lewis Structures. a**, Standards have also been included on the TLCs. Enzyme assays were carried out at a final substrate concentration of 1 mM, pH 7, 37 °C, overnight, and with 1 μM enzyme. **b**, Heat map of recombinant enzyme activities against defined oligosaccharides. The dark blue and white indicate full and no activity, respectively, and partial activities are represented by the lighter blues.

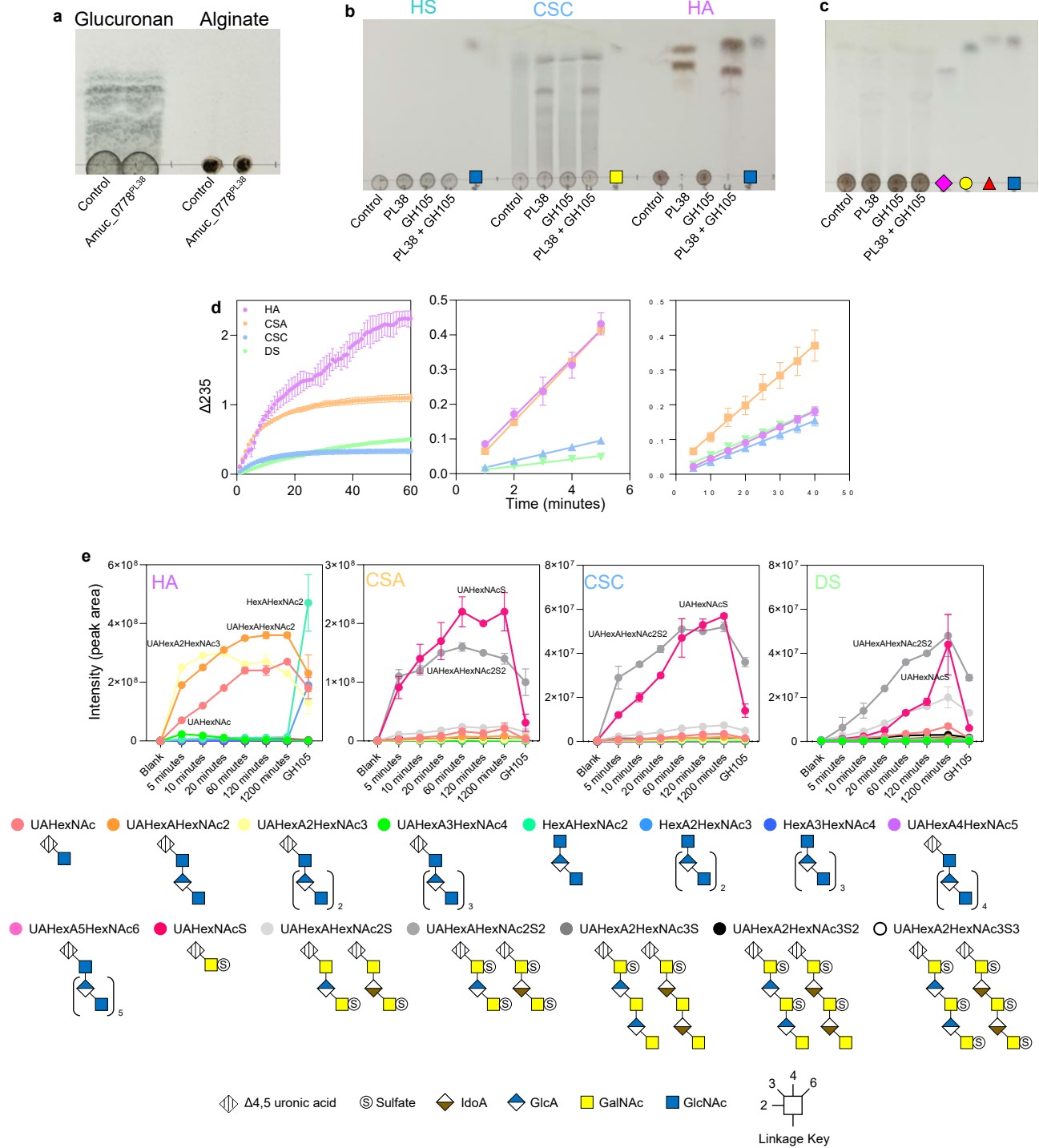

**Extended Data Fig. 10 | Activity of the GAG-active enzymes and breakdown of sulfated host glycans by AM. a**, Activity of Amuc_0778[PL38] against known PL38 substrates glucuronan (left) and alginate (right). Enzyme assays were carried out at pH 7, 37 °C, overnight, and with 1 μM enzyme. **b**, Activities of the Amuc_0771[PL38] and Amuc_0863[GH105]. **c**, Testing Amuc_0771[PL38] and Amuc_0863[GH105] against PGMIII. **d**, Left: substrate depletion of different GAGs (2 g/L) by Amuc_0771[PL38]. This wavelength allows the monitoring of the generation of unsaturated products (double bonds) of the PL38. Middle: Highest initial rates of Amuc_0771[PL38]. Right: Initial rates of the Amuc_0863[GH105]. The change in absorbance monitors the GH105 activity against the Amuc_0771[PL38] product. **e**, Quantification of the Amuc_0771[PL38] activity over time from four different GAG substrates by LC-MS. The area under the peaks from the chromatograms are shown and each glycan has been given a different colour. For sulfated glycans, the position of the sulfation is unknown. CSA and DS typically has 4S on GalNAc, whereas CSC has relatively high levels of 6S on GalNAc, but also low levels of 4S. The GlcA in CSC can also have 2S sulfation. HA has no sulfation. Data are presented as mean values +/− standard deviation and experiments were completed in triplicate.

# Reporting Summary

## Statistics

For all statistical analyses, confirm that the following items are present in the figure legend, table legend, main text, or Methods section.

| n/a | Confirmed | |
|---|---|---|
| ☒ | ☐ | The exact sample size (*n*) for each experimental group/condition, given as a discrete number and unit of measurement |
| ☒ | ☐ | A statement on whether measurements were taken from distinct samples or whether the same sample was measured repeatedly |
| ☒ | ☐ | The statistical test(s) used AND whether they are one- or two-sided<br>*Only common tests should be described solely by name; describe more complex techniques in the Methods section.* |
| ☒ | ☐ | A description of all covariates tested |
| ☒ | ☐ | A description of any assumptions or corrections, such as tests of normality and adjustment for multiple comparisons |
| ☐ | ☒ | A full description of the statistical parameters including central tendency (e.g. means) or other basic estimates (e.g. regression coefficient) AND variation (e.g. standard deviation) or associated estimates of uncertainty (e.g. confidence intervals) |
| ☒ | ☐ | For null hypothesis testing, the test statistic (e.g. *F*, *t*, *r*) with confidence intervals, effect sizes, degrees of freedom and *P* value noted<br>*Give P values as exact values whenever suitable.* |
| ☒ | ☐ | For Bayesian analysis, information on the choice of priors and Markov chain Monte Carlo settings |
| ☒ | ☐ | For hierarchical and complex designs, identification of the appropriate level for tests and full reporting of outcomes |
| ☒ | ☐ | Estimates of effect sizes (e.g. Cohen's *d*, Pearson's *r*), indicating how they were calculated |

*Our web collection on statistics for biologists contains articles on many of the points above.*

## Software and code

Policy information about availability of computer code

| | |
|---|---|
| Data collection | Chromeleon Chromatography Data System Software v7.3, AlphFold2 Collab, SignalP 6.0, UNIPROT, Bio3d from RStudio, Tabulartofasta from GitHub, Bruker Compass HyStar 5.1.8.1 to collect GAG data, Procainamide-labelling data: Thermo Scientific UltiMate 3000 UPLC instrument with a fluorescence detector controlled by HyStar software version 3.2. MS data collection: AmaZon Speed ETD electrospray mass spectrometer (Bruker Daltonics, Bremen, Germany). |
| Data analysis | Graphpad Prism to produce graphs and heatmaps, ClustalOmega to generate sequence alignments, Pymol to produce structural images, Coot to overlay structures, GLYCAM carbohydrate builder, SeaView to generate phylogeny trees, iTOL to present phylogeny trees, Bruker Compass Data Analysis 6.1 to analyse GAG data, TASQ 2.2 for relative compound quantification of GAGs, ChemDraw 18.1 was used to draw carbohydrate structures, RNA-seq data processing: fastp v.0.23.2, kallisto v.0.46.2, Voom/limma v.3.40.6, Degust v.4.2-dev, Volcano plot produced using R v.4.4.0 and EnhancedVolcano v.1.22.0. Procainamide-labelling data analysis: ESI-MS and MS/MS data analysis was performed using Bruker Compass DataAnalysis V4.1 software. |

For manuscripts utilizing custom algorithms or software that are central to the research but not yet described in published literature, software must be made available to editors and reviewers. We strongly encourage code deposition in a community repository (e.g. GitHub). See the Nature Portfolio guidelines for submitting code & software for further information.

## Data

Policy information about availability of data

All manuscripts must include a data availability statement. This statement should provide the following information, where applicable:
- Accession codes, unique identifiers, or web links for publicly available datasets
- A description of any restrictions on data availability
- For clinical datasets or third party data, please ensure that the statement adheres to our policy

> The full RNA-seq data are provided in Supplementary Data 1 and submitted to https://www.ebi.ac.uk/ena/browser/home with accession number PRJEB76658. The data that support the findings presented in this manuscript are available upon request from the corresponding authors.

## Research involving human participants, their data, or biological material

Policy information about studies with human participants or human data. See also policy information about sex, gender (identity/presentation), and sexual orientation and race, ethnicity and racism.

| | |
|---|---|
| Reporting on sex and gender | N/A |
| Reporting on race, ethnicity, or other socially relevant groupings | N/A |
| Population characteristics | N/A |
| Recruitment | N/A |
| Ethics oversight | N/A |

Note that full information on the approval of the study protocol must also be provided in the manuscript.

# Field-specific reporting

Please select the one below that is the best fit for your research. If you are not sure, read the appropriate sections before making your selection.

☒ Life sciences ☐ Behavioural & social sciences ☐ Ecological, evolutionary & environmental sciences

For a reference copy of the document with all sections, see nature.com/documents/nr-reporting-summary-flat.pdf

# Life sciences study design

All studies must disclose on these points even when the disclosure is negative.

| | |
|---|---|
| Sample size | No sample size calculation was performed during this work. Bacterial growth on different glycans/monosaccharides was carried out in triplicate and the experiments repeated at least once (ie. minimum of two biological replicates with triplicate technical replicates each time). Non-kinetic enzyme assays were repeated at least once for each substrate tested with different enzyme preparations. For kinetics against GAGs, the enzymes were produced at least twice and no variability in rates was observed between enzyme preparations. |
| Data exclusions | A few individual growths curves have been removed at the request of a reviewer. |
| Replication | Growth assays were generally repeated at least twice. Exceptions to this were for the dialysed Scmannan experiment and the different ratios of mucin:GAG, which were only carried out once. Assays with defined oligosaccharides have been carried out at least twice. The BSM assays to quantify monosaccharide through HPAEC-PAD was carried out in triplicate. The assays using the alpha enzymes against PGM was carried out once, with and without sialidase and fucosidase. Two separate whole cell assays were carried out. HPAEC-PAD data from the BSM whole cell assay was carried out once. LC-ESI-FLD-MS was carried out once as when we get to the stage of procainamide labelling a sample for characterization, we are sure that the experiment is reproducible and this process is very expensive. Comparison between the different LC-ESI-FLD-MS samples also provides confidence in the results. RNA-seq was carried out in triplicate. |
| Randomization | Randomization was not necessary for any of the experiments carried out during this work. We always have negative controls and, where possible, positive controls |
| Blinding | True blinding was not necessary for any of the experiments carried out during this work. Throughout the production of enzymes, the locus tags were used, but the researchers were not told to expect particular activities. Substrates were often provided letters and the samples (control/enzyme) with numbers to set up and track assays easily. The identity of substrate and enzyme were then reassigned after the data had been collected. Growth curves in 96-well plates were subject to similar coding, which were again reassigned when looking at the data. |

# Reporting for specific materials, systems and methods

We require information from authors about some types of materials, experimental systems and methods used in many studies. Here, indicate whether each material, system or method listed is relevant to your study. If you are not sure if a list item applies to your research, read the appropriate section before selecting a response.

## Materials & experimental systems

| n/a | Involved in the study |
|-----|----------------------|
| ☒ ☐ | Antibodies |
| ☒ ☐ | Eukaryotic cell lines |
| ☒ ☐ | Palaeontology and archaeology |
| ☒ ☐ | Animals and other organisms |
| ☒ ☐ | Clinical data |
| ☒ ☐ | Dual use research of concern |
| ☒ ☐ | Plants |

## Methods

| n/a | Involved in the study |
|-----|----------------------|
| ☒ ☐ | ChIP-seq |
| ☒ ☐ | Flow cytometry |
| ☒ ☐ | MRI-based neuroimaging |

## Plants

| | |
|---|---|
| Seed stocks | N/A |
| Novel plant genotypes | N/A |
| Authentication | N/A |

