## [Peer Review File · Nature Microbiology]

Carbohydrate-active enzymes from *Akkermansia muciniphila* breakdown mucin O-glycans to completion

Corresponding Author: Dr Lucy Crouch

Version 0:

Reviewer comments:

Reviewer #1

(Remarks to the Author)

Bakshani et al. report a lot of initial data that may be valuable to the glycobiology field. The manuscript describes the activities of 55 glycoside hydrolases, 12 sulfatases and a polysaccharide lyase from *Akkermansia muciniphila*. This manuscript provides some insights into the mechanisms that AM utilizes to degrade complex glycans from a variety of sources including porcine gastric mucin, human milk oligosaccharides, and plant polysaccharides. While this work attempts to cover a lot of enzymes, the manuscript does not have a cohesive story and good deal of the results presented are the bare minimum it would take to begin an analysis. In many respects, it appears to be 2-3 papers worth of preliminary data put together. It is hard to imagine that it will make a strong impact on the field because it is a survey of initial conclusions one can make about these enzymes. Furthermore, there are significant concerns about the way key assays were performed. Major and minor comments follow. Taken together, this manuscript cannot be accepted in its current form without major additional experiments and a significant reorganization.

Major:

1. This manuscript is a challenge to read and needs considerable work to make this more approachable for readers. In its current state it reads like a statement of observations and needs significant changes to correct this. The manuscript reads like it was meant to be separated into 2 or more manuscripts. I am not sure the starch degrading enzymes are necessary in this manuscript or the high mannose N-glycoproteins sections.
2. I have concerns with the way the whole cell assays were performed. Several instances throughout the manuscript these whole cell assays do not degrade relatively simple sugars. Do the authors find this concerning? Clarification and commentary are essential.
3. Cell lysate experiments should be conducted against all substrates, not just whole cell assays with substrates. This is especially important since so many whole cell experiments showed no activity.
4. Several figures need reworking to make them easier to view. The TLC images should be scaled similarly so the lanes align in many supplemental figures. Additional annotation of TLC images, like labeling specific lanes, would improve clarity.
5. The pNP assays should be conducted again and absorbance measurements should be taken rather than images. In no way are photos of a color change acceptable as data on enzyme assays. Color changes can be measured precisely and should be, rather than a yes/no report that barely passes the bar as an initial experiment one might show at lab meeting. Additionally, this would allow for rates to be reported, which should be.
6. Why are Michaelis-Menten kinetics performed for Amuc_0778 and not others? This seems odd/out of place. As enzyme scientists know, Michaelis-Menten kinetics are critical for a complete and appropriate understanding of enzyme function.
7. It is not clear that GalNAc is being released in Figure 1. Why is a more thorough experiment not conducted for PGMIII like what was performed for BSM (Suppl Fig 6)? This is one of the most important findings in the study but the control lane for GalNAc is the furthest from the experimental lane. Additionally, in figure 1D the first two lanes are quite messy and not much can be concluded from them.
8. It is mentioned that fucosidases cannot degrade intact mucin oligosaccharides. I suggest the authors conduct additional experiments where PGMIII is incubated with a panel of other enzymes and followed up with these fucosidases, like what is done in Suppl Fig 6. I think that incubating GH16 enzymes prior would be especially useful.
9. Several instances throughout the manuscript the author claims that the experiments conducted suggested one enzyme being more efficient than another and that it was not pursued further. I think that such statements should not be included without data to back up these claims. These repeated statements significantly contribute to the sense that this is an initial report that adds little to the field without the further, essential follow-up work. Furthermore, adequate quantification of enzyme activity using the pNP assays or additional HPAEC-PAD assays would greatly increase the value of Figure 6 for the field, as accurate comparisons could be made between enzymes at each step of mucus degradation.
10. Several growth curves look abnormal and should be performed again. It is also unclear what the multiple lines on each graph represent.
11. Do you think some enzymes that had low activity might have low activity due to a different pH optimum? This may be particularly important for sequences that lack a signal peptide and are not trafficked outside the cell. It is essential that pH screens are performed as it is common knowledge that enzymes have pH optima that change their activities. Without this

information, little can be learned from enzymes with no activity at a single pH.

12. Please clarify how partial activity is being assigned.

13. I think the AF2 models should be excluded unless they are used to explain certain substrate preferences. They do not add much to the overall manuscript. Similarly, the glycan models in Supplemental 4 do not significantly add to the manuscript.

14. The discussion section of the manuscript fails to successfully present the key takeaway points from the results. Further expansion on the importance and relevance of key findings needs to be emphasized. Additionally, the introduction of mucinosomes (line 697) is confusing and does not fit in the context of the presented results.

Minor:

1. Several typos throughout:

a. Oligosaccharides is misspelled in Supplementary Figure 2,3

b. Line 273 – “Tn-antigen was not a degraded”

c. Line 574 – “continud”

d. Line 575 – “degradatioonto”

e. Line 577 – “patter”

f. Line 990 – “(refs)” – missing reference?

g. Figure 4 caption – “...apart from xxx For sulfated...” – not sure what the xxx is referring to

2. Can a representation of colominic acid be added to Supplemental 1E?

3. Line 91 – Disagreement in the proteins mentioned and in supplemental 1 Amuc_0625 is 0623 in supplemental and 1825 is 1835 in supplemental 1.

4. Line 118 – I don't think statements about rates should be included when they were not pursued further. Also, this line references Supplemental 5, however the correct figure to reference is Supplemental 6.

5. Line 123 – What commercially available HMOs were assayed? Were they a mixture of several different HMOs or was just one representative HMO chosen for each group?

6. Line 142-144 – This also indicates that it prefers 1,2 which is not mentioned.

7. Line 185-188 – Where there any differences in active site architecture that would allow for a BGB glycan to fit into the active sites of the two “active” proteins while not fitting into the “inactive” protein? A superimposition could be performed with the model in Supplemental Figure 17.

8. Line 190 – You state “one” protein while there are two proteins in the GH27 family shown in figure 1. It appears that Amuc_1220 was mislabeled and should be in the GH89 family.

9. Line 201 - Was there an increase in galactose when preincubated with sialidases/fucosidases? It appears in Supplementary Figure 11 that we see less of galactose when preincubated with sialidase/fucosidase. Could the authors comment on this?

10. Line 216 – the heat map has two GH97 enzymes but the text says one.

11. Other peaks are in Figure 1C that are not labeled. Can the authors comment on these unlabeled peaks?

12. Line 311 – are there other substrates to test?

13. Line 322-326 – this section needs revising and is not appropriate for a results section.

14. Line 427-428 – the wording here is awkward.

15. Line 429-433 – Consider restructuring here. Perhaps enzyme activities should be described first.

16. Line 484-484 – what is the one feature that you could speculate on? It is odd to make this statement and then not describe what it is.

17. Line 485 – Why are you not able to explore biochemically?

18. Line 508 – I don't think this title is appropriate. I think it is more accurate to state that some of the glycans are completely degraded but these data do not “completely” degrade.

19. Line 534 – what are the predicted functions of the c terminal domain?

20. Line 555 – it is odd to have extensively studied this protein and not others.

21. Line 636 – I don't think this section should be included in this manuscript.

22. Line 710 – this could be because different pHs were not tested.

Reviewer #2

(Remarks to the Author)

Crouch et al. provide a complete picture of colonic mucin glycan degradation by *Akkermansia muciniphilla*, a symbiont associated with gut health. Beyond extensive characterization of the multiple glycoside hydrolases, sulfatases, and lyases involved in mucin breakdown, the authors demonstrate that these enzymes can cooperatively break down complex mucin glycans to completion and provide evidence to suggest the intra/extracellular localization of these events. The authors also expand their characterization of *Akkermansia* enzymatic cleavage activities to other polysaccharides of high relevance to the microbiota, such as gangliosides and human milk oligosaccharides. Taken together, the characterization of these enzymes, and associated data concerning the order of enzyme activity and localization, represent a significant advance for the field. We recommend publication of this work provided that the following concerns regarding presentation of the findings can be addressed.

1. Results: α -sialidases

“Interestingly, there was some indication that Amuc_0625GH33 could release Neu5Gc more efficiently than the other enzymes, but this was not explored further. Release of sialic acid also carried out on porcine gastric mucin III (PGM III) assessed by thin layer chromatography (TLC) indicates that Amuc_0625GH33 and Amuc_1825GH33 are releasing the highest amounts, but this was not quantified (Supplementary Figure 1)”

Taking the first sentence to refer to the chromatography data in Supplementary Figure 6 (box d), it is not apparent that the concentration of Neu5Gc is significantly higher in the Amuc_0625 treated sample compared to the Amuc_1835 sample; as such, it is not clear to us why the authors would make this claim. The subsequent discussion of the relative enzymatic activity as per qualitative TLC staining does not seem significant or supportive to the overall conclusions made by the authors, and we feel this was not presented convincingly in the absence of replicates or more rigorous characterization.

2. Results: Alpha-galactosidases

“Amuc_0480GH110 was able to release more galactose than Amuc_1463GH110, however this was not quantified or the reasons for this difference explored further in this study.”

Like the previous example, this point does not seem significant to the overall conclusions made by the authors, and without more analytical rigor or characterization, the statement is poorly supported.

3. “We could not find any activities for Amuc_0623GHnc or Amuc_1547GH177 during this work despite their associations with sialic acid hydrolysis (Supplementary Figure 1).”

A citation should be provided supporting the notion that Amuc_0623 and Amuc_1547 are associated with sialic acid hydrolysis. Additionally, reference 14 in the manuscript appears to address this aspect of Amuc_0623 activity.

4. We find that the fundamental understanding of Akkermansia’s mucin degradation capabilities afforded by this work represents an important addition to the field given the established prevalence of this microbe in gut health. Another point of significance for the field is the authors’ highly efficient recapitulation of Akkermansia enzymes to act on specific glycan motifs in whole mucin samples, and to furthermore analyze the glycan composition of these complex molecules. These enzyme cocktails could therefore be used to directly study glycan compositions in other contexts, such as gut microbiota dysbiosis and other such “diseased” mucus states.

We would, however like to address this quote from the discussion section: “One of the most significant aspects of this report was showing that we could use a cocktail of CAZymes from AM to reach the core GalNAcs of the O-glycan, thus demonstrating complete degradation of the constituent glycan of this complex substrate (Figure 1).”

We feel this aspect of the work, as significant as it was, was not properly highlighted in the results section. We felt that the wide breadth of enzyme activities characterized, and their relationships to other organisms and substrates other than mucin, received much more written focus. An effort should be made to streamline the writing to highlight the most significant results.

5. Supplemental figure 1: We believe there is a typo with the labeling as “Amuc1835” throughout the figure, where we are led to believe in the corresponding text that “Amuc1825” is the subject of the figure.

6. “An attempt to resolve the kinetic parameters of Amuc_0863GH105 was performed by running the reactions of Amuc_0778PL38 to completion (overnight), then adding Amuc_0863GH105 and following the loss of the double bond of the unsaturated uronic acid at 235nm. While linear decreasing initial rates were observed for the lower substrate concentrations, the higher concentration reactions were not reliable because of the absorbance ceiling of the spectrophotometer. Attempts to downscale or dilute the reactions did not provide data for which a reliable kinetic model could be fitted (Supplementary Figure 40). However, by isolating the highest observed initial rate for each substrate, a clear selectivity of Amuc_0863GH105 was observed towards CSA, followed by the three other GAGs at 36-40% relative activity (Figure 4, Supplementary Table 6).”

That the reaction has reached completion does not follow from it being run overnight, and the authors make no mention of how the substrate concentration was controlled, only that it was measured.

In terms of presentation, this section could be edited to be much clearer and more concise.

7. There are several instances such as the quotes below in the results section where the authors extensively qualify the broader significance of the data to the field or highlight data from other studies. While this is pertinent information that strengthens the impact of the results, we believe that these statements are better placed in the discussion section, to enable better understanding of the results that the authors themselves generated. Alternatively, the writing can be condensed, with these statements made in direct reference to the data.

α -galactosidases

“The whole cell assay results from the blood group hexasaccharides do show breakdown, but the type of glycobiology was not further explored in this work. However, we previously characterised GH16 activity against these substrates and this activity is prominent on the outside of the cell (15; Supplementary Figure 8).”

“Amuc_1187GH27 can hydrolyse Gal α 1,3Gal/GalNAc and globotriose, but less activity was seen against P1 antigen. Gal α 1,3Gal (Galili antigen) is expressed by most mammals and some other animals, but not humans, apes, and Old World monkeys²⁹. Therefore, gut microbes would only have access to this structure through animal products in

the diet. Conversely, the similar Gal α 1,4Gal epitopes of P1 and Pk antigens are generally present in human tissues, so they would be present in the mucosal surface of the human large intestine. P1 and Pk structures are receptors for a variety of pathogens and their toxins³⁰.”

α -GalNAc'ases

“Forssman antigen is normally present in only 0.01 % of the human population but in that relatively small population this antigen would be present in the GI tract³⁴. It can change susceptibility to some diseases and is also expressed in some cancers, but its true prevalence and impact on different diseases is understudied³⁵. BGA is much more prevalent within the human population with this epitope being expressed in 10-40 % of the population depending on where you sample globally³⁶. We were only able to test BGA type II in this study, but given the broad activity of Amuc_0216GH³⁶, it is likely that it will act on all BGA structures.”

Investigating β -galactosidase activity

“These branching structures, formed through a GlcNAc β 1,6Gal linkage, are important due to their prevalence in humans (99 % of population) and are referred to as “I antigens” (no branching is referred to as “i antigen”). Mucins across most tissues will have this branching¹⁷. Complementary to the screening carried out in this report, Amuc_1666GH² has previously been shown to have a preference for LacNAc⁴⁴ and Amuc_0771GH³⁵ has also previously been shown to have activity against mucin core 1 and core 2 structures, both β 1,3 linkages⁴⁵”

8. We feel that the following statement lacks proper context and clarity, even when considering the surrounding sentences: “It was generally difficult to make any observations about how the different structures were related to the observed specificities. However, there was one feature where we felt we could speculate, but frustratingly it was untestable in this work.”

9. In the introduction, we suggest the authors speak more to the biological implications of Akkermansia mucin glycan degradation, and the context in which it operates, in more detail. Being able to completely break down the glycans to the backbone with just a cocktail of isolated enzymes is a significant finding in the context of Akkermansia, but why is that glycan breakdown significant in its own right towards health and disease?

10. “The mucosal surface of the human large intestine is predominantly comprised of gel-forming secreted mucins and approximately 80 % of this glycoprotein is O-glycan.”

What are the authors referring to by “80% of mucin is O-glycan”; by mass?

Reviewer #3

(Remarks to the Author)

The manuscript about the ‘Carbohydrate-active enzymes from Akkermansia muciniphila breakdown mucin O-glycans to completion’ by Cassie R. Bakshani et al. describes an extensive biochemical and structural characterization of the numerous carbohydrate active enzymes, present in the genome of Akkermansia muciniphila, and showing hydrolytic activity on mucin or mucin-related glycan/oligosaccharide structures. Activities of recombinantly produced, pure enzymes were screened by various (complementary) methods such as thin-layer chromatography, colorimetric assays on artificial substrates, or product analyses by anion-exchange chromatography, as best adapted to the nature of the substrate. The study is complemented with physiological growth experiments and pangenomic analyses to highlight the importance of the complete breakdown of this complex substrate present in the human intestine by A. muciniphila. Importantly, the study provides detailed experimental, biochemical proof of hydrolytic activities towards mucin and mucin-related substrates, thus ideally complementing previous studies that have analyzed mucin metabolism physiologically or by transcriptomics, such as those reported by Glover JS Sci. Rep. 2022 or Davey LE Nat Microbiol. 2023.

A total of fifty-five glycoside hydrolases, twelve sulfatases, and one polysaccharide lyase were characterized, an impressive feat of strength! The results allow to conclude that this important intestinal verrucomicrobiales, A. muciniphila, is capable of fully degrading host mucin and similar glycan structures down to the core GalNAc in a synergistic, orchestrated manner. A very important and noteworthy aspect of the work is that the enzymes were tested against a very broad range of substrates, which is not always the case, a fact that helps to pin down particular specificities, exclude others and understand the finetuned orchestration of the breakdown of host mucins or related substrates, the latter being only present through animal products in the diet. These biochemical data are completed by some growth experiments, pangenomic comparisons and structural (alphafold) analyses that help decipher and cautiously underscore potential absence or presence of particular hydrolytic activities.

Overall, the study reports an outstanding wealth of data and results, all of which are novel biochemical characterizations, highlighting that a large and complex set of enzymes is necessary for the efficient uptake and usage of the complex substrate mucin. The approach is perfectly valid and methods are well described and documented. Despite this extremely rich and dense data, supplemental material and information, the manuscript is well structured and comprehensively illustrated so that the reader can follow the overall workflow and seize the important messages. Taken together all these facts, I therefore believe that the manuscript is perfectly adapted and worthwhile publishing in Nature Microbiology. I only have spotted some minor issues that might be taken into account and a number of typos, wrong numbers, easily overlooked when describing such an immense load of data. (I am sure that I did not spot all mistakes, though!)

Comments

1. Abstract, last sentence: “how AM degrades mucin O-glycans, related structures and other host carbohydrates”.

The manuscript does report a few results on substrates that can only be taken up through diet, such as starch or the Galili

antigen – this does not become entirely clear by this statement (summarized as “host carbohydrates”). I suggest reformulating the sentence to clarify this, even though Amuc does not grow on starch for example as sole carbon source.

2. Results page 3, lines 118-119: “Interestingly, there was some indication that Amuc_0625GH33 could release Neu5Gc more efficiently than the other enzymes, but this was not explored further.”

I don't want to ask to explore this further but the statement is intriguing, begging to know why the authors find this an interesting aspect? Could a short comment be included in additional discussions?

3. Page 7 paragraph from 322 to 332. The authors highlight here that one of the Amuc GH89 enzymes, Amuc_0060, has similarities to the human homolog active against heparan sulfate. They evoke that mutations of this enzyme leads to specific syndromes for which no treatments are available. This statement is a bit “floating in the air” – it is a pity that such an important detail is not picked up in the discussion, since, in addition to the “major impacts of this being the ability to characterize different mucins in much more detail and in a higher throughput manner” perhaps some of the enzyme characterizations will also help better understand and find ways of tackling human disorders involving homologous enzymes?

4a. Page 10 lines 483-486. The expression is such that the reader has strong empathy with the authors in respect to the frustration – and is frustrated in turn; is this untestable because of inaccessibility of adequate substrates? If so, it should be mentioned. If not, it should be tested.

4b. And following up on the tempted structural speculation of Amuc GH20 enzymes, why not include an indication or outlook to the paragraph lines 497 to 506, opening to the fact that the presence of varying sulfation patterns could be accepted by some of the (redundant) enzymes? Isn't this summarizing paragraph tailored for being picked up in the discussion?

5a. Discussion. Overall, I think the discussion needs to be reworked a bit. First, I would reorganize the construction (order of the paragraphs) to finish on a more positive note than the enzymes for which no activity has been found. Secondly, when reading the discussion, one gets the impression that more is said about the GH16 enzymes (from a previous study) than on the huge number of novel data that has been experimentally dissected in this study. Figure 6 might deserve a bit more description. I think some of the key features should be highlighted here in a more explicit way, rather than in the general summary presented at the end of the first paragraph (that, by the way, could be the last concluding paragraph). Mentioning the difficulties to assess certain specificities because of lack of suitable substrates could be included here, rather than in the results section and be discussed as a perspective.

5b. Discussion, last paragraph about the GH57 enzyme. For non-specialist readers the link between Amuc GTs, exopolysaccharides and GH57, in particular, does not become clear from the text here, nor in the short description in supplementary information, lines 908 to 916. Indeed, several GH57 enzymes have been reported to have transglycosylating activities, for characterized enzymes to date mainly observed on 1,4-alpha-glucose containing substrates, such as starch, amylopectin or malto-oligosaccharides. This transglycosylating activity could indeed be linked to the recycling of taken up sugars to synthesize capsular or exopolysaccharides. This enzyme might be in some way linked to the starch active enzymes? As far as I could see, besides starch, dextran and maltose, no intermediate size (DP3 to 7) alpha-linked glucose sugars were tested for this enzyme, which might be the missing link (or not)? I am not especially asking for additional tests.... but perhaps opening up the discussion about potential missing/additional substrates that could be tested, despite the huge amount already tested.

Typos

1. Page 2 line 56 : “The details of O-glycan structure varies along” should be changed to “The details of O-glycan structure vary along”

2. Page 4 lines 133-134: “..., so these glycan epitopes are first partially broken down by other CAZymes encoded in the AM genome to produce blood group H structures first.”

The sentence contains a repeat of “first” – one of both should be deleted.

3. Page 6 line 273: in “however, contrary to this, Tn-antigen was not a degraded” “a” should be deleted.

4. page 8 line 370-371: “...mucin core 1 and core 2 structures, both β 1,3 linkages”

I feel that this needs to be changed to “which both have β 1,3 linkages” or “both containing β 1,3 linkages”. Perhaps also cite supplementary figure 4 here, where these structures are schematically presented.

5. Page 10 line 495 “Supplementary Figure 33”. This should be Supplementary Figure 31 (phylogeny of GH20) and not 33 (phylogeny of GH84)

6. Page 12 line 574 change “the disaccharide continud to increase” “the disaccharide continued to increase”

7. Page 12 line 619. “subfamily 11 enzymes Amuc_1033Sulf 16 and Amuc_1074Sulf 16”

I suppose the authors meant Amuc_1033Sulf 11 and Amuc_1074Sulf 11.

8. Page 14 Discussion second line (675). I believe there is an error in one of the enzymes' names. The previous work was on GH16 enzymes but the list also includes Amuc_2136GH20; possibly this should be replaced by Amuc2108GH16

9. Page 18 line 900 – a leftover (refs) should be replaced by the corresponding references; line 902 “produced by some organisms and is two glucose monosaccharides” should be replaced by “produced by some organisms and composed of two glucose monosaccharides”

10. Page 19 Supplementary information line 909: Supplementary Figure 20 is a phylogenetic tree, instead I believe the authors refer to Supplementary Figure 28 which includes the colorimetric assay on pNP- -Glc.

11. Page 19 Supp. Information line 958: “There has been not BGA structure crystallised with a GH109...” this sentence is a bit awkward why not say “No GH109 enzyme has been crystallized with BGA so far...”?

12. Legend of Figure 2 c. “This layer chromatography” should be changed to “Thin layer chromatography”

13. Legend of Figure 4 d. “each glycan has been given a different colour, apart from xxx” – xxx should be replaced

14. Legend of Supplementary Figure 5. “... these are placed in a separate columns” should be replaced by either “in separate columns” or “in a separate column”; I suppose that the numbers all on top of the horizontal gene denomination correspond to

identity percentage (i.e. 87 above one of the most left GH2 Amuc_0539) – but this is not specified anywhere. If this is the case it should be explicitly mentioned in the legend, if not these numbers need explanation.

15. Supplementary Figure 7 panels a and b. The enzymes names only have the Amuc gene number but do not carry the GH family designation as in all of the other Figures.

16. Supplementary Figure 42 b. The TLC panels in b do not indicate which substrate is tested after pretreatment with Amuc_1621GH77.

Decision Letter:

6th June 2024

Dear Lucy,

Thank you for your patience while your manuscript "Carbohydrate-active enzymes from Akkermansia muciniphila breakdown mucin O-glycans to completion" was under peer review at Nature Microbiology. It has now been seen by our referees, whose expertise and comments you will find at the end of this email. In the light of their advice, we have decided that we cannot offer to publish your manuscript in Nature Microbiology.

From the reports, you will see that while they find your work of some potential interest, the referees raise concerns about the advance your findings represent over earlier work and the strength of the novel conclusions that can be drawn at this stage. In particular, while some of the referees are interested in the work and find it useful for the field, we find the concerns raised by referee #1 important, in particular the depth of insight provided for the identified enzymes i.e. this is a broad preliminary analysis rather than a detailed mechanistic resource, together with the technical concerns raised. Unfortunately, these criticisms are sufficiently important as to preclude publication of your work in Nature Microbiology.

Although we regret that we cannot offer to publish your paper in Nature Microbiology given these reviews, I have discussed your manuscript and the reviewers' comments with our colleagues at Nature Communications. They would send the appropriately revised version out for further review if you transfer the revised manuscript to Nature Communications. Should you wish to have your revised paper considered by Nature Communications, please use the link to the Springer Nature manuscript transfer service below once the revision is ready, and include a point-by-point response to the reviewers' concerns.

Please note that Nature Communications are satisfied that the findings represent a sufficient advance for publication and acknowledge the resource value of your work. While the editors at Nature Communications will not insist on re-doing all biochemical assays with lysed cells as suggested by reviewer #1, they ask you to clearly outline (including in the abstract) that your work focuses on enzyme activity on the cell surface and more importantly provide clarifications on reviewer #1's concern on the lack of activity in whole cell assays throughout your manuscript. Related to this, please avoid statements about differences in activity that are not backed up by data as highlighted by reviewers #1 and #2, and include quantitative data on enzyme activity instead of qualitative assessment (reviewer #1). Eventually, please improve clarity and data presentation throughout your whole manuscript along the lines suggested by all three referees.

Your handling editor at Nature Communications would be Dr. Madlen Luckner (madlen.luckner@nature.com). If there is anything you would like to discuss before transferring the paper and its reviews, please don't hesitate to contact her by e-mail.

Please note that Nature Communications is a fully open access journal. For information about article processing charges, open access funding, and advice and support from Springer Nature, please consult the Nature Communications Open Access page (www.nature.com/ncomms/open_access/index.html).

To transfer your manuscript please use our manuscript transfer portal. You will not have to re-supply manuscript metadata and files, unless you wish to make modifications. For more information, please see our [manuscript transfer FAQ](http://www.nature.com/authors/author_resources/transfer_manuscripts.html?WT.mc_id=EMI_NPG_1511_AUTHORTRANSF&WT.ec_id=AUTHOR) page.

I am sorry that we cannot be more positive on this occasion, but hope that you find the referees' comments helpful when preparing your paper for resubmission elsewhere.

Yours sincerely,

Reviewer Expertise:

Referee #1: microbiome, biochemistry
Referee #2: bacteria-mucin interactions
Referee #3: glycobiology

Reviewers Comments:

Reviewer #1 (Remarks to the Author):

Bakshani et al. report a lot of initial f data that may be valuable to the glycobiology field. The manuscript describes the activities

of 55 glycoside hydrolases, 12 sulfatases and a polysaccharide lyase from *Akkermansia muciniphila*. This manuscript provides some insights into the mechanisms that AM utilizes to degrade complex glycans from a variety of sources including porcine gastric mucin, human milk oligosaccharides, and plant polysaccharides. While this work attempts to cover a lot of enzymes, the manuscript does not have a cohesive story and good deal of the results presented are the bare minimum it would take to begin an analysis. In many respects, it appears to be 2-3 papers worth of preliminary data put together. It is hard to imagine that it will make a strong impact on the field because it is a survey of initial conclusions one can make about these enzymes. Furthermore, there are significant concerns about the way key assays were performed. Major and minor comments follow. Taken together, this manuscript cannot be accepted in its current form without major additional experiments and a significant reorganization.

Major:

1. This manuscript is a challenge to read and needs considerable work to make this more approachable for readers. In its current state it reads like a statement of observations and needs significant changes to correct this. The manuscript reads like it was meant to be separated into 2 or more manuscripts. I am not sure the starch degrading enzymes are necessary in this manuscript or the high mannose N-glycoproteins sections.
2. I have concerns with the way the whole cell assays were performed. Several instances throughout the manuscript these whole cell assays do not degrade relatively simple sugars. Do the authors find this concerning? Clarification and commentary are essential.
3. Cell lysate experiments should be conducted against all substrates, not just whole cell assays with substrates. This is especially important since so many whole cell experiments showed no activity.
4. Several figures need reworking to make them easier to view. The TLC images should be scaled similarly so the lanes align in many supplemental figures. Additional annotation of TLC images, like labeling specific lanes, would improve clarity.
5. The pNP assays should be conducted again and absorbance measurements should be taken rather than images. In no way are photos of a color change acceptable as data on enzyme assays. Color changes can be measured precisely and should be, rather than a yes/no report that barely passes the bar as an initial experiment one might show at lab meeting. Additionally, this would allow for rates to be reported, which should be.
6. Why are Michaelis-Menten kinetics performed for Amuc_0778 and not others? This seems odd/out of place. As enzyme scientists know, Michaelis-Menten kinetics are critical for a complete and appropriate understanding of enzyme function.
7. It is not clear that GalNAc is being released in Figure 1. Why is a more thorough experiment not conducted for PGMIII like what was performed for BSM (Suppl Fig 6)? This is one of the most important findings in the study but the control lane for GalNAc is the furthest from the experimental lane. Additionally, in figure 1D the first two lanes are quite messy and not much can be concluded from them.
8. It is mentioned that fucosidases cannot degrade intact mucin oligosaccharides. I suggest the authors conduct additional experiments where PGMIII is incubated with a panel of other enzymes and followed up with these fucosidases, like what is done in Suppl Fig 6. I think that incubating GH16 enzymes prior would be especially useful.
9. Several instances throughout the manuscript the author claims that the experiments conducted suggested one enzyme being more efficient than another and that it was not pursued further. I think that such statements should not be included without data to back up these claims. These repeated statements significantly contribute to the sense that this is an initial report that adds little to the field without the further, essential follow-up work. Furthermore, adequate quantification of enzyme activity using the pNP assays or additional HPAEC-PAD assays would greatly increase the value of Figure 6 for the field, as accurate comparisons could be made between enzymes at each step of mucus degradation.
10. Several growth curves look abnormal and should be performed again. It is also unclear what the multiple lines on each graph represent.
11. Do you think some enzymes that had low activity might have low activity due to a different pH optimum? This may be particularly important for sequences that lack a signal peptide and are not trafficked outside the cell. It is essential that pH screens are performed as it is common knowledge that enzymes have pH optima that change their activities. Without this information, little can be learned from enzymes with no activity at a single pH.
12. Please clarify how partial activity is being assigned.
13. I think the AF2 models should be excluded unless they are used to explain certain substrate preferences. They do not add much to the overall manuscript. Similarly, the glycan models in Supplemental 4 do not significantly add to the manuscript.
14. The discussion section of the manuscript fails to successfully present the key takeaway points from the results. Further expansion on the importance and relevance of key findings needs to be emphasized. Additionally, the introduction of mucinosomes (line 697) is confusing and does not fit in the context of the presented results.

Minor:

1. Several typos throughout:
 - a. Oligosaccharides is misspelled in Supplementary Figure 2,3
 - b. Line 273 – “Tn-antigen was not a degraded”
 - c. Line 574 – “continud”
 - d. Line 575 – “degradatioonto”
 - e. Line 577 – “patter”
 - f. Line 990 – “(refs)” – missing reference?
 - g. Figure 4 caption – “...apart from xxx For sulfated...” – not sure what the xxx is referring to
2. Can a representation of colominic acid be added to Supplemental 1E?
3. Line 91 – Disagreement in the proteins mentioned and in supplemental 1 Amuc_0625 is 0623 in supplemental and 1825 is 1835 in supplemental 1.
4. Line 118 – I don't think statements about rates should be included when they were not pursued further. Also, this line references Supplemental 5, however the correct figure to reference is Supplemental 6.
5. Line 123 – What commercially available HMOs were assayed? Were they a mixture of several different HMOs or was just one representative HMO chosen for each group?
6. Line 142-144 – This also indicates that it prefers 1,2 which is not mentioned.
7. Line 185-188 – Where there any differences in active site architecture that would allow for a BGB glycan to fit into the active

sites of the two “active” proteins while not fitting into the “inactive” protein? A superimposition could be performed with the model in Supplemental Figure 17.

8. Line 190 – You state “one” protein while there are two proteins in the GH27 family shown in figure 1. It appears that Amuc_1220 was mislabeled and should be in the GH89 family.

9. Line 201 - Was there an increase in galactose when preincubated with sialidases/fucosidases? It appears in Supplementary Figure 11 that we see less of galactose when preincubated with sialidase/fucosidase. Could the authors comment on this?

10. Line 216 – the heat map has two GH97 enzymes but the text says one.

11. Other peaks are in Figure 1C that are not labeled. Can the authors comment on these unlabeled peaks?

12. Line 311 – are there other substrates to test?

13. Line 322-326 – this section needs revising and is not appropriate for a results section.

14. Line 427-428 – the wording here is awkward.

15. Line 429-433 – Consider restructuring here. Perhaps enzyme activities should be described first.

16. Line 484-484 – what is the one feature that you could speculate on? It is odd to make this statement and then not describe what it is.

17. Line 485 – Why are you not able to explore biochemically?

18. Line 508 – I don’t think this title is appropriate. I think it is more accurate to state that some of the glycans are completely degraded but these data do not “completely” degrade.

19. Line 534 – what are the predicted functions of the c terminal domain?

20. Line 555 – it is odd to have extensively studied this protein and not others.

21. Line 636 – I don’t think this section should be included in this manuscript.

22. Line 710 – this could be because different pHs were not tested.

Reviewer #2 (Remarks to the Author):

Crouch et al. provide a complete picture of colonic mucin glycan degradation by *Akkermansia muciniphilla*, a symbiont associated with gut health. Beyond extensive characterization of the multiple glycoside hydrolases, sulfatases, and lyases involved in mucin breakdown, the authors demonstrate that these enzymes can cooperatively break down complex mucin glycans to completion and provide evidence to suggest the intra/extracellular localization of these events. The authors also expand their characterization of *Akkermansia* enzymatic cleavage activities to other polysaccharides of high relevance to the microbiota, such as gangliosides and human milk oligosaccharides. Taken together, the characterization of these enzymes, and associated data concerning the order of enzyme activity and localization, represent a significant advance for the field. We recommend publication of this work provided that the following concerns regarding presentation of the findings can be addressed.

1. Results: α -sialidases

“Interestingly, there was some indication that Amuc_0625GH33 could release Neu5Gc more efficiently than the other enzymes, but this was not explored further. Release of sialic acid also carried out on porcine gastric mucin III (PGM III) assessed by thin layer

chromatography (TLC) indicates that Amuc_0625GH33 and Amuc_1825GH33 are releasing the highest amounts, but this was not quantified (Supplementary Figure 1)”

Taking the first sentence to refer to the chromatography data in Supplementary Figure 6 (box d), it is not apparent that the concentration of Neu5Gc is significantly higher in the Amuc_0625 treated sample compared to the Amuc_1835 sample; as such, it is not clear to us why the authors would make this claim. The subsequent discussion of the relative enzymatic activity as per qualitative TLC staining does not seem significant or supportive to the overall conclusions made by the authors, and we feel this was not presented convincingly in the absence of replicates or more rigorous characterization.

2. Results: Alpha-galactosidases

“Amuc_0480GH110 was able to release more galactose than Amuc_1463GH110, however this was not quantified or the reasons for this difference explored further in this study.”

Like the previous example, this point does not seem significant to the overall conclusions made by the authors, and without more analytical rigor or characterization, the statement is poorly supported.

3. “We could not find any activities for Amuc_0623GHnc or Amuc_1547GH177 during this work despite their associations with sialic acid hydrolysis (Supplementary Figure 1).”

A citation should be provided supporting the notion that Amuc_0623 and Amuc_1547 are associated with sialic acid hydrolysis. Additionally, reference 14 in the manuscript appears to address this aspect of Amuc_0623 activity.

4. We find that the fundamental understanding of *Akkermansia*’s mucin degradation capabilities afforded by this work represents an important addition to the field given the established prevalence of this microbe in gut health. Another point of significance for the field is the authors’ highly efficient recapitulation of *Akkermansia* enzymes to act on specific glycan motifs in whole mucin samples, and to furthermore analyze the glycan composition of these complex molecules. These enzyme cocktails could therefore be used to directly study glycan compositions in other contexts, such as gut microbiota dysbiosis and other such “diseased” mucus states.

We would, however like to address this quote from the discussion section: "One of the most significant aspects of this report was showing that we could use a cocktail of CAZymes from AM to reach the core GalNAcs of the O-glycan, thus demonstrating complete degradation of the constituent glycan of this complex substrate (Figure 1)."

We feel this aspect of the work, as significant as it was, was not properly highlighted in the results section. We felt that the wide breadth of enzyme activities characterized, and their relationships to other organisms and substrates other than mucin, received much more written focus. An effort should be made to streamline the writing to highlight the most significant results.

5. Supplemental figure 1: We believe there is a typo with the labeling as "Amuc1835" throughout the figure, where we are led to believe in the corresponding text that "Amuc1825" is the subject of the figure.

6. "An attempt to resolve the kinetic parameters of Amuc_0863GH105 was performed by running the reactions of Amuc_0778PL38 to completion (overnight), then adding Amuc_0863GH105 and following the loss of the double bond of the unsaturated uronic acid at 235nm. While linear decreasing initial rates were observed for the lower substrate concentrations, the higher concentration reactions were not reliable because of the absorbance ceiling of the spectrophotometer. Attempts to downscale or dilute the reactions did not provide data for which a reliable kinetic model could be fitted (Supplementary Figure 40). However, by isolating the highest observed initial rate for each substrate, a clear selectivity of Amuc_0863GH105 was observed towards CSA, followed by the three other GAGs at 36-40% relative activity (Figure 4, Supplementary Table 6)."

That the reaction has reached completion does not follow from it being run overnight, and the authors make no mention of how the substrate concentration was controlled, only that it was measured.

In terms of presentation, this section could be edited to be much clearer and more concise.

7. There are several instances such as the quotes below in the results section where the authors extensively qualify the broader significance of the data to the field or highlight data from other studies. While this is pertinent information that strengthens the impact of the results, we believe that these statements are better placed in the discussion section, to enable better understanding of the results that the authors themselves generated. Alternatively, the writing can be condensed, with these statements made in direct reference to the data.

α -galactosidases

"The whole cell assay results from the blood group hexasaccharides do show breakdown, but the type of glycobiochemistry was not further explored in this work. However, we previously characterised GH16 activity against these substrates and this activity is prominent on the outside of the cell (15; Supplementary Figure 8)."

"Amuc_1187GH27 can hydrolyse Gal α 1,3Gal/GalNAc and globotriose, but less activity was seen against P1 antigen. Gal α 1,3Gal (Galili antigen) is expressed by most mammals and some other animals, but not humans, apes, and Old World monkeys²⁹. Therefore, gut microbes would only have access to this structure through animal products in the diet. Conversely, the similar Gal α 1,4Gal epitopes of P1 and Pk antigens are generally present in human tissues, so they would be present in the mucosal surface of the human large intestine. P1 and Pk structures are receptors for a variety of pathogens and their toxins³⁰."

α -GalNAc'ases

"Forssman antigen is normally present in only 0.01 % of the human population but in that relatively small population this antigen would be present in the GI tract³⁴. It can change susceptibility to some diseases and is also expressed in some cancers, but its true prevalence and impact on different diseases is understudied³⁵. BGA is much more prevalent within the human population with this epitope being expressed in 10-40 % of the population depending on where you sample globally³⁶. We were only able to test BGA type II in this study, but given the broad activity of Amuc_0216GH36, it is likely that it will act on all BGA structures."

Investigating β -galactosidase activity

"These branching structures, formed through a GlcNAc β 1,6Gal linkage, are important due to their prevalence in humans (99 % of population) and are referred to as "I antigens" (no branching is referred to as "i antigen"). Mucins across most tissues will have this branching¹⁷. Complementary to the screening carried out in this report, Amuc_1666GH2 has previously been shown to have a preference for LacNAc⁴⁴ and Amuc_0771GH35 has also previously been shown to have activity against mucin core 1 and core 2 structures, both β 1,3 linkages⁴⁵"

8. We feel that the following statement lacks proper context and clarity, even when considering the surrounding sentences: "It was generally difficult to make any observations about how the different structures were related to the observed specificities. However, there was one feature where we felt we could speculate, but frustratingly it was untestable in this work."

9. In the introduction, we suggest the authors speak more to the biological implications of

Akkermansia mucin glycan degradation, and the context in which it operates, in more detail. Being able to completely break down the glycans to the backbone with just a cocktail of isolated enzymes is a significant finding in the context of Akkermansia, but why is that glycan breakdown significant in its own right towards health and disease?

10. "The mucosal surface of the human large intestine is predominantly comprised of gel-forming secreted mucins and approximately 80 % of this glycoprotein is O-glycan."

What are the authors referring to by "80% of mucin is O-glycan"; by mass?

Reviewer #3 (Remarks to the Author):

The manuscript about the 'Carbohydrate-active enzymes from Akkermansia muciniphila breakdown mucin O-glycans to completion' by Cassie R. Bakshani et al. describes an extensive biochemical and structural characterization of the numerous carbohydrate active enzymes, present in the genome of Akkermansia muciniphila, and showing hydrolytic activity on mucin or mucin-related glycan/oligosaccharide structures. Activities of recombinantly produced, pure enzymes were screened by various (complementary) methods such as thin-layer chromatography, colorimetric assays on artificial substrates, or product analyses by anion-exchange chromatography, as best adapted to the nature of the substrate. The study is complemented with physiological growth experiments and pangenomic analyses to highlight the importance of the complete breakdown of this complex substrate present in the human intestine by A. muciniphila. Importantly, the study provides detailed experimental, biochemical proof of hydrolytic activities towards mucin and mucin-related substrates, thus ideally complementing previous studies that have analyzed mucin metabolism physiologically or by transcriptomics, such as those reported by Glover JS Sci. Rep. 2022 or Davey LE Nat Microbiol. 2023.

A total of fifty-five glycoside hydrolases, twelve sulfatases, and one polysaccharide lyase were characterized, an impressive feat of strength! The results allow to conclude that this important intestinal verrucomicrobiales, A. muciniphila, is capable of fully degrading host mucin and similar glycan structures down to the core GalNAc in a synergistic, orchestrated manner. A very important and noteworthy aspect of the work is that the enzymes were tested against a very broad range of substrates, which is not always the case, a fact that helps to pin down particular specificities, exclude others and understand the finetuned orchestration of the breakdown of host mucins or related substrates, the latter being only present through animal products in the diet. These biochemical data are completed by some growth experiments, pangenomic comparisons and structural (alphafold) analyses that help decipher and cautiously underscore potential absence or presence of particular hydrolytic activities. Overall, the study reports an outstanding wealth of data and results, all of which are novel biochemical characterizations, highlighting that a large and complex set of enzymes is necessary for the efficient uptake and usage of the complex substrate mucin. The approach is perfectly valid and methods are well described and documented. Despite this extremely rich and dense data, supplemental material and information, the manuscript is well structured and comprehensively illustrated so that the reader can follow the overall workflow and seize the important messages. Taken together all these facts, I therefore believe that the manuscript is perfectly adapted and worthwhile publishing in Nature Microbiology. I only have spotted some minor issues that might be taken into account and a number of typos, wrong numbers, easily overlooked when describing such an immense load of data. (I am sure that I did not spot all mistakes, though!)

Comments

1. Abstract, last sentence: "how AM degrades mucin O-glycans, related structures and other host carbohydrates".

The manuscript does report a few results on substrates that can only be taken up through diet, such as starch or the Galili antigen – this does not become entirely clear by this statement (summarized as "host carbohydrates"). I suggest reformulating the sentence to clarify this, even though Amuc does not grow on starch for example as sole carbon source.

2. Results page 3, lines 118-119: "Interestingly, there was some indication that Amuc_0625GH33 could release Neu5Gc more efficiently than the other enzymes, but this was not explored further."

I don't want to ask to explore this further but the statement is intriguing, begging to know why the authors find this an interesting aspect? Could a short comment be included in additional discussions?

3. Page 7 paragraph from 322 to 332. The authors highlight here that one of the Amuc GH89 enzymes, Amuc_0060, has similarities to the human homolog active against heparan sulfate. They evoke that mutations of this enzyme leads to specific syndromes for which no treatments are available. This statement is a bit "floating in the air" – it is a pity that such an important detail is not picked up in the discussion, since, in addition to the "major impacts of this being the ability to characterize different mucins in much more detail and in a higher throughput manner" perhaps some of the enzyme characterizations will also help better understand and find ways of tackling human disorders involving homologous enzymes?

4a. Page 10 lines 483-486. The expression is such that the reader has strong empathy with the authors in respect to the frustration – and is frustrated in turn; is this untestable because of inaccessibility of adequate substrates? If so, it should be mentioned. If not, it should be tested.

4b. And following up on the tempted structural speculation of Amuc GH20 enzymes, why not include an indication or outlook to the paragraph lines 497 to 506, opening to the fact that the presence of varying sulfation patterns could be accepted by some of the (redundant) enzymes? Isn't this summarizing paragraph tailored for being picked up in the discussion?

5a. Discussion. Overall, I think the discussion needs to be reworked a bit. First, I would reorganize the construction (order of the paragraphs) to finish on a more positive note than the enzymes for which no activity has been found. Secondly, when reading the discussion, one gets the impression that more is said about the GH16 enzymes (from a previous study) than on the huge

number of novel data that has been experimentally dissected in this study. Figure 6 might deserve a bit more description. I think some of the key features should be highlighted here in a more explicit way, rather than in the general summary presented at the end of the first paragraph (that, by the way, could be the last concluding paragraph). Mentioning the difficulties to assess certain specificities because of lack of suitable substrates could be included here, rather than in the results section and be discussed as a prospective.

5b. Discussion, last paragraph about the GH57 enzyme. For non-specialist readers the link between Amuc GTs, exopolysaccharides and GH57, in particular, does not become clear from the text here, nor in the short description in supplementary information, lines 908 to 916. Indeed, several GH57 enzymes have been reported to have transglycosylating activities, for characterized enzymes to date mainly observed on 1,4-alpha-glucose containing substrates, such as starch, amylopectin or malto-oligosaccharides. This transglycosylating activity could indeed be linked to the recycling of taken up sugars to synthesize capsular or exopolysaccharides. This enzyme might be in some way linked to the starch active enzymes? As far as I could see, besides starch, dextran and maltose, no intermediate size (DP3 to 7) alpha-linked glucose sugars were tested for this enzyme, which might be the missing link (or not)? I am not especially asking for additional tests.... but perhaps opening up the discussion about potential missing/additional substrates that could be tested, despite the huge amount already tested.

Typos

1. Page 2 line 56 : "The details of O-glycan structure varies along" should be changed to "The details of O-glycan structure vary along"
2. Page 4 lines 133-134: "..., so these glycan epitopes are first partially broken down by other CAZymes encoded in the AM genome to produce blood group H structures first."
The sentence contains a repeat of "first" – one of both should be deleted.
3. Page 6 line 273: in "however, contrary to this, Tn-antigen was not a degraded" "a" should be deleted.
4. page 8 line 370-371: "...mucin core 1 and core 2 structures, both β 1,3 linkages"
I feel that this needs to be changed to "which both have β 1,3 linkages" or "both containing β 1,3 linkages". Perhaps also cite supplementary figure 4 here, where these structures are schematically presented.
5. Page 10 line 495 "Supplementary Figure 33". This should be Supplementary Figure 31 (phylogeny of GH20) and not 33 (phylogeny of GH84)
6. Page 12 line 574 change "the disaccharide continud to increase" "the disaccharide continued to increase"
7. Page 12 line 619. "subfamily 11 enzymes Amuc_1033Sulf 16 and Amuc_1074Sulf 16"
I suppose the authors meant Amuc_1033Sulf 11 and Amuc_1074Sulf 11.
8. Page 14 Discussion second line (675). I believe there is an error in one of the enzymes' names. The previous work was on GH16 enzymes but the list also includes Amuc_2136GH20; possibly this should be replaced by Amuc2108GH16
9. Page 18 line 900 – a leftover (refs) should be replaced by the corresponding references; line 902 "produced by some organisms and is two glucose monosaccharides" should be replaced by "produced by some organisms and composed of two glucose monosaccharides"
10. Page 19 Supplementary information line 909: Supplementary Figure 20 is a phylogenetic tree, instead I believe the authors refer to Supplementary Figure 28 which includes the colorimetric assay on pNP- -Glc.
11. Page 19 Supp. Information line 958: "There has been not BGA structure crystallised with a GH109..." this sentence is a bit awkward why not say "No GH109 enzyme has been crystallized with BGA so far...?"
12. Legend of Figure 2 c. "This layer chromatography" should be changed to "Thin layer chromatography"
13. Legend of Figure 4 d. "each glycan has been given a different colour, apart from xxx" – xxx should be replaced
14. Legend of Supplementary Figure 5. "... these are placed in a separate columns" should be replaced by either "in separate columns" or "in a separate column"; I suppose that the numbers all on top of the horizontal gene denomination correspond to identity percentage (i.e 87 above one of the most left GH2 Amuc_0539) – but this is not specified anywhere. If this is the case it should be explicitly mentioned in the legend, if not these numbers need explanation.
15. Supplementary Figure 7 panels a and b. The enzymes names only have the Amuc gene number but do not carry the GH family designation as in all of the other Figures.
16. Supplementary Figure 42 b. The TLC panels in b do not indicate which substrate is tested after pretreatment with Amuc_1621GH77.

Version 1:

Decision Letter:

4th July 2024

Dear Lucy

Thank you for your letter asking us to reconsider our decision on your Article entitled "Carbohydrate-active enzymes from *Akkermansia muciniphila* breakdown mucin O-glycans to completion". After careful consideration we have decided that we would be willing to consider a revised version of your manuscript that addresses the points outlined in your point by point response.

However, please note that we are not convinced that your proposed revisions will address this concern raised by referee #1, which was shared by the editorial team:

While this work attempts to cover a lot of enzymes, the manuscript does not have a cohesive story and good deal of the results presented are the bare minimum it would take to begin an analysis. In many respects, it appears to be 2-3 papers worth of preliminary data put together. It is hard to imagine that it will make a strong impact on the field because it is a survey of initial

conclusions one can make about these enzymes.

Therefore if you decide to submit a revised manuscript, we will recruit an additional referee with more general microbiome expertise to comment on the general interest and resource value of the dataset for the field, which would be necessary for publication in Nature Microbiology.

Along with your revised manuscript, you should also submit a separate point-by-point response to all of the concerns raised by the referees, in each case describing what changes have been made to the manuscript or, alternatively, if no action has been taken, providing a compelling argument for why that is the case. If we feel that a substantial attempt has been made to address the referees' comments, this response will be sent back to the referees - along with the revised manuscript - so that they can judge whether their concerns have been addressed satisfactorily or otherwise.

I should stress, however, that we would be reluctant to trouble our referees again unless we thought that their comments had been addressed in full.

- ensure it complies with our format requirements for Letters as set out in our guide to authors at www.nature.com/nmicrobiol/authors/index.html

- state in a cover note the length of the text, methods and legends; the number of references and the number of display items.

Please ensure that all correspondence is marked with your Nature Microbiology reference number in the subject line.

Please use the following link to submit your revised manuscript:

Link Redacted

We hope to receive your revised paper within four weeks. If you cannot send it within this time, please let us know so that we can close your file. In this event, we will still be happy to reconsider your paper at a later date so long as nothing similar has been accepted for publication at Nature Microbiology or published elsewhere in the meantime. Should you miss the four-week deadline and your paper is eventually published, the received date will be that of the revised, not the original, version.

I would appreciate it if you could tell me if you think you will be able to submit a revised manuscript, and also the likely timescale.

I look forward to hearing from you soon.

Yours sincerely,

Version 2:

Reviewer comments:

Reviewer #1

(Remarks to the Author)

Several changes to the original manuscript have improved quality and clarity. The addition of LC-FLD-ESI-MS experiments added to the rigor of the assays. Additionally, several figures are much easier to interpret and many figures have been adequately condensed. Changes to the organization and writing of the manuscript have made it much easier to read. However, the discussion is still lacking in its attempt to summarize all important results, and explain the significance of the breadth of this work. Similarly, several fundamental concerns were not adequately addressed in the revisions. To fully understand how AM interacts with even simple substrates, cell lysate experiments should be conducted. Additionally, pH screens and quantitative pNP assays are critical to completely understanding enzyme function and their importance should not be overlooked. Overall, this manuscript was improved upon in several ways, and presents important findings to the glycobiology field. Yet, these findings could be more impactful with a more thorough analysis of AM's capabilities.

Reviewer #2

(Remarks to the Author)

The authors have implemented all points of criticism and thereby substantially improved the manuscript.

Reviewer #3

(Remarks to the Author)

The authors have substantially rewritten and reorganized their results and discussions to respond to reviewers comments. They also have included some relevant additional experiments, clarifying some of the raised issues. In my opinion the authors have done very well and have largely enhanced the quality of the manuscript. It is now suitable for publication. I have only found two minor typos that could be corrected :

Lines 85-86: was performed using "with" liquid chromatography-fluorescence (delete "with")
Line 240: and a GH16 a range "of" sizes and compositions of O-glycan (include "of" in this sentence)

Reviewer #4

(Remarks to the Author)

This study presents a systematic biochemical examination of the predicted carbohydrate active enzymes in Akkermansia muciniphila. These enzymes were all tested for activity on mucin or related glycans. I believe the dataset will be extremely useful reference for those interested in the biology of Akkermansia and for folks in the glycobiology field.

Decision Letter:

Our ref: NMICROBIOL-24041192B

16th October 2024

Dear Lucy,

Thank you for submitting your revised manuscript "Carbohydrate-active enzymes from Akkermansia muciniphila breakdown mucin O-glycans to completion" (NMICROBIOL-24041192B). It has now been seen by the original referees and their comments are below. Given the ongoing concerns from reviewer #1, I asked reviewer #3 to comment on their concerns. Reviewer #3 did not think these additional experiments were required for publication. As mentioned, I also recruited an additional referee with general microbiome expertise focusing on Akkermansia. This referee agreed that the data would be a good resource for the field. The reviewers find that the paper has improved in revision, and therefore we'll be happy in principle to publish it in Nature Microbiology, pending minor revisions to satisfy the referees' final requests and to comply with our editorial and formatting guidelines.

Thank you again for your interest in Nature Microbiology Please do not hesitate to contact me if you have any questions.

Sincerely,

Reviewer #1 (Remarks to the Author):

Several changes to the original manuscript have improved quality and clarity. The addition of LC-FLD-ESI-MS experiments added to the rigor of the assays. Additionally, several figures are much easier to interpret and many figures have been adequately condensed. Changes to the organization and writing of the manuscript have made it much easier to read. However, the discussion is still lacking in its attempt to summarize all important results, and explain the significance of the breadth of this work. Similarly, several fundamental concerns were not adequately addressed in the revisions. To fully understand how AM interacts with even simple substrates, cell lysate experiments should be conducted. Additionally, pH screens and quantitative pNP assays are critical to completely understanding enzyme function and their importance should not be overlooked. Overall, this manuscript was improved upon in several ways, and presents important findings to the glycobiology field. Yet, these findings could be more impactful with a more thorough analysis of AM's capabilities.

Reviewer #2 (Remarks to the Author):

The authors have implemented all points of criticism and thereby substantially improved the manuscript.

Reviewer #3 (Remarks to the Author):

The authors have substantially rewritten and reorganized their results and discussions to respond to reviewers comments. They also have included some relevant additional experiments, clarifying some of the raised issues. In my opinion the authors have done very well and have largely enhanced the quality of the manuscript. It is now suitable for publication. I have only found two minor typos that could be corrected :

Lines 85-86: was performed using "with" liquid chromatography-fluorescence (delete "with")
Line 240: and a GH16 a range "of" sizes and compositions of O-glycan (include "of" in this sentence)

Reviewer #4 (Remarks to the Author):

This study presents a systematic biochemical examination of the predicted carbohydrate active enzymes in *Akkermansia muciniphila*. These enzymes were all tested for activity on mucin or related glycans. I believe the dataset will be extremely useful reference for those interested in the biology of *Akkermansia* and for folks in the glycobiology field.

Version 3:

Decision Letter:

10th December 2024

Dear Dr Crouch,

I am pleased to accept your Resource "Carbohydrate-active enzymes from *Akkermansia muciniphila* breakdown mucin O-glycans to completion" for publication in *Nature Microbiology*. Thank you for having chosen to submit your work to us and many congratulations.

Over the next few weeks, your paper will be copyedited to ensure that it conforms to *Nature Microbiology* style. We look particularly carefully at the titles of all papers to ensure that they are relatively brief and understandable.

You may wish to make your media relations office aware of your accepted publication, in case they consider it appropriate to organize some internal or external publicity. Once your paper has been scheduled you will receive an email confirming the publication details. This is normally 3-4 working days in advance of publication. If you need additional notice of the date and time of publication, please let the production team know when you receive the proof of your article to ensure there is sufficient time to coordinate. Further information on our embargo policies can be found here:

<https://www.nature.com/authors/policies/embargo.html>

Please note that *Nature Microbiology* is a Transformative Journal (TJ). Authors may publish their research with us through the traditional subscription access route or make their paper immediately open access through payment of an article-processing charge (APC). Authors will not be required to make a final decision about access to their article until it has been accepted. [Find out more about Transformative Journals](https://www.springernature.com/gp/open-research/transformative-journals)

Authors may need to take specific actions to achieve [compliance](https://www.springernature.com/gp/open-research/funding/policy-compliance-faqs) with funder and institutional open access mandates. If your research is supported by a funder that requires immediate open access (e.g. according to [Plan S principles](https://www.springernature.com/gp/open-research/plan-s-compliance)) then you should select the gold OA route, and we will direct you to the compliant route where possible. For authors selecting the subscription publication route, the journal's standard licensing terms will need to be accepted, including [self-archiving policies](https://www.nature.com/nature-portfolio/editorial-policies/self-archiving-and-license-to-publish). Those licensing terms will supersede any other terms that the author or any third party may assert apply to any version of the manuscript.

An online order form for reprints of your paper is available at <https://www.nature.com/reprints/author->

reprints.html"><https://www.nature.com/reprints/author-reprints.html>. All co-authors, authors' institutions and authors' funding agencies can order reprints using the form appropriate to their geographical region.

With kind regards,

P.S. Click on the following link if you would like to recommend Nature Microbiology to your librarian
<http://www.nature.com/subscriptions/recommend.html#forms>

** Visit the Springer Nature Editorial and Publishing website at http://editorial-jobs.springernature.com?utm_source=ejP_NMicro_email&utm_medium=ejP_NMicro_email&utm_campaign=ejp_NMicro>www.springernature.com/editorial-and-publishing-jobs for more information about our career opportunities. If you have any questions please click [here](mailto:editorial.publishing.jobs@springernature.com).**

Open Access This Peer Review File is licensed under a Creative Commons Attribution 4.0 International License, which permits use, sharing, adaptation, distribution and reproduction in any medium or format, as long as you give appropriate credit to the original author(s) and the source, provide a link to the Creative Commons license, and indicate if changes were made. In cases where reviewers are anonymous, credit should be given to 'Anonymous Referee' and the source.

Reviewer #1 (Remarks to the Author):

Bakshani et al. report a lot of initial data that may be valuable to the glycobiology field. The manuscript describes the activities of 55 glycoside hydrolases, 12 sulfatases and a polysaccharide lyase from *Akkermansia muciniphila*. This manuscript provides some insights into the mechanisms that AM utilizes to degrade complex glycans from a variety of sources including porcine gastric mucin, human milk oligosaccharides, and plant polysaccharides. While this work attempts to cover a lot of enzymes, the manuscript does not have a cohesive story and good deal of the results presented are the bare minimum it would take to begin an analysis. In many respects, it appears to be 2-3 papers worth of preliminary data put together. It is hard to imagine that it will make a strong impact on the field because it is a survey of initial conclusions one can make about these enzymes. Furthermore, there are significant concerns about the way key assays were performed. Major and minor comments follow. Taken together, this manuscript cannot be accepted in its current form without major additional experiments and a significant reorganization.

Thank you for taking the time to review our work and share your valuable insights. We were disappointed that you did not like what we have done more, but I am hoping to win you round on a few points.

Major:

1. This manuscript is a challenge to read and needs considerable work to make this more approachable for readers. In its current state it reads like a statement of observations and needs significant changes to correct this.

Yes, we agree with you. We wrote it in the order of degradation (capping to core) and hopefully to reduce the amount of flicking back and forth between sections a reviewer would have to do. We will now aim to put the work into a more digestible journal format.

The manuscript reads like it was meant to be separated into 2 or more manuscripts. I am not sure the starch degrading enzymes are necessary in this manuscript or the high mannose N-glycoproteins sections.

We set out to cover all AM CAZymes and that has naturally highlighted several puzzles and new avenues where full characterisation is beyond the scope of this manuscript. I can see how this work could be cut up into several papers, but we wanted one all-encompassing reference with a focus on host glycans. In terms of the "starch enzymes", I think it's important to highlight that we have screened them against the obvious substrates and not leave them out. In terms of the high-mannose N-glycan observation, we believe this is of significance due to the abundance of these in the gut, both from host and dietary sources. Mucins and secreted glycoproteins in the gut (e.g. antibodies) are decorated with these glycan modifications, so how different host-derived nutrients are accessed and cross fed between species will be of interest to the field.

2. I have concerns with the way the whole cell assays were performed. Several instances throughout the manuscript these whole cell assays do not degrade relatively simple sugars. Do the authors find this concerning? Clarification and commentary are essential.

I am not concerned with a lack of activity against some substrates, and we were not expecting that the surface-localised enzymes would be able to tackle the full panel of substrates that we have. With the whole cell assay work, we wanted to understand what capabilities are on the surface as the first keystone activities. AM likely does not want all activities occurring on the surface so it can keep nutrients for itself.

Since we first submitted, we have now been able to look at the glycans released from mucin during whole cell assays in greater detail. We labelled the reducing ends with a fluorescent

label and used liquid chromatography-fluorescence detection-electrospray ionization-mass spectrometry (LC-FLD-ESI-MS). We can determine the composition of the different glycans using the MS/MS data. The data shows that most of the O-glycans have a galactose at the reducing end, which are GH16 endo-O-glycanase products. This data provides more of a detailed analysis of AM surface activities when presented with a complex substrate.

3. Cell lysate experiments should be conducted against all substrates, not just whole cell assays with substrates. This is especially important since so many whole cell experiments showed no activity.

You don't state a reason for doing this, is this purely to look at AM capability? I am happy to do this if required, but I don't think it would add to the story. There's quite a bit of redundancy in the activity of recombinant enzymes against the panel of defined substrate we have, so I am unsure what would be gained unless we would be just checking what AM can breakdown. The only application I think would be interesting to do this for 6'S-Lewis X, where we couldn't find a way to degrade it, to see if AM can do this, but it wouldn't highlight the enzyme responsible. If we did whole cell lysate assays with mucin, I think there would be a lot of activities happening and it would be difficult untangle what was happening.

4. Several figures need reworking to make them easier to view. The TLC images should be scaled similarly so the lanes align in many supplemental figures. Additional annotation of TLC images, like labeling specific lanes, would improve clarity.

We will check all figures

5. The pNP assays should be conducted again and absorbance measurements should be taken rather than images. In no way are photos of a color change acceptable as data on enzyme assays. Color changes can be measured precisely and should be, rather than a yes/no report that barely passes the bar as an initial experiment one might show at lab meeting. Additionally, this would allow for rates to be reported, which should be.

Kinetics on pNP-sugars are meaningless and those numbers wouldn't tell you anything about how the enzyme behaves with its true substrate. We only use pNPs to identify the -1 sugar and as you can see from our photos, we are only getting one hit for some of the enzymes. I realise other people report pNP yes/no without showing any evidence and I am happy to remove this figure. The aim was to show that for the 'puzzle' enzymes we had tried to cover all the bases.

6. Why are Michaelis-Menten kinetics performed for Amuc_0778 and not others? This seems odd/out of place. As enzyme scientists know, Michaelis-Menten kinetics are critical for a complete and appropriate understanding of enzyme function.

We were able to perform kinetics for those enzymes active on GAGs as these substrates are available in large quantities and are relatively cheap. We would love to be able to perform kinetics on all enzymes vs substrates, but even if we did this by HPAEC the cost would run into the tens of thousands. We spent ~£20k on substrates alone for this work just to top-up what we already had in our collection. The other issue with HPAEC and O-glycans is that they aren't detected as well as other glycans. For instance, N-glycans do much better using this technique. Labelling substrates with a fluorophore to enhance detection could affect enzyme activity and labelling post-reaction introduces a lot of error with the clean-up processes. Ideally, I would like enough substrate to carry out assays on the spec, but that is not economically feasible.

The enzymology we have completed provides a wide selection of different specificities, and therefore new tools, for the breakdown of O-glycans and other host glycans.

7. It is not clear that GalNAc is being released in Figure 1. Why is a more thorough experiment not conducted for PGMIII like what was performed for BSM (Suppl Fig 6)? This is one of the most important findings in the study but the control lane for GalNAc is the furthest from the experimental lane.

We will re-order the chromatograms and try to present the data better to make everything clearer.

The great aspect of BSM is that it only has one O-glycan structure, so we know exactly what our substrate is. PGM is much more heterogeneous so you wouldn't be able to conclude much about the specificity of different enzymes. We would produce bar charts of amounts of different sugars released and we would not be able to analyse the data. I'm not sure what an assay against PGM, like the BSM one, would be aiming for, if you wouldn't mind expanding? However, I do have some alternative new data that might suffice – see point 8.

Additionally, in figure 1D the first two lanes are quite messy and not much can be concluded from them.

These lanes are to demonstrate that the enzyme cocktail releases a large amount of monosaccharides. They are to show the process we put the PGM through before testing for the core GalNAc with 1008. These are difficult to separate by TLC, however, galactose and GlcNAc stain blue and brown, respectively, and the GlcNAc runs slightly faster. You can see these on the TLC. We have altered the text to incorporate a better explanation.

8. It is mentioned that fucosidases cannot degrade intact mucin oligosaccharides.

I think you are referring to the sentences starting 139. This is poorly-worded, apologies. We added the fucosidases individually to PGM III and we could only see fucose being released by 1120. This means that for this intact mucin substrate only 1120 can access the capping fucose and this may vary for other types of mucin. We did not mean to imply that this was the only fucose that could ever be removed from PGM.

I suggest the authors conduct additional experiments where PGMIII is incubated with a panel of other enzymes and followed up with these fucosidases, like what is done in Suppl Fig 6. I think that incubating GH16 enzymes prior would be especially useful.

We completed a variety of assays with the panels of different enzymes and stopped the reactions in between. These were then labelled with procainamide and analysed by LC-FLD-ESI-MS as described above.

9. Several instances throughout the manuscript the author claims that the experiments conducted suggested one enzyme being more efficient than another and that it was not pursued further. I think that such statements should not be included without data to back up these claims. These repeated statements significantly contribute to the sense that this is an initial report that adds little to the field without the further, essential follow-up work.

We will remove these.

Furthermore, adequate quantification of enzyme activity using the pNP assays or additional HPAEC-PAD assays would greatly increase the value of Figure 6 for the field, as accurate comparisons could be made between enzymes at each step of mucus degradation.

We agree that this would add to the report, but it's not feasible as discussed above. However, we hope you can see the contribution the LC-FLD-ESI-MS analysis brings to the work as a whole.

10. Several growth curves look abnormal and should be performed again. It is also unclear what the multiple lines on each graph represent.

Data is not always perfect and it's not unusual (but relatively rare) to have one growth behaving slightly differently to the rest. I am happy to take the anomalies out, but I thought I was being honest by leaving them in. Each line is a growth in a 96-well plate and we will make this clearer in the text. From my experience, these growth curves look very consistent for being separate growths. If you could point out which ones you are unhappy with, I can repeat them.

11. Do you think some enzymes that had low activity might have low activity due to a different pH optimum? This may be particularly important for sequences that lack a signal peptide and are not trafficked outside the cell. It is essential that pH screens are performed as it is common knowledge that enzymes have pH optima that change their activities. Without this information, little can be learned from enzymes with no activity at a single pH.

From my experience, I don't believe in this case if we tested for activity at different pHs we would find the activity. For this microbe, in this environment, the optima are highly likely to be close to neutral. It's highly unlikely we missed activities based on this. Furthermore, our assays were run for a relatively long period of time, so even if we weren't at the optimal pH, we would have picked up activity regardless. We have tried to troubleshoot finding activity of CAZymes from gut microbes before by testing different pHs, but this has never been successful. CAZymes operating at extreme pHs are usually from fungal or environmental sources. You are correct the activity of CAZymes can be altered with pH, e.g. transglycosylation, but it's not applicable here.

12. Please clarify how partial activity is being assigned.

We will do this.

13. I think the AF2 models should be excluded unless they are used to explain certain substrate preferences. They do not add much to the overall manuscript. Similarly, the glycan models in Supplemental 4 do not significantly add to the manuscript.

We will remove the ones if they are not explicitly mentioned.

14. The discussion section of the manuscript fails to successfully present the key takeaway points from the results. Further expansion on the importance and relevance of key findings needs to be emphasized. Additionally, the introduction of mucinosomes (line 697) is confusing and does not fit in the context of the presented results.

We agree that this can be improved and will take your advice and the other reviewers to produce a better discussion.

Minor:

1. Several typos throughout:

a. Oligosaccharides is misspelled in Supplementary Figure 2,3

b. Line 273 – "Tn-antigen was not a degraded"

c. Line 574 – "continud"

d. Line 575 – "degradatonto"

e. Line 577 – "patter"

f. Line 990 – "(refs)" – missing reference?

g. Figure 4 caption – "...apart from xxx For sulfated..." – not sure what the xxx is referring to

2. Can a representation of colominic acid be added to Supplemental 1E?

3. Line 91 – Disagreement in the proteins mentioned and in supplemental 1 Amuc_0625 is 0623 in supplemental and 1825 is 1835 in supplemental 1.

4. Line 118 – I don't think statements about rates should be included when they were not

pursued further. Also, this line references Supplemental 5, however the correct figure to reference is Supplemental 6.

5. Line 123 – What commercially available HMOs were assayed? Were they a mixture of several different HMOs or was just one representative HMO chosen for each group?

6. Line 142-144 – This also indicates that it prefers 1,2 which is not mentioned.

7. Line 185-188 – Where there any differences in active site architecture that would allow for a BGB glycan to fit into the active sites of the two “active” proteins while not fitting into the “inactive” protein? A superimposition could be performed with the model in Supplemental Figure 17.

8. Line 190 – You state “one” protein while there are two proteins in the GH27 family shown in figure 1. It appears that Amuc_1220 was mislabeled and should be in the GH89 family.

9. Line 201 - Was there an increase in galactose when preincubated with sialidases/fucosidases? It appears in Supplementary Figure 11 that we see less of galactose when preincubated with sialidase/fucosidase. Could the authors comment on this?

10. Line 216 – the heat map has two GH97 enzymes but the text says one.

11. Other peaks are in Figure 1C that are not labeled. Can the authors comment on these unlabeled peaks?

12. Line 311 – are there other substrates to test?

13. Line 322-326 – this section needs revising and is not appropriate for a results section.

14. Line 427-428 – the wording here is awkward.

15. Line 429-433 – Consider restructuring here. Perhaps enzyme activities should be described first.

16. Line 484-484 – what is the one feature that you could speculate on? It is odd to make this statement and then not describe what it is.

17. Line 485 – Why are you not able to explore biochemically?

18. Line 508 – I don’t think this title is appropriate. I think it is more accurate to state that some of the glycans are completely degraded but these data do not “completely” degrade.

19. Line 534 – what are the predicted functions of the c terminal domain?

20. Line 555 – it is odd to have extensively studied this protein and not others.

21. Line 636 – I don’t think this section should be included in this manuscript.

22. Line 710 – this could be because different pHs were not tested.

Reviewer #2 (Remarks to the Author):

Crouch et al. provide a complete picture of colonic mucin glycan degradation by Akkermansia muciniphilla, a symbiont associated with gut health. Beyond extensive characterization of the multiple glycoside hydrolases, sulfatases, and lyases involved in mucin breakdown, the authors demonstrate that these enzymes can cooperatively break down complex mucin glycans to completion and provide evidence to suggest the intra/extracellular localization of these events. The authors also expand their characterization of Akkermansia enzymatic cleavage activities to other polysaccharides of high relevance to the microbiota, such as gangliosides and human milk oligosaccharides. Taken together, the characterization of these enzymes, and associated data concerning the order of enzyme activity and localization, represent a significant advance for the field. We recommend publication of this work provided that the following concerns regarding presentation of the findings can be addressed.

Thank you very much for taking the time to review this work and for you positive comments.

1. Results: α -sialidases

“Interestingly, there was some indication that Amuc_0625GH33 could release Neu5Gc more efficiently than the other enzymes, but this was not explored further. Release of sialic acid also carried out on porcine gastric mucin III (PGM III) assessed by thin layer

chromatography (TLC) indicates that Amuc_0625GH33 and Amuc_1825GH33 are releasing the highest amounts, but this was not quantified (Supplementary Figure 1)”

Taking the first sentence to refer to the chromatography data in Supplementary Figure 6 (box d), it is not apparent that the concentration of Neu5Gc is significantly higher in the Amuc_0625 treated sample compared to the Amuc_1835 sample; as such, it is not clear to us why the authors would make this claim. The subsequent discussion of the relative enzymatic activity as per qualitative TLC staining does not seem significant or supportive to the overall conclusions made by the authors, and we feel this was not presented convincingly in the absence of replicates or more rigorous characterization.

We agree that this is a step too far in the interpretation and have removed this and other statements like it.

2. Results: Alpha-galactosidases

“Amuc_0480GH110 was able to release more galactose than Amuc_1463GH110, however this was not quantified or the reasons for this difference explored further in this study.”

Like the previous example, this point does not seem significant to the overall conclusions made by the authors, and without more analytical rigor or characterization, the statement is poorly supported.

Yes, you are correct, we wanted to show we were thinking about our data, but it is a step too far.

3. “We could not find any activities for Amuc_0623GHnc or Amuc_1547GH177 during this work despite their associations with sialic acid hydrolysis (Supplementary Figure 1).”

A citation should be provided supporting the notion that Amuc_0623 and Amuc_1547 are associated with sialic acid hydrolysis. Additionally, reference 14 in the manuscript appears to address this aspect of Amuc_0623 activity.

Thank you for pointing this out, we have incorporated it into the manuscript.

4. We find that the fundamental understanding of Akkermansia’s mucin degradation capabilities afforded by this work represents an important addition to the field given the established prevalence of this microbe in gut health. Another point of significance for the field is the authors’ highly efficient recapitulation of Akkermansia enzymes to act on specific glycan motifs in whole mucin samples, and to furthermore analyze the glycan composition of these complex molecules.

We have completed this analysis with another liquid chromatography-mass spec technique (discussed in Rev 1, point 2).

These enzyme cocktails could therefore be used to directly study glycan compositions in other contexts, such as gut microbiota dysbiosis and other such “diseased” mucus states.

We agree and are excited about this prospect. We are currently setting up pipeline to produce human colon and stomach organoids that produce mucus continuously. We will be able to collect healthy and disease samples from patients to test out applications soon.

We would, however like to address this quote from the discussion section: “One of the most significant aspects of this report was showing that we could use a cocktail of CAZymes from AM to reach the core GalNAcs of the O-glycan, thus demonstrating complete degradation of the constituent glycan of this complex substrate (Figure 1).”

We feel this aspect of the work, as significant as it was, was not properly highlighted in the

results section. We felt that the wide breadth of enzyme activities characterized, and their relationships to other organisms and substrates other than mucin, received much more written focus. An effort should be made to streamline the writing to highlight the most significant results.

Yes, we agree. As mentioned in Reviewer 1 point 1, we wrote this in the order of degradation (cap to core) and the manuscript needs converting into something more readable to highlight the most significant aspects.

5. Supplemental figure 1: We believe there is a typo with the labeling as “Amuc1835” throughout the figure, where we are led to believe in the corresponding text that “Amuc1825” is the subject of the figure.

Thank you.

6. “An attempt to resolve the kinetic parameters of Amuc_0863GH105 was performed by running the reactions of Amuc_0778PL38 to completion (overnight), then adding Amuc_0863GH105 and following the loss of the double bond of the unsaturated uronic acid at 235nm. While linear decreasing initial rates were observed for the lower substrate concentrations, the higher concentration reactions were not reliable because of the absorbance ceiling of the spectrophotometer. Attempts to downscale or dilute the reactions did not provide data for which a reliable kinetic model could be fitted (Supplementary Figure 40). However, by isolating the highest observed initial rate for each substrate, a clear selectivity of Amuc_0863GH105 was observed towards CSA, followed by the three other GAGs at 36-40% relative activity (Figure 4, Supplementary Table 6).”

That the reaction has reached completion does not follow from it being run overnight, and the authors make no mention of how the substrate concentration was controlled, only that it was measured.

In terms of presentation, this section could be edited to be much clearer and more concise.

7. There are several instances such as the quotes below in the results section where the authors extensively qualify the broader significance of the data to the field or highlight data from other studies. While this is pertinent information that strengthens the impact of the results, we believe

that these statements are better placed in the discussion section, to enable better understanding of the results that the authors themselves generated. Alternatively, the writing can be condensed, with these statements made in direct reference to the data.

Thank you for pulling these out for us, we will use them to re-work our discussion.

α -galactosidases

“The whole cell assay results from the blood group hexasaccharides do show breakdown, but the type of glycobiochemistry was not further explored in this work. However, we previously characterised GH16 activity against these substrates and this activity is prominent on the outside of the cell (15; Supplementary Figure 8).”

“Amuc_1187GH27 can hydrolyse Gal α 1,3Gal/GalNAc and globotriose, but less activity was seen against P1 antigen. Gal α 1,3Gal (Galili antigen) is expressed by most mammals and some other animals, but not humans, apes, and Old World monkeys²⁹. Therefore, gut microbes would only have access to this structure through animal products in the diet. Conversely, the similar Gal α 1,4Gal epitopes of P1 and Pk antigens are generally present in human tissues, so they would be present in the mucosal surface of the human large intestine. P1 and Pk structures

are receptors for a variety of pathogens and their toxins³⁰.”

α -GalNAc’ases

“Forsman antigen is normally present in only 0.01 % of the human population but in that relatively small population this antigen would be present in the GI tract³⁴. It can change susceptibility to some diseases and is also expressed in some cancers, but its true prevalence and impact on different diseases is understudied³⁵. BGA is much more prevalent within the human population with this epitope being expressed in 10-40 % of the population depending on where you sample globally³⁶. We were only able to test BGA type II in this study, but given the broad activity of Amuc_0216GH³⁶, it is likely that it will act on all BGA structures.”

Investigating β -galactosidase activity

“These branching structures, formed through a GlcNAc β 1,6Gal linkage, are important due to their prevalence in humans (99 % of population) and are referred to as “I antigens” (no branching is referred to as “i antigen”). Mucins across most tissues will have this branching¹⁷. Complementary to the screening carried out in this report, Amuc_1666GH² has previously been shown to have a preferences for LacNAc⁴⁴ and Amuc_0771GH³⁵ has also previously been shown to have activity against mucin core 1 and core 2 structures, both β 1,3 linkages⁴⁵”

8. We feel that the following statement lacks proper context and clarity, even when considering the surrounding sentences: “It was generally difficult to make any observations about how the different structures were related to the observed specificities. However, there was one feature where we felt we could speculate, but frustratingly it was untestable in this work.”

This is not worded well, apologies. XXX

9. In the introduction, we suggest the authors speak more to the biological implications of Akkermansia mucin glycan degradation, and the context in which it operates, in more detail. Being able to completely break down the glycans to the backbone with just a cocktail of isolated enzymes is a significant finding in the context of Akkermansia, but why is that glycan breakdown significant in its own right towards health and disease?

Thank you for the advice, we will discuss the relationship between health/disease and mucin degradation in the introduction.

10. “The mucosal surface of the human large intestine is predominantly comprised of gel-forming secreted mucins and approximately 80 % of this glycoprotein is O-glycan.”

What are the authors referring to by “80% of mucin is O-glycan”; by mass?

Reviewer #3 (Remarks to the Author):

The manuscript about the ‘Carbohydrate-active enzymes from Akkermansia muciniphila breakdown mucin O-glycans to completion’ by Cassie R. Bakshani et al. describes an extensive biochemical and structural characterization of the numerous carbohydrate active enzymes, present in the genome of Akkermansia muciniphila, and showing hydrolytic activity on mucin or mucin-related glycan/oligosaccharide structures. Activities of recombinantly produced, pure enzymes were screened by various (complementary) methods such as thin-layer chromatography, colorimetric assays on artificial substrates, or product analyses by anion-exchange chromatography, as best adapted to the nature of the substrate. The study is complemented with physiological growth experiments and pangenomic analyses to highlight the importance of the complete breakdown of this complex substrate present in the human

intestine by *A. muciniphila*. Importantly, the study provides detailed experimental, biochemical proof of hydrolytic activities towards mucin and mucin-related substrates, thus ideally complementing previous studies that have analyzed mucin metabolism physiologically or by transcriptomics, such as those reported by Glover JS *Sci. Rep.* 2022 or Davey LE *Nat Microbiol.* 2023.

A total of fifty-five glycoside hydrolases, twelve sulfatases, and one polysaccharide lyase were characterized, an impressive feat of strength! The results allow to conclude that this important intestinal verrucomicrobiales, *A. muciniphila*, is capable of fully degrading host mucin and similar glycan structures down to the core GalNAc in a synergistic, orchestrated manner. A very important and noteworthy aspect of the work is that the enzymes were tested against a very broad range of substrates, which is not always the case, a fact that helps to pin down particular specificities, exclude others and understand the finetuned orchestration of the breakdown of host mucins or related substrates, the latter being only present through animal products in the diet. These biochemical data are completed by some growth experiments, pangenomic comparisons and structural (alphafold) analyses that help decipher and cautiously underscore potential absence or presence of particular hydrolytic activities.

Overall, the study reports an outstanding wealth of data and results, all of which are novel biochemical characterizations, highlighting that a large and complex set of enzymes is necessary for the efficient uptake and usage of the complex substrate mucin. The approach is perfectly valid and methods are well described and documented. Despite this extremely rich and dense data, supplemental material and information, the manuscript is well structured and comprehensively illustrated so that the reader can follow the overall workflow and seize the important messages. Taken together all these facts, I therefore believe that the manuscript is perfectly adapted and worthwhile publishing in *Nature Microbiology*. I only have spotted some minor issues that might be taken into account and a number of typos, wrong numbers, easily overlooked when describing such an immense load of data. (I am sure that I did not spot all mistakes, though!)

Thank you for your positive words and for your comments, we really appreciate them.

Comments

1. Abstract, last sentence: “how AM degrades mucin O-glycans, related structures and other host carbohydrates”.

The manuscript does report a few results on substrates that can only be taken up through diet, such as starch or the Galili antigen – this does not become entirely clear by this statement (summarized as “host carbohydrates”). I suggest reformulating the sentence to clarify this, even though *Amuc* does not grow on starch for example as sole carbon source.

No problem.

2. Results page 3, lines 118-119: “Interestingly, there was some indication that *Amuc_0625GH33* could release Neu5Gc more efficiently than the other enzymes, but this was not explored further.”

I don’t want to ask to explore this further but the statement is intriguing, begging to know why the authors find this an interesting aspect? Could a short comment be included in additional discussions?

We have deleted this as it was a step too far in the analysis.

3. Page 7 paragraph from 322 to 332. The authors highlight here that one of the *Amuc* GH89 enzymes, *Amuc_0060*, has similarities to the human homolog active against heparan sulfate. They evoke that mutations of this enzyme leads to specific syndromes for which no treatments are available. This statement is a bit “floating in the air” – it is a pity that such an important detail is not picked up in the discussion, since, in addition to the “major impacts of this being the

ability to characterize different mucins in much more detail and in a higher throughput manner” perhaps some of the enzyme characterizations will also help better understand and find ways of tackling human disorders involving homologous enzymes?

4a. Page 10 lines 483-486. The expression is such that the reader has strong empathy with the authors in respect to the frustration – and is frustrated in turn; is this untestable because of inaccessibility of adequate substrates? If so, it should be mentioned. If not, it should be tested.

4b. And following up on the tempted structural speculation of Amuc GH20 enzymes, why not include an indication or outlook to the paragraph lines 497 to 506, opening to the fact that the presence of varying sulfation patterns could be accepted by some of the (redundant) enzymes? Isn't this summarizing paragraph tailored for being picked up in the discussion?

5a. Discussion. Overall, I think the discussion needs to be reworked a bit. First, I would reorganize the construction (order of the paragraphs) to finish on a more positive note than the enzymes for which no activity has been found. Secondly, when reading the discussion, one gets the impression that more is said about the GH16 enzymes (from a previous study) than on the huge number of novel data that has been experimentally dissected in this study. Figure 6 might deserve a bit more description. I think some of the key features should be highlighted here in a more explicit way, rather than in the general summary presented at the end of the first paragraph (that, by the way, could be the last concluding paragraph). Mentioning the difficulties to assess certain specificities because of lack of suitable substrates could be included here, rather than in the results section and be discussed as a prospective.

5b. Discussion, last paragraph about the GH57 enzyme. For non-specialist readers the link between Amuc GTs, exopolysaccharides and GH57, in particular, does not become clear from the text here, nor in the short description in supplementary information, lines 908 to 916. Indeed, several GH57 enzymes have been reported to have transglycosylating activities, for characterized enzymes to date mainly observed on 1,4-alpha-glucose containing substrates, such as starch, amylopectin or malto-oligosaccharides. This transglycosylating activity could indeed be linked to the recycling of taken up sugars to synthesize capsular or exopolysaccharides. This enzyme might be in some way linked to the starch active enzymes? As far as I could see, besides starch, dextran and maltose, no intermediate size (DP3 to 7) alpha-linked glucose sugars were tested for this enzyme, which might be the missing link (or not)? I am not especially asking for additional tests.... but perhaps opening up the discussion about potential missing/additional substrates that could be tested, despite the huge amount already tested.

Typos

1. Page 2 line 56 : “The details of O-glycan structure varies along” should be changed to “The details of O-glycan structure vary along”

2. Page 4 lines 133-134: “..., so these glycan epitopes are first partially broken down by other CAZymes encoded in the AM genome to produce blood group H structures first.”

The sentence contains a repeat of “first” – one of both should be deleted.

3. Page 6 line 273: in “however, contrary to this, Tn-antigen was not a degraded” “a” should be deleted.

4. page 8 line 370-371: "...mucin core 1 and core 2 structures, both β 1,3 linkages"
I feel that this needs to be changed to "which both have β 1,3 linkages" or "both containing β 1,3 linkages". Perhaps also cite supplementary figure 4 here, where these structures are schematically presented.
5. Page 10 line 495 "Supplementary Figure 33". This should be Supplementary Figure 31 (phylogeny of GH20) and not 33 (phylogeny of GH84)
6. Page 12 line 574 change "the disaccharide continued to increase" "the disaccharide continued to increase"
7. Page 12 line 619. "subfamily 11 enzymes Amuc_1033Sulf 16 and Amuc_1074Sulf 16"
I suppose the authors meant Amuc_1033Sulf 11 and Amuc_1074Sulf 11.
8. Page 14 Discussion second line (675). I believe there is an error in one of the enzymes' names. The previous work was on GH16 enzymes but the list also includes Amuc_2136GH20; possibly this should be replaced by Amuc2108GH16
9. Page 18 line 900 – a leftover (refs) should be replaced by the corresponding references; line 902 "produced by some organisms and is two glucose monosaccharides" should be replaced by "produced by some organisms and composed of two glucose monosaccharides"
10. Page 19 Supplementary information line 909: Supplementary Figure 20 is a phylogenetic tree, instead I believe the authors refer to Supplementary Figure 28 which includes the colorimetric assay on pNP- α -Glc.

Dear Reviewers,

Thank you for taking the time to read our first manuscript. We realise now that the way it was written made it a bit of a chore for you, although that was not what we intended. We have re-written it into something we hope is more digestible and some Supplementary Figures from the original manuscript have been pruned. We really appreciated all the advice and suggestions. We have added RNA-seq data of AM on mucin and some liquid chromatography-fluorescence detection-electrospray-mass spectrometry (LC-FLD-ESI-MS) of sequential reactions using the different enzymes and, we think, that this data brings everything into a more cohesive story now. We hope you find this to be true.

All the best

Lucy and Team

Reviewer #1 (Remarks to the Author):

Bakshani et al. report a lot of initial data that may be valuable to the glycobiology field. The manuscript describes the activities of 55 glycoside hydrolases, 12 sulfatases and a polysaccharide lyase from *Akkermansia muciniphila*. This manuscript provides some insights into the mechanisms that AM utilizes to degrade complex glycans from a variety of sources including porcine gastric mucin, human milk oligosaccharides, and plant polysaccharides. While this work attempts to cover a lot of enzymes, the manuscript does not have a cohesive story and good deal of the results presented are the bare minimum it would take to begin an analysis. In many respects, it appears to be 2-3 papers worth of preliminary data put together. It is hard to imagine that it will make a strong impact on the field because it is a survey of initial conclusions one can make about these enzymes. Furthermore, there are significant concerns about the way key assays were performed. Major and minor comments follow. Taken together, this manuscript cannot be accepted in its current form without major additional experiments and a significant reorganization.

Thank you for taking the time to review our work and sharing your valuable insights. We were disappointed that you did not like the work at all, but I am hoping to win you round on a few points.

Major:

1. This manuscript is a challenge to read and needs considerable work to make this more approachable for readers. In its current state it reads like a statement of observations and needs significant changes to correct this.

Yes, we agree with you. We wrote it in the order of degradation (capping to core) and hopefully to reduce the amount of flicking back and forth between sections a reviewer would have to do. We realise that this was a mistake now and the main text is quite different. Hopefully, you will find this version more accessible.

The manuscript reads like it was meant to be separated into 2 or more manuscripts. I am not sure the starch degrading enzymes are necessary in this manuscript or the high mannose N-glycoproteins sections.

We set out to cover all AM CAZymes and that has naturally highlighted several puzzles and new avenues where full characterisation is beyond the scope of this manuscript. I can see how this work could be cut up into several papers suitable for other journals, but we wanted one all-encompassing reference with a focus on host glycans. We stand by our ambitious vision for a holistic and systematic approach. We strongly believe that all components of this work contribute to the understanding of the biology of *Akkermansia muciniphila*.

In terms of the “starch enzymes”, I think it’s important to highlight that we have screened them against the obvious substrates and not leave them out. In terms of the high-mannose N-glycan observation, we believe this is of significance due to the abundance of these in the gut, both from host and dietary sources. Mucins and secreted glycoproteins in the gut (e.g. antibodies) are decorated with these glycan modifications, so how different host-derived nutrients are accessed and cross fed between species will be of interest to the field.

2. I have concerns with the way the whole cell assays were performed. Several instances throughout the manuscript these whole cell assays do not degrade relatively simple sugars. Do the authors find this concerning? Clarification and commentary are essential.

I am not concerned with a lack of activity against some substrates, and we were not expecting that the surface-localised enzymes would be able to tackle the full panel of substrates that we have. We are not sure why you are concerned. With the whole cell assay work, we wanted to understand what capabilities are on the surface as the first keystone activities. AM likely does not want all activities occurring on the surface so it can keep nutrients for itself.

Since we first submitted, we have now been able to look at the glycans released from mucin during whole cell assays in greater detail. We labelled the reducing ends with a fluorescent label and used liquid chromatography-fluorescence detection-electrospray ionization-mass spectrometry (LC-FLD-ESI-MS). We determined the composition of the different glycans using the MS/MS data. The data shows that most of the O-glycans have a galactose at the reducing end, which are GH16 endo-O-glycanase products. This data provides a more detailed analysis of AM surface activities when presented with a complex substrate. See Figure 1.

3. Cell lysate experiments should be conducted against all substrates, not just whole cell assays with substrates. This is especially important since so many whole cell experiments showed no activity.

You don’t state a reason for doing this, is this purely to look at AM capability? I am happy to do this if required, but I don’t think it would add to the story. There’s quite a bit of redundancy in the activity of the AM enzymes against the panel of defined substrate we have, so I am unsure what would be gained unless we would be just checking what AM can breakdown? The only application I think would be interesting to do this for 6’S-Lewis X, where we couldn’t find a way to degrade it, to see if AM can do this, but it wouldn’t highlight the enzyme responsible. If we did whole cell lysate assays with mucin, I think there would be a lot of activities happening and it would be difficult untangle what was happening or draw any conclusions.

4. Several figures need reworking to make them easier to view. The TLC images should be scaled similarly so the lanes align in many supplemental figures. Additional annotation of TLC images, like labeling specific lanes, would improve clarity.

We have endeavoured to enhance the presentation of the data throughout.

5. The pNP assays should be conducted again and absorbance measurements should be taken rather than images. In no way are photos of a color change acceptable as data on enzyme assays. Color changes can be measured precisely and should be, rather than a yes/no report that barely passes the bar as an initial experiment one might show at lab meeting. Additionally, this would allow for rates to be reported, which should be.

Kinetics on pNP-sugars are meaningless and those numbers wouldn’t tell you anything about how the enzyme behaves with its true substrate. We only use pNPs to identify the -1 sugar and as you can see from our photos, we are only getting one hit for some of the enzymes. I realise other people report pNP yes/no without showing any evidence, so I have removed the figure and

just commented in the text. The aim was to show that for the 'puzzle' enzymes we had tried to cover all the bases.

6. Why are Michaelis-Menten kinetics performed for Amuc_0778 and not others? This seems odd/out of place. As enzyme scientists know, Michaelis-Menten kinetics are critical for a complete and appropriate understanding of enzyme function.

We were able to perform kinetics for those enzymes active on GAGs as these substrates are available in large quantities and are relatively cheap. The major obstacle we have with kinetics on O-glycan-active enzymes is the cost - even if we did this by HPAEC the cost would run into the tens of thousands. For context, we spent ~£20k on substrates alone for this work and that was just to top-up what we already had in our substantial collection. Ideally, we would like enough substrates to carry out assays on the spectrophotometer, which would be much easier and more accurate, but that is not economically feasible as this would require even more substrate than HPAEC.

The other issue with HPAEC and O-glycans is that they aren't detected as well as other glycans (monosaccharides are fine, but we are finding O-glycan type structures require a relatively high concentration). In contrast, for example, N-glycans are detected much more easily using this technique. Labelling to increase detection: labelling substrates with a fluorophore to enhance detection prior to an assay could affect enzyme activity and labelling post-reaction introduces a lot of error/loss with the clean-up processes. This also costs more time.

In summary, we think that for this paper investing that amount of money and time (which we do not have) would not be proportional to what further detail we may elucidate. Ultimately, the enzymology we have been able to complete highlighted a wide selection of different specificities, and therefore new tools, for the breakdown of O-glycans and other host glycans. This is everything we can possibly do with our current resources and is not a reflection of our abilities.

7. It is not clear that GalNAc is being released in Figure 1. Why is a more thorough experiment not conducted for PGMIII like what was performed for BSM (Suppl Fig 6)? This is one of the most important findings in the study but the control lane for GalNAc is the furthest from the experimental lane.

We have re-ordered the chromatograms.

The great aspect of BSM is that it only has one O-glycan structure, so we know exactly what our substrate is. PGM is much more heterogeneous so you wouldn't be able to conclude much about the specificity of different enzymes if just detecting monosaccharide release. I'm not sure what an assay against PGM, like the BSM one, would be aiming for, if you wouldn't mind expanding? However, I do have some alternative new data that you may find satisfactory in terms of wanting more detail - see point 8.

Additionally, in figure 1D the first two lanes are quite messy and not much can be concluded from them.

These lanes are to demonstrate that the enzyme cocktail releases a large amount of monosaccharides. They are to show the process we put the PGM through before testing for the core GalNAc with 1008. These are difficult to separate by TLC, however, galactose and GlcNAc stain blue and brown, respectively, and the GlcNAc runs slightly faster. You can see these on the TLC. We have altered the text to incorporate a better explanation.

8. It is mentioned that fucosidases cannot degrade intact mucin oligosaccharides.

I think you are referring to the sentences starting 139 in the first manuscript. This is poorly-worded, apologies. We added the fucosidases individually to PGMIII and we could only see fucose being released by 1120. This means that for this intact mucin substrate, only 1120 can access the capping fucose, but this may vary for other types of mucin. We did not mean to imply that this was the only fucose that could ever be removed from PGMIII (see the next point). It has been re-worded (Line 184 onwards).

I suggest the authors conduct additional experiments where PGMIII is incubated with a panel of other enzymes and followed up with these fucosidases, like what is done in Suppl Fig 6. I think that incubating GH16 enzymes prior would be especially useful.

As your comments came back to us, we also got some data back from our industrial collaborators analysing the glycans produced in sequential reactions with the AM CAZymes from PGMIII, as you have suggested. We completed a variety of assays with the panels of different enzymes and stopped the reactions in between steps to ensure they were sequential. These were then labelled with procainamide and analysed by LC-FLD-ESI-MS as described above. The fucosidase data on GH16 fragments is now in Figure 2. The data shows that four of the fucosidases and three of the enzymes removing α -capping monosaccharides (1220, 0216, and 0517) show clear activities against different O-glycan fragments (red asterisks indicate activity). We were also able to do this for the β -galactosidases (Figure 3) and β -HexNAc'ases (Figure 4). We have integrated the results of these experiments into the rest of the results throughout the new manuscript.

9. Several instances throughout the manuscript the author claims that the experiments conducted suggested one enzyme being more efficient than another and that it was not pursued further. I think that such statements should not be included without data to back up these claims. These repeated statements significantly contribute to the sense that this is an initial report that adds little to the field without the further, essential follow-up work.

You are correct, we have removed these.

Furthermore, adequate quantification of enzyme activity using the pNP assays or additional HPAEC-PAD assays would greatly increase the value of Figure 6 for the field, as accurate comparisons could be made between enzymes at each step of mucus degradation.

We hope you find that the sequential reactions analysed by LC-FLD-ESI-MS analysis fills this requirement.

10. Several growth curves look abnormal and should be performed again. It is also unclear what the multiple lines on each graph represent.

Data is not always perfect and it's not unusual (but relatively rare) to have one growth behaving slightly differently to the rest. We have taken the anomalies out, although we think this is generally bad practice.

Each line is a growth in a 96-well plate, and we will make this clearer in the text. From my experience, these growth curves look very consistent for being separate growths. If you could point out which ones you are unhappy with, I can repeat them.

11. Do you think some enzymes that had low activity might have low activity due to a different pH optimum? This may be particularly important for sequences that lack a signal peptide and are not trafficked outside the cell. It is essential that pH screens are performed as it is common knowledge that enzymes have pH optima that change their activities. Without this information, little can be learned from enzymes with no activity at a single pH.

For this microbe, in this environment, the optima are highly likely to be close to neutral. It's highly unlikely we missed activities based on a pH issue as these were relatively long assays completed multiple times on multiple substrates.

We have previously tried to troubleshoot finding activity of CAZymes from gut microbes before by testing different pHs, but this has never been successful (we recently tried this with *Bacteroides* and *Bifidobacterium* enzymes with no positive results, for example). CAZymes operating naturally only at extreme pHs are usually from fungal or environmental sources. You are correct the activity of CAZymes can be altered with pH, e.g. transglycosylation, but it's not applicable here.

12. Please clarify how partial activity is being assigned.

We have added a key to each heat map and an explanation in the Figure 3 legend.

13. I think the AF2 models should be excluded unless they are used to explain certain substrate preferences. They do not add much to the overall manuscript. Similarly, the glycan models in Supplemental 4 do not significantly add to the manuscript.

Some of these have been removed.

14. The discussion section of the manuscript fails to successfully present the key takeaway points from the results. Further expansion on the importance and relevance of key findings needs to be emphasized. Additionally, the introduction of mucinosomes (line 697) is confusing and does not fit in the context of the presented results.

We agree and the discussion has now been re-written. We did remove some information and wider-context points and this included the mucinisomes section.

Minor:

1. Several typos throughout: Thanks for these.

a. Oligosaccharides is misspelled in Supplementary Figure 2,3 Thank you

b. Line 273 – “Tn-antigen was not a degraded” Thank you

c. Line 574 – “continud” Thank you

d. Line 575 – “degradatonto” Thank you

e. Line 577 – “patter” Thank you

f. Line 990 – “(refs)” – missing reference? Added

g. Figure 4 caption – “...apart from xxx For sulfated...” – not sure what the xxx is referring to Deleted

2. Can a representation of colominic acid be added to Supplemental 1E? Done

3. Line 91 – Disagreement in the proteins mentioned and in supplemental 1 Amuc_0625 is 0623 in supplemental and 1825 is 1835 in supplemental 1. Thank you

4. Line 118 – I don't think statements about rates should be included when they were not pursued further. Removed.

Also, this line references Supplemental 5, however the correct figure to reference is Supplemental 6. Thank you

5. Line 123 – What commercially available HMOs were assayed? Were they a mixture of several different HMOs or was just one representative HMO chosen for each group? They are the sialylated and neutral HMOs from Biosynth as detailed in Supplementary Table 2 and link below. We could never get any sialic acid off either type with the AM sialidases (and other broad-acting sialidases). I did ask Biosynth about this, but they did not reply to my emails. I am assuming that pools HMOs are purified in some way and then split into two elutions/fractions. From our TLCs, I assume that this will be to do with size. Furthermore, we can breakdown all the HMO spots, so I think there is virtually no sialic acid present in these products (TLC data). We are working on getting our own supply of human milk established, but the ethics takes a while.

<https://www.biosynth.com/p/OH165970/human-milk-sialylated-oligosaccharides>

6. Line 142-144 – This also indicates that it prefers 1,2 which is not mentioned. I'm not sure what you mean here. We already know that both enzymes prefer 1,2 from the defined substrate screen. I have included a more detailed fucosidase description in the Supplementary Discussion.

7. Line 185-188 – Where there any differences in active site architecture that would allow for a BGB glycan to fit into the active sites of the two “active” proteins while not fitting into the “inactive” protein? A superimposition could be performed with the model in Supplemental Figure 17. Since first submitting this manuscript, a structure of this enzymes has now been released, so we have cut analysis of models on this topic from this paper.

8. Line 190 – You state “one” protein while there are two proteins in the GH27 family shown in figure 1. It appears that Amuc_1220 was mislabeled and should be in the GH89 family. Thank you, this was an error.

9. Line 201 - Was there an increase in galactose when preincubated with sialidases/fucosidases? It appears in Supplementary Figure 11 that we see less of galactose when preincubated with sialidase/fucosidase. Could the authors comment on this? The amount of galactose released is actually very low, so it is difficult to answer this question. I think other types of mucin with more α -galactose would be able to answer this question. The with-and-without sialidase and fucosidase were different HPAEC runs, so we would have to do a direct comparison multiple times to answer that question. My current opinion would be no or very little difference with the pre-treated sample and the not pre-treated. We were probing for a very obvious difference, which we did not see. Based on our LC-FLD-ESI-MS data, if we looked at α -fucose/ β -galactose release after α -fucose/ α -sialic acid/ α -GalNAc/ α -GlcNAc pre-treatment we would see an obvious difference. These are the types of methods that can be applied to analyse different mucins now.

10. Line 216 – the heat map has two GH97 enzymes but the text says one. Thank you, this was an error.

11. Other peaks are in Figure 1C that are not labeled. Can the authors comment on these unlabeled peaks? PGMIII is quite an unpure product I believe, Sigma will not give details about its production, so I think it will be to do with that. We know that PGMIII does contain some GAGs, but this is probably quite hard to separate well (Supp Fig 42c). Those peaks are in the

controls too. We see 'extra' peaks in the LC-FLD-ESI-MS data also (same time points for different samples) you may have noticed, and the mass spec is not glycan.

12. Line 311 – are there other substrates to test? I am not sure what you are getting at here.

13. Line 322-326 – this section needs revising and is not appropriate for a results section. This has been put in the Supplementary Discussion.

14. Line 427-428 – the wording here is awkward. Changed.

15. Line 429-433 – Consider restructuring here. Perhaps enzyme activities should be described first. Changed.

16. Line 484-484 – what is the one feature that you could speculate on? It is odd to make this statement and then not describe what it is. Sorry, this is poorly worded. This section has been deleted now to cut the different aspects down.

17. Line 485 – Why are you not able to explore biochemically? We didn't have any substrates with a sulfated GlcNAc, but this section has been removed now.

18. Line 508 – I don't think this title is appropriate. I think it is more accurate to state that some of the glycans are completely degraded but these data do not "completely" degrade. This has been changed.

19. Line 534 – what are the predicted functions of the c terminal domain? One of these domains is classed as a CBM32 and this has now been commented on in the text with reference to the characterised GH31-CBM32 from *Clostridium perfringens*.

20. Line 555 – it is odd to have extensively studied this protein and not others. Explained above.

21. Line 636 – I don't think this section should be included in this manuscript. We have left it in for now, see if you find this acceptable.

22. Line 710 – this could be because different pHs were not tested. See above.

Reviewer #2 (Remarks to the Author):

Crouch et al. provide a complete picture of colonic mucin glycan degradation by Akkermansia muciniphilla, a symbiont associated with gut health. Beyond extensive characterization of the multiple glycoside hydrolases, sulfatases, and lyases involved in mucin breakdown, the authors demonstrate that these enzymes can cooperatively break down complex mucin glycans to completion and provide evidence to suggest the intra/extracellular localization of these events. The authors also expand their characterization of Akkermansia enzymatic cleavage activities to other polysaccharides of high relevance to the microbiota, such as gangliosides and human milk oligosaccharides. Taken together, the characterization of these enzymes, and associated data concerning the order of enzyme activity and localization, represent a significant advance for the field. We recommend publication of this work provided that the following concerns regarding presentation of the findings can be addressed.

Thank you very much for taking the time to review this work and for your positive comments.

1. Results: α -sialidases

“Interestingly, there was some indication that Amuc_0625GH33 could release Neu5Gc more efficiently than the other enzymes, but this was not explored further. Release of sialic acid also carried out on porcine gastric mucin III (PGM III) assessed by thin layer chromatography (TLC) indicates that Amuc_0625GH33 and Amuc_1825GH33 are releasing the highest amounts, but this was not quantified (Supplementary Figure 1)”

Taking the first sentence to refer to the chromatography data in Supplementary Figure 6 (box d), it is not apparent that the concentration of Neu5Gc is significantly higher in the Amuc_0625 treated sample compared to the Amuc_1835 sample; as such, it is not clear to us why the authors would make this claim. The subsequent discussion of the relative enzymatic activity as per qualitative TLC staining does not seem significant or supportive to the overall conclusions made by the authors, and we feel this was not presented convincingly in the absence of replicates or more rigorous characterization.

We agree that this is a step too far in the interpretation and have removed this and other statements like it.

2. Results: Alpha-galactosidases

“Amuc_0480GH110 was able to release more galactose than Amuc_1463GH110, however this was not quantified or the reasons for this difference explored further in this study.”

Like the previous example, this point does not seem significant to the overall conclusions made by the authors, and without more analytical rigor or characterization, the statement is poorly supported.

Yes, you are correct, we wanted to show we were thinking about our data, but it is a step too far.

3. “We could not find any activities for Amuc_0623GHnc or Amuc_1547GH177 during this work despite their associations with sialic acid hydrolysis (Supplementary Figure 1).”

A citation should be provided supporting the notion that Amuc_0623 and Amuc_1547 are associated with sialic acid hydrolysis. Additionally, reference 14 in the manuscript appears to address this aspect of Amuc_0623 activity.

Thank you for pointing this out, we have incorporated it into the manuscript and added the references.

4. We find that the fundamental understanding of Akkermansia’s mucin degradation capabilities afforded by this work represents an important addition to the field given the established prevalence of this microbe in gut health.

Thank you.

Another point of significance for the field is the authors’ highly efficient recapitulation of Akkermansia enzymes to act on specific glycan motifs in whole mucin samples, and to furthermore analyze the glycan composition of these complex molecules.

We have completed this analysis of whole cell assays on mucin using LC-FLD-ESI-MS to provide further details (Figure 1, discussed in Rev 1, point 2).

These enzyme cocktails could therefore be used to directly study glycan compositions in other contexts, such as gut microbiota dysbiosis and other such “diseased” mucus states.

We agree and are excited about this prospect. We are currently setting up pipeline to produce human colon and stomach organoids that produce mucus continuously. We will be able to collect healthy and disease samples from patients to test out applications like the ones you suggest in the future.

We would, however like to address this quote from the discussion section: “One of the most significant aspects of this report was showing that we could use a cocktail of CAZymes from AM to reach the core GalNAcs of the O-glycan, thus demonstrating complete degradation of the constituent glycan of this complex substrate (Figure 1).”

We feel this aspect of the work, as significant as it was, was not properly highlighted in the results section. We felt that the wide breadth of enzyme activities characterized, and their relationships to other organisms and substrates other than mucin, received much more written focus. An effort should be made to streamline the writing to highlight the most significant results.

Yes, we agree. As mentioned in Reviewer 1 point 1, we wrote this manuscript to match the order of degradation (cap to core) and, in retrospect, this was probably a mistake. The manuscript has been altered considerably now. To really highlight that these enzymes can get down to the core GalNAc, we have described this experiment in the second section and before the more detailed description of the enzymes. Hopefully, you will find this version much more streamlined.

5. Supplemental figure 1: We believe there is a typo with the labeling as “Amuc1835” throughout the figure, where we are led to believe in the corresponding text that “Amuc1825” is the subject of the figure.

Thank you.

6. “An attempt to resolve the kinetic parameters of Amuc_0863GH105 was performed by running the reactions of Amuc_0778PL38 to completion (overnight), then adding Amuc_0863GH105 and following the loss of the double bond of the unsaturated uronic acid at 235nm. While linear decreasing initial rates were observed for the lower substrate concentrations, the higher concentration reactions were not reliable because of the absorbance ceiling of the spectrophotometer. Attempts to downscale or dilute the reactions did not provide data for which a reliable kinetic model could be fitted (Supplementary Figure 40). However, by isolating the highest observed initial rate for each substrate, a clear selectivity of Amuc_0863GH105 was observed towards CSA, followed by the three other GAGs at 36-40% relative activity (Figure 4, Supplementary Table 6).”

That the reaction has reached completion does not follow from it being run overnight, and the authors make no mention of how the substrate concentration was controlled, only that it was measured.

We have edited this to provide a better description. We do have some high-performance size-exclusion chromatography (HP-SEC) data suggesting that the reactions have stopped after two hours and that for some of the substrates (like HA) it means complete degradation of the large polymeric fraction, but we feel that the paper is already very figure-heavy so we did not include it.

In the method section we have described how we calculate the concentration in uM of lyase product (unsaturated ends) by using Lambert-Beer. This is the substrate for the GH105. By using the same kinetic reactions from the PL38 we achieve a series of increasing and measurable substrate concentrations for the GH105 applicable for kinetic analysis.

In terms of presentation, this section could be edited to be much clearer and more concise. The data in figures has now been re-arranged and the text edited.

7. There are several instances such as the quotes below in the results section where the authors

extensively qualify the broader significance of the data to the field or highlight data from other studies. While this is pertinent information that strengthens the impact of the results, we believe

that these statements are better placed in the discussion section, to enable better understanding of the results that the authors themselves generated. Alternatively, the writing can be condensed, with these statements made in direct reference to the data.

Thank you for pulling these out for us, we have used them to re-work our discussion.

α -galactosidases

“The whole cell assay results from the blood group hexasaccharides do show breakdown, but the type of glycobiochemistry was not further explored in this work. However, we previously characterised GH16 activity against these substrates and this activity is prominent on the outside of the cell (15; Supplementary Figure 8).”

“Amuc_1187GH27 can hydrolyse Gal α 1,3Gal/GalNAc and globotriose, but less activity was seen against P1 antigen. Gal α 1,3Gal (Galili antigen) is expressed by most mammals and some other animals, but not humans, apes, and Old World monkeys²⁹. Therefore, gut microbes would only have access to this structure through animal products in the diet. Conversely, the similar Gal α 1,4Gal epitopes of P1 and Pk antigens are generally present in human tissues, so they would be present in the mucosal surface of the human large intestine. P1 and Pk structures are receptors for a variety of pathogens and their toxins³⁰.”

α -GalNAc'ases

“Forsman antigen is normally present in only 0.01 % of the human population but in that relatively small population this antigen would be present in the GI tract³⁴. It can change susceptibility to some diseases and is also expressed in some cancers, but its true prevalence and impact on different diseases is understudied³⁵. BGA is much more prevalent within the human population with this epitope being expressed in 10-40 % of the population depending on where you sample globally³⁶. We were only able to test BGA type II in this study, but given the broad activity of Amuc_0216GH36, it is likely that it will act on all BGA structures.”

Investigating β -galactosidase activity

“These branching structures, formed through a GlcNAc β 1,6Gal linkage, are important due to their prevalence in humans (99 % of population) and are referred to as “I antigens” (no branching is referred to as “i antigen”). Mucins across most tissues will have this branching¹⁷. Complementary to the screening carried out in this report, Amuc_1666GH2 has previously been shown to have a preference for LacNAc⁴⁴ and Amuc_0771GH35 has also previously been shown to have activity against mucin core 1 and core 2 structures, both β 1,3 linkages⁴⁵”

8. We feel that the following statement lacks proper context and clarity, even when considering the surrounding sentences: “It was generally difficult to make any observations about how the different structures were related to the observed specificities. However, there was one feature where we felt we could speculate, but frustratingly it was untestable in this work.”

This is not worded well, apologies. We have deleted this section now.

9. In the introduction, we suggest the authors speak more to the biological implications of Akkermansia mucin glycan degradation, and the context in which it operates, in more detail. Being able to completely break down the glycans to the backbone with just a cocktail of isolated enzymes is a significant finding in the context of Akkermansia, but why is that glycan breakdown significant in its own right towards health and disease?

Thank you for the advice. We have re-written the intro and discussion to cover the wider context and make the impact more understandable. Please let us know what you think.

10. “The mucosal surface of the human large intestine is predominantly comprised of gel-forming secreted mucins and approximately 80 % of this glycoprotein is O-glycan.”

What are the authors referring to by “80% of mucin is O-glycan”; by mass?

Yes, we have altered the text, thank you.

Reviewer #3 (Remarks to the Author):

The manuscript about the ‘Carbohydrate-active enzymes from *Akkermansia muciniphila* breakdown mucin O-glycans to completion’ by Cassie R. Bakshani et al. describes an extensive biochemical and structural characterization of the numerous carbohydrate active enzymes, present in the genome of *Akkermansia muciniphila*, and showing hydrolytic activity on mucin or mucin-related glycan/oligosaccharide structures. Activities of recombinantly produced, pure enzymes were screened by various (complementary) methods such as thin-layer chromatography, colorimetric assays on artificial substrates, or product analyses by anion-exchange chromatography, as best adapted to the nature of the substrate. The study is complemented with physiological growth experiments and pangenomic analyses to highlight the importance of the complete breakdown of this complex substrate present in the human intestine by *A. muciniphila*. Importantly, the study provides detailed experimental, biochemical proof of hydrolytic activities towards mucin and mucin-related substrates, thus ideally complementing previous studies that have analyzed mucin metabolism physiologically or by transcriptomics, such as those reported by Glover JS Sci. Rep. 2022 or Davey LE Nat Microbiol. 2023.

A total of fifty-five glycoside hydrolases, twelve sulfatases, and one polysaccharide lyase were characterized, an impressive feat of strength! The results allow to conclude that this important intestinal verrucomicrobiales, *A. muciniphila*, is capable of fully degrading host mucin and similar glycan structures down to the core GalNAc in a synergistic, orchestrated manner. A very important and noteworthy aspect of the work is that the enzymes were tested against a very broad range of substrates, which is not always the case, a fact that helps to pin down particular specificities, exclude others and understand the finetuned orchestration of the breakdown of host mucins or related substrates, the latter being only present through animal products in the diet. These biochemical data are completed by some growth experiments, pangenomic comparisons and structural (alphafold) analyses that help decipher and cautiously underscore potential absence or presence of particular hydrolytic activities.

Overall, the study reports an outstanding wealth of data and results, all of which are novel biochemical characterizations, highlighting that a large and complex set of enzymes is necessary for the efficient uptake and usage of the complex substrate mucin. The approach is perfectly valid and methods are well described and documented. Despite this extremely rich and dense data, supplemental material and information, the manuscript is well structured and comprehensively illustrated so that the reader can follow the overall workflow and seize the important messages. Taken together all these facts, I therefore believe that the manuscript is perfectly adapted and worthwhile publishing in *Nature Microbiology*. I only have spotted some minor issues that might be taken into account and a number of typos, wrong numbers, easily overlooked when describing such an immense load of data. (I am sure that I did not spot all mistakes, though!)

Thank you for your positive words and for your comments, we really appreciate them.

Comments

1. Abstract, last sentence: “how AM degrades mucin O-glycans, related structures and other host carbohydrates”.

The manuscript does report a few results on substrates that can only be taken up through diet, such as starch or the Galili antigen – this does not become entirely clear by this statement (summarized as “host carbohydrates”). I suggest reformulating the sentence to clarify this, even though Amuc does not grow on starch for example as sole carbon source.

No problem, we have altered this sentence

2. Results page 3, lines 118-119: “Interestingly, there was some indication that Amuc_0625GH33 could release Neu5Gc more efficiently than the other enzymes, but this was not explored further.”

I don't want to ask to explore this further but the statement is intriguing, begging to know why the authors find this an interesting aspect? Could a short comment be included in additional discussions?

We have deleted this as it was a step too far in the analysis.

3. Page 7 paragraph from 322 to 332. The authors highlight here that one of the Amuc GH89 enzymes, Amuc_0060, has similarities to the human homolog active against heparan sulfate. They evoke that mutations of this enzyme leads to specific syndromes for which no treatments are available. This statement is a bit “floating in the air” – it is a pity that such an important detail is not picked up in the discussion, since, in addition to the “major impacts of this being the ability to characterize different mucins in much more detail and in a higher throughput manner” perhaps some of the enzyme characterizations will also help better understand and find ways of tackling human disorders involving homologous enzymes?

Thanks for pointing this out. We have actually moved this discussion to the Supplementary.

You have caught on to exactly what I wanted to say without saying directly as I think it is a step too far. There are over a hundred human congenital disorders associated with glycosylation that lead to disease (identified to date). It is possible that characterisation of enzymes where their absence leads to disease will help to identify remedies. We have added another sentence to pin the sentence down a bit more:

“Detailed characterisations of glycobiology (or absence of) leading to disease may highlight potential ideas for treatments in the future.”

4a. Page 10 lines 483-486. The expression is such that the reader has strong empathy with the authors in respect to the frustration – and is frustrated in turn; is this untestable because of inaccessibility of adequate substrates? If so, it should be mentioned. If not, it should be tested.

We have actually deleted this bit completely as we could not test this.

4b. And following up on the tempted structural speculation of Amuc GH20 enzymes, why not include an indication or outlook to the paragraph lines 497 to 506, opening to the fact that the presence of varying sulfation patterns could be accepted by some of the (redundant) enzymes? Isn't this summarizing paragraph tailored for being picked up in the discussion?

Thank you for this point. We could explore accommodation of sulfation in one respect with the LC-FLD-ESI-MS experiments. Figure 4 now shows those enzymes that can accommodate sulfated Gal in the +1 position, for example.

5a. Discussion. Overall, I think the discussion needs to be reworked a bit. First, I would reorganize the construction (order of the paragraphs) to finish on a more positive note than the enzymes for which no activity has been found. Secondly, when reading the discussion, one gets the impression that more is said about the GH16 enzymes (from a previous study) than on the

huge number of novel data that has been experimentally dissected in this study. Figure 6 might deserve a bit more description. I think some of the key features should be highlighted here in a more explicit way, rather than in the general summary presented at the end of the first paragraph (that, by the way, could be the last concluding paragraph). Mentioning the difficulties to assess certain specificities because of lack of suitable substrates could be included here, rather than in the results section and be discussed as a prospective.

Thank you for the advice. We have re-written the discussion.

5b. Discussion, last paragraph about the GH57 enzyme. For non-specialist readers the link between Amuc GTs, exopolysaccharides and GH57, in particular, does not become clear from the text here, nor in the short description in supplementary information, lines 908 to 916. Indeed, several GH57 enzymes have been reported to have transglycosylating activities, for characterized enzymes to date mainly observed on 1,4-alpha-glucose containing substrates, such as starch, amylopectin or malto-oligosaccharides. This transglycosylating activity could indeed be linked to the recycling of taken up sugars to synthesize capsular or exopolysaccharides. This enzyme might be in some way linked to the starch active enzymes? As far as I could see, besides starch, dextran and maltose, no intermediate size (DP3 to 7) alpha-linked glucose sugars were tested for this enzyme, which might be the missing link (or not)? I am not especially asking for additional tests.... but perhaps opening up the discussion about potential missing/additional substrates that could be tested, despite the huge amount already tested.

Yes, I think you are correct that the GH57 should also be explored against these in combination with the other starch-link GH families. We did try this GH57 against maltose, dextran, starch, laminarin, and isomaltose with no positive hits like you said (Supplementary Figure 13).

We have rearranged the text now so hopefully the description of the puzzle enzymes is easier to follow with the CPS/EPS idea.

Typos

1. Page 2 line 56 : “The details of O-glycan structure varies along” should be changed to “The details of O-glycan structure vary along”

Thank you.

2. Page 4 lines 133-134: “..., so these glycan epitopes are first partially broken down by other CAZymes encoded in the AM genome to produce blood group H structures first.”

The sentence contains a repeat of “first” – one of both should be deleted.

Thank you.

3. Page 6 line 273: in “however, contrary to this, Tn-antigen was not a degraded” “a” should be deleted.

Thank you.

4. page 8 line 370-371: “...mucin core 1 and core 2 structures, both β 1,3 linkages”

I feel that this needs to be changed to “which both have β 1,3 linkages” or “both containing β 1,3 linkages”. Perhaps also cite supplementary figure 4 here, where these structures are schematically presented.

Thank you.

5. Page 10 line 495 “Supplementary Figure 33”. This should be Supplementary Figure 31 (phylogeny of GH20) and not 33 (phylogeny of GH84)

Thank you.

6. Page 12 line 574 change “the disaccharide continued to increase” “the disaccharide continued to increase”

Thank you.

7. Page 12 line 619. “subfamily 11 enzymes Amuc_1033Sulf 16 and Amuc_1074Sulf 16”

I suppose the authors meant Amuc_1033Sulf 11 and Amuc_1074Sulf 11.

Thank you.

8. Page 14 Discussion second line (675). I believe there is an error in one of the enzymes' names. The previous work was on GH16 enzymes but the list also includes Amuc_2136GH20; possibly this should be replaced by Amuc2108GH16

This has changed now.

9. Page 18 line 900 – a leftover (refs) should be replaced by the corresponding references; line 902 “produced by some organisms and is two glucose monosaccharides” should be replaced by “produced by some organisms and composed of two glucose monosaccharides”

Thank you.

10. Page 19 Supplementary information line 909: Supplementary Figure 20 is a phylogenetic tree, instead I believe the authors refer to Supplementary Figure 28 which includes the colorimetric assay on pNP- α -Glc.

Thank you.

Dear Reviewers,

Thank you for taking the time to read our manuscript again and for your positive comments. We really appreciate them. Below are responses to the most recent comments.

All the best

Lucy and Team

Reviewer #1 (Remarks to the Author):

Several changes to the original manuscript have improved quality and clarity. The addition of LC-FLD-ESI-MS experiments added to the rigor of the assays. Additionally, several figures are much easier to interpret and many figures have been adequately condensed. Changes to the organization and writing of the manuscript have made it much easier to read.

Thank you for taking the time to look at this again and for your positive comments.

However, the discussion is still lacking in its attempt to summarize all important results, and explain the significance of the breadth of this work.

We have altered the discussion again.

Similarly, several fundamental concerns were not adequately addressed in the revisions. To fully understand how AM interacts with even simple substrates, cell lysate experiments should be conducted. Additionally, pH screens and quantitative pNP assays are critical to completely understanding enzyme function and their importance should not be overlooked.

Thank you for these suggestions. We agree that in some cases further biochemical characterisation of the enzymes will be required in the context of other studies and will be a part of our future work.

Overall, this manuscript was improved upon in several ways, and presents important findings to the glycobiology field. Yet, these findings could be more impactful with a more thorough analysis of AM's capabilities.

Reviewer #2 (Remarks to the Author):

The authors have implemented all points of criticism and thereby substantially improved the manuscript.

Thank you.

Reviewer #3 (Remarks to the Author):

The authors have substantially rewritten and reorganized their results and discussions to respond to reviewers comments. They also have included some relevant additional experiments, clarifying some of the raised issues. In my opinion the authors have done very well and have largely enhanced the quality of the manuscript. It is now suitable for publication. I have only found two minor typos that could be corrected :

Lines 85-86: was performed using "with" liquid chromatography-fluorescence (delete "with")

Line 240: and a GH16 a range "of" sizes and compositions of O-glycan (include "of" in this sentence)

Thank you for highlighting these.

Reviewer #4 (Remarks to the Author):

This study presents a systematic biochemical examination of the predicted carbohydrate active enzymes in *Akkermensia muciniphila*. These enzymes were all tested for activity on mucin or related glycans. I believe the dataset will be extremely useful reference for those interested in the biology of *Akkermensia* and for folks in the glycobiology field.

Thank you for taking the time to look at our work and for your positive comments.